# Towards Homogeneous Lexical Tone Decoding from Heterogeneous Intracranial Recordings

**Di Wu**[1,2]  **Siyuan Li**[1,2]  **Chen Feng**[2]  **Lu Cao**[1]  **Yue Zhang**[1]  **Jie Yang**[1*]  **Mohamad Sawan**[1*]
[1]Westlake University, Hangzhou, China  [2]Zhejiang University, Hangzhou, China

## Abstract

Recent advancements in brain-computer interfaces (BCIs) have enabled the decoding of lexical tones from intracranial recordings, offering the potential to restore the communication abilities of speech-impaired tonal language speakers. However, data heterogeneity induced by both physiological and instrumental factors poses a significant challenge for unified invasive brain tone decoding. Traditional subject-specific models, which operate under a heterogeneous decoding paradigm, fail to capture generalized neural representations and cannot effectively leverage data across subjects. To address these limitations, we introduce **H**omogeneity-**H**eterogeneity **Di**sentangled **L**earning for neural **R**epresentations (H2DiLR), a novel framework that disentangles and learns both the homogeneity and heterogeneity from intracranial recordings across multiple subjects. To evaluate H2DiLR, we collected stereoelectroencephalography (sEEG) data from multiple participants reading Mandarin materials comprising 407 syllables, representing nearly all Mandarin characters. Extensive experiments demonstrate that H2DiLR, as a unified decoding paradigm, significantly outperforms the conventional heterogeneous decoding approach. Furthermore, we empirically confirm that H2DiLR effectively captures both homogeneity and heterogeneity during neural representation learning.

## 1 Introduction

The human language system, with its intricate and expansive syntactic structure, enables rich and complex communication. Decoding spoken language from within human brains has emerged as a significant topic of interest in neuroscience (Anumanchipalli et al., 2019; Willett et al., 2023; Feng et al., 2023; Lu et al., 2023; Liu et al., 2023). The decoding of vocal tone from brain measurements (Lu et al., 2023; Liu et al., 2023) is of particular research interest, due to the prominence of tonal languages, which make up over 60% of the world's languages (Yip, 2002) and are spoken by approximately one-third of the global population (Dryer & Haspelmath, 2013). In these languages, tone plays a critical role in distinguishing lexical meaning at the syllable level.

Mandarin, for instance, is a widely spoken tonal language that has an extensive inventory of over 50,000 characters, with each associated with a syllable composed of an initial, a final, and a tone (Duanmu, 2007). Mandarin features four tones, each characterized by starting pitch height and contour. The same initial and final components can yield entirely different semantic meanings when uttered with different tones, as illustrated in Fig. 1. For instance, the syllable formed by the initial /b/ and the final /a/ can represent vastly different concepts depending on the tone: a high-level tone (Tone 1) signifies 'eight' (八), a rising pitch contour (Tone 2) indicates 'pull' (拔), a low falling-rising tone (Tone 3) means 'handle' (把), and a high falling tone (Tone 4) translates to 'father' (爸). Consequently, precise tone identification is crucial for brain sentence decoding of tonal languages.

Recent studies have shown the feasibility of decoding tones using non-invasive neurophysiological signals such as electroencephalogram (EEG) (Yang et al., 2021; Li et al., 2021a) and more promisingly, intracranial recordings such as electrocorticography (ECoG) (Liu et al., 2023). While EEG provides a non-invasive method, ECoG offers superior spatiotemporal resolution and reduced signal attenuation, leading to better decoding performance and interpretability. Nonetheless, the heterogeneity evoked by

---

*Corrsponding Authors at `yangjie@westlake.edu.cn` and `sawan@westlake.edu.cn`.

both physiological and instrumental factors is a major challenge for invasive brain neural decoding. In particular, physiological variations among subjects and discrepancies in electrode configuration due to diverse electrode implantation conditions prevent unified decoding across subjects. Conversely, developing heterogeneous models for each subject leads to poor generalization ability due to data scarcity, which is primarily due to the cumbersome process of intracranial data acquisition and the inability of participants to wear electrodes for long time periods. Thus, efficiently integrating heterogeneous intracranial recordings from multiple subjects to enable consistent decoding remains a critical challenge.

To tackle this, we propose a two-stage neural representation learning framework called **H**omogeneity-**H**eterogeneity **Di**sentangled **L**earning for neural **R**epresentations (H2DiLR) that captures both homogeneous and heterogeneous information from intracranial recordings across multiple subjects, enabling unified neural decoding. The intuition behind our proposed H2DiLR is straightforward: **although physiological and instrumental differences exist among different subjects (heterogeneity), the same brain regions in different individuals exhibit similar functions during the tone production process (homogeneity)**. Figure 1 provides an overview of H2DiLR in the context of lexical tone decoding. Concretely, in the homogeneity-heterogeneity disentanglement (H2D) stage (first stage), we perform neural feature extraction from the novel perspective of vector quantization (Van Den Oord et al., 2017), where the learned latent neural embedding (neural code) captures discriminative pattern-aware information.

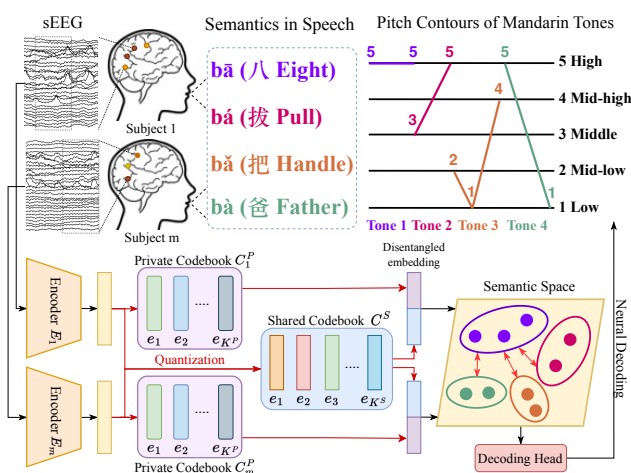

Figure 1: Illustration of H2DiLR for unified lexical tone decoding with sEEG from multiple participants. In the homo-heterogeneity disentanglement (H2D) stage, the continuous latent representations from the encoders are disentangled into H2D representations, which are constructed by discretized code embeddings in a shared codebook (homogeneous tone articulation neural codes) and private codebooks (heterogeneous personalized neural codes). The learned H2D representations are utilized for tone decoding in the second stage.

We further maintain *online-optimizable* codebooks to store these learned neural codes as matching templates for disentangled neural representation learning: one shared codebook for all subjects to capture homogeneity, while private codebooks for each subject capture heterogeneity. The disentangled H2D neural representations are then used for tone decoding in the second stage.

We comprehensively evaluate the effectiveness of H2DiLR using stereoelectroencephalography (sEEG) data collected from multiple participants reading Mandarin materials containing 407 syllables. Compared to previous tone decoding studies that focused on smaller subsets of syllables, our study has greater practical relevance. Extensive experiments demonstrate that H2DiLR outperforms traditional subject-specific decoding paradigms by a significant margin. Moreover, we empirically show that H2DiLR successfully disentangles homogeneous and heterogeneous components in the learned neural representations, as evidenced by visualizations of neural codes and subject categorization tests. H2DiLR, as a general-purpose framework for homogeneity-heterogeneity disentangled neural representation learning, can be applied to various neural decoding applications. Overall, our contributions are as follows:

(1) We introduce a neural representation learning framework, H2DiLR, which disentangles homogeneity and heterogeneity, enabling unified neural decoding across multiple subjects with heterogeneous recordings caused by physiological and instrumental factors;

(2) To the best of our knowledge, we are **the first** to perform unified tone decoding across subjects on a **comprehensive set of 407 Mandarin syllables**, covering nearly all Mandarin characters, providing results of greater practical significance.

(3) Extensive experiments demonstrate the effectiveness of H2DiLR, with an average top-1 tone decoding accuracy improvement of 12.2% over conventional subject-specific models. Moreover, we empirically verify that H2DiLR successfully captures both homogeneous and heterogeneous neural representations.

## 2 RELATED WORK

**Brain Language Decoding.** With the mature application of biomedical circuits and the advancement of deep learning, the in-depth exploration of the perceptual and processing mechanisms of the human brain in response to language and speech has attracted increasing research attention in neuroscience. Recent studies have demonstrated the feasibility of decoding language and speech intentions from both non-invasive (Défossez et al., 2023; Tang et al., 2023; Sereshkeh et al., 2018; Si et al., 2021; Deng et al., 2010; DaSalla et al., 2009; Chi et al., 2011) and invasive neural recordings (Moses et al., 2019; Makin et al., 2020; Moses et al., 2021; Angrick et al., 2019). Early studies primarily focused on the binary classification of language components such as syllables, phonemes, and words from non-invasive brain signals like functional magnetic resonance imaging (fMRI), functional near-infrared spectroscopy (fNIRS), and electroencephalography (EEG)(DaSalla et al., 2009; Deng et al., 2010; Chi et al., 2011). Similar to language decoding, recent work has also demonstrated the feasibility of non-invasively decoding perceived music(Denk et al., 2023). In contrast to non-invasive approaches, intracranial electroencephalography, such as electrocorticography (ECoG), offers superior spatial resolution and signal-to-noise ratios, leading to more robust decoding performance (Willett et al., 2023; Metzger et al., 2023). Anumanchipalli et al. (2019) pioneered sentence-level English decoding using recurrent neural networks (RNNs) to predict Mel-frequency cepstral coefficients (MFCC) from ECoG signals, which were then converted into speech via a vocoder (Anumanchipalli et al., 2019). Beyond non-tonal languages, accurate recognition of lexical tones is crucial for decoding tonal languages from brain signals due to their distinctive pitch articulation and their critical role in differentiating lexical meaning(Yang et al., 2021; Li et al., 2021a; Liu et al., 2023; Lu et al., 2023; Li et al., 2021b). For instance, Liu et al. (2023) employed long short-term memory (LSTM) networks to predict Mel spectrograms from ECoG signals, successfully generating sound waves of syllables /mi/ and /ma/ along with their corresponding four tones using the Griffin-Lim algorithm. However, these studies are often limited by small datasets, as data is typically recorded from patients undergoing neurosurgery, resulting in restricted data availability. Guo & Chen (2022) performed multi-class classification of four tones across vowels /a/, /i/, /o/, and /u/ using manually extracted features from fNIRS. Despite these advancements, existing studies are mostly confined to tone decoding on limited syllables, using small subject-specific datasets. In this work, we expand upon prior research by performing full-spectrum tone decoding across **all possible syllables** in Mandarin Chinese using stereoelectroencephalography (sEEG). Additionally, we propose a unified brain tone decoding framework that integrates neural recordings from multiple subjects.

**Heterogeneity in Neural Representation Learning.** Benefiting from large-scale training corpora, recent breakthroughs in natural language processing (NLP) have demonstrated the exceptional capabilities of large language models as general-purpose task solvers (Brown et al., 2020). Similarly, in computer vision (CV), generative models have shown remarkable performance when trained on extensive datasets (Ho et al., 2020). However, these advances often rely on the assumption that data is independent and identically distributed (IID), which is rarely applicable to neural representation learning due to the heterogeneity inherent in neurological data acquisition. Heterogeneity in neural data manifests in various ways, including physiological and neural differences across individuals, variations in electrode configurations during data collection, and fluctuations in neural activity across recording sessions (Wu et al., 2023). To address these challenges, several researchers have attempted to train universal models—primarily spatiotemporal encoders—by combining data from multiple subjects or sessions to overcome the heterogeneity caused by physiological and neural variability (Zhang et al., 2024; Cai et al., 2023; Jiang et al., 2024; Ye et al., 2024; Fazli et al., 2009; Autthasan et al., 2022; Liu et al., 2021; Ng & Guan, 2024). However, most existing approaches assume homogeneous experimental setups, where the number and placement of electrodes are consistent across subjects. In addressing heterogeneity due to different electrode configurations, LaBraM (Jiang et al., 2024) and MMM (Yi et al., 2023) introduced pre-training strategies based on the standardized 10-20 and 10-10 EEG acquisition systems to mitigate channel compatibility issues during model training. However, these methods are restricted to non-invasive EEG systems,

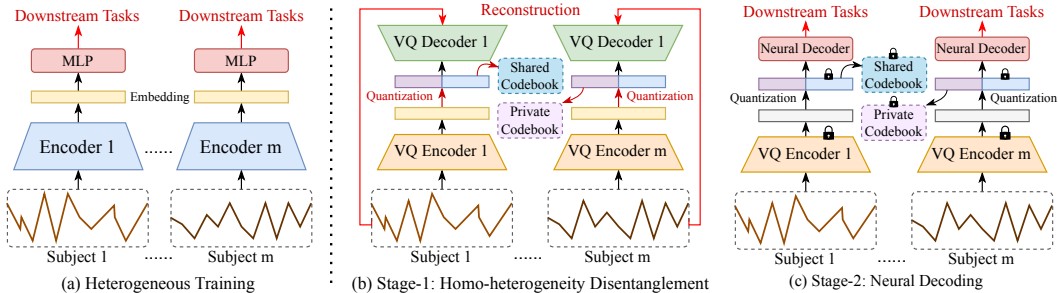

Figure 2: Overview of the proposed H2DiLR learning paradigm compared to the heterogeneous learning paradigm. The VQ encoders, decoders, a shared codebook, and private codebooks are learnable and trained in a self-supervised manner during the H2D stage. It is worth noticing that the VQ decoders are discarded after stage one. In the neural decoding stage, all encoders and codebooks are frozen for H2D representation generation, which is used for further decoding with transformers. The red lines and marks denote loss propagation.

limiting their applicability to broader neural decoding tasks. BIOT (Yang et al., 2023) tackled data heterogeneity by introducing handcrafted embeddings to align neural representations across subjects. However, BIOT primarily focuses on developing encoder architectures for multi-dataset training and does not sufficiently address the nuances of homogeneity and heterogeneity across subjects or datasets. In contrast, this work proposes a novel learning paradigm that disentangles and models both homogeneity and heterogeneity from heterogeneous neural data, enabling unified neural decoding across subjects.

## 3 HOMOGENEITY-HETEROGENEITY DISENTANGLED LEARNING FOR NEURAL REPRESENTATION

We first present the overall learning paradigm of H2DiLR in comparison to other existing learning paradigms in Sec. 3.1. We then propose unified pattern-aware neural tokenization (UPaNT) in Sec. 3.2, the prerequisite for homogeneity-heterogeneity disentanglement (H2D). The details of H2D are elaborated in Sec. 3.3, along with the corresponding model architectures described in Sec. 3.4.

### 3.1 LEARNING PARADIGM OF H2DiLR

Managing heterogeneity caused by physiological or instrumental factors remains a fundamental challenge in neural representation learning. As illustrated in Fig. 2-(a), existing approaches typically adopt a purely heterogeneous training paradigm, wherein subject-specific models are trained independently for each individual. This learning paradigm effectively handles heterogeneity with apparent drawbacks: the lack of unified neural representation learning capability across individuals and poor generalization of learned representations, particularly in invasive scenarios with limited data.

To overcome these limitations, we propose a novel learning paradigm named H2DiLR, which contains an H2D stage and a neural decoding (ND) stage, as shown in Fig. 2. In the H2D stage, we learn neural representations that capture both homogeneous and heterogeneous features by leveraging data from all subjects in an unsupervised, task-agnostic manner through vector-quantized (VQ) style reconstruction(Van Den Oord et al., 2017). We will elaborate on VQ in Sec. 3.2. At a high level, H2D stores homogeneous information in a shared, trainable codebook accessible to all subjects, while subject-specific private codebooks capture heterogeneous information. In the ND stage, the parameters of encoders and cookbooks are frozen for H2D representation extraction. A lightweight transformer is adopted for specific downstream neural decoding tasks in a supervised manner.

### 3.2 UNIFIED PATTERN-AWARE NEURAL TOKENIZATION

**Prerequisites for Homogeneity-heterogeneity Disentanglement.** To successfully disentangle homogeneity and heterogeneity in neural recordings across multiple subjects, neural representation learning must meet two key requirements: **(i) Unify the latent representation space of hetero-**

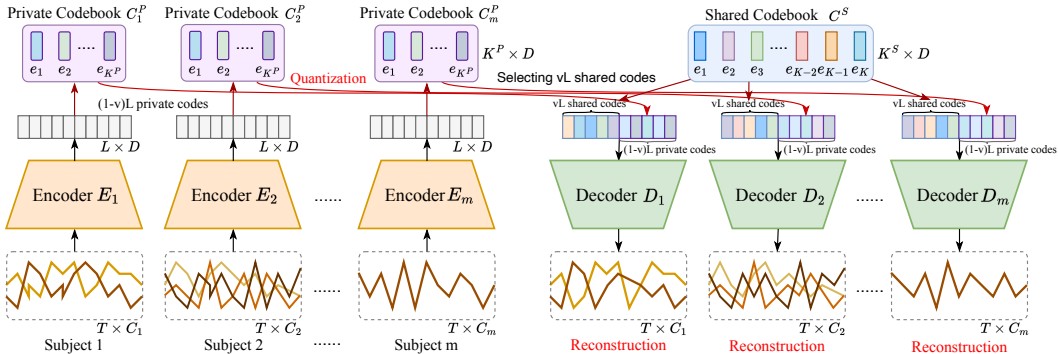

Figure 3: Illustration of the proposed Homogeneity-heterogeneity Disentanglement (H2D) for $m$ subjects, which contains encoders $\{E_i\}_{i=1}^m$, decoders $\{D_i\}_{i=1}^m$, and a shared codebook $\mathbb{C}^S$ and private codebooks $\{\mathbb{C}_i^P\}_{i=1}^m$ for quantization. For each sample, $\nu L$ tokens are selected from its embedding and discretized with the shared codebook, while the rest of $(1-\nu)L$ tokens are quantized by the corresponding private codebook The red lines and marks denote training loss propagation.

**geneous neural recordings for neural decoding.** This property allows the neural representation learning algorithm to handle data heterogeneity and build a single decoding model; **(ii) Extract features with explicit semantic patterns for further H2D.** Take speech decoding for an example. The articulation of speech involves the intricate coordination of oral organs, including the tongue, larynx, vocal tract, and others. During vocalization, these organs exhibit explicit muscle movements associated with specific states under neural control (Jürgens, 2009). For instance, the muscles in the larynx bring the vocal cords closer to realize pitch voicing. Consequently, extracting neural representations associated with the opening and closing of the vocal cords is critical for lexical tone decoding. Likewise, capturing additional critical articulation patterns, such as manner of articulation (MOA) and place of articulation (POA), will benefit neural representation learning.

Driven by the above design principles, we propose unified pattern-aware neural tokenization (UPaNT) to characterize neural patterns (pitch articulation in our case) in a unified manner during neural representation learning based on vector-quantized (VQ) autoencoding (Van Den Oord et al., 2017). It's important to note that UPaNT, without H2D, could be considered a homogeneous training paradigm since it manages to learn neural representations from multiple subjects in a unified manner. The concept of vector quantization (VQ) was initially introduced for learning discrete representations in the context of natural images. Holistically, VQ discretizes the continuous latent representations generated by the encoder by substituting them with the nearest quantized embeddings from an online optimizable codebook and further reconstructs the original input with quantized representations. The discretized embeddings in the codebook demonstrate explicit semantic information (Zhou et al., 2022). In vision representation learning, for example, these discretized embeddings often correspond to interpretable patterns like color and texture.

Consider we collect intracranial recordings from $m$ subjects, $\{\mathcal{S}_i\}_{i=1}^m$, given the same neural decoding task. We assume substantial differences exist among the $m$ sets of recordings $\{\mathcal{S}_i\}_{i=1}^m$ due to electrode configuration variations among subjects. $\mathcal{X}_i \in \mathbb{R}^{N_i \times T \times C_i}$ denotes the set of recordings collected from subject $i$ with different number of data samples $N_i$, different number of channels $C_i$, and the same segment length $T$. The neural recordings $\mathcal{X}$ are first mapped into latent feature, $z = E_i(x, \theta_i) \in \mathbb{R}^{L \times D}$, in the continuous latent space by a set of encoders $\{E_i(\cdot; \theta_i)\}_{i=1}^m$ parameterized by network parameters $\{\theta_i\}_{i=1}^m$. A finite VQ codebook of $K$ key-value pairs, $\mathcal{C} = \{(k, e(k))\}_{k \in [K]}$, where each code $k$ owns its *learnable* code embedding $e(k) \in \mathbb{R}^D$, can discretize each token in $z$ by a quantization function $\mathcal{Q}(\cdot, \cdot)$:

$$M_j = \mathcal{Q}(z_j; \mathcal{C}) = \text{argmin}_{k \in [K]} \|z_j - e(k)\|_2, \qquad (1)$$

where $z_j \in \mathbb{R}^{1 \times D}$ denotes the $j^{th}$ token of $z$ with $1 \leq j \leq L$, $M_j \in [K]^L$ is code mapping indices.

With assigned $M$, the latent feature $z$ can be indexed and quantized to the VQ embedding by the closest 1-of-K embedding vectors in the codebook $\mathcal{C}$ as $\hat{z}_j = e(M_j)$. Then, a set of decoders $\{D_i(\cdot; \psi_i)\}_{i=1}^m$ parameterized by $\{\psi_i\}_{i=1}^m$ maps the VQ embedding $\hat{z}$ back to the input space to

reconstruct the original neural recordings $\hat{x}$:

$$\hat{x} = D_i(\hat{z}; \psi_i) = D_i(e(M); \psi_i), \tag{2}$$

Since differentiation through the quantization in Eq. (1) is ill-posed during gradient backward, the straight-through-estimator (STE) (Bengio et al., 2013) is employed as the approximation, *i.e.,* $(e(M_j) + z_j) - z_j$. Overall, the learning objective of VQVAE on $\mathcal{X}$ includes $\mathcal{L}_{\text{rec}}$ for reconstruction of autoencoders, $\mathcal{L}_{\text{code}}$ for the codebook learning, and $\mathcal{L}_{\text{commit}}$ for quantization:

$$\mathcal{L}_{\text{VQ}} = \sum_{i=1}^{m} \underbrace{\|\mathcal{X}_i - \hat{\mathcal{X}}_i\|^2}_{\mathcal{L}_{\text{rec}}} + \underbrace{\|\text{sg}[\mathcal{Z}_i] - \hat{\mathcal{Z}}_i\|_2^2}_{\mathcal{L}_{\text{code}}} \\ + \beta \underbrace{\|\mathcal{Z}_i - \text{sg}[\hat{\mathcal{Z}}_i]\|_2^2}_{\mathcal{L}_{\text{commit}}}, \tag{3}$$

where $\text{sg}[\cdot]$ denotes the stop gradient operation and $\beta > 0$ is a hyper-parameter set to 0.25 by default.

### 3.3 HOMOGENEITY-HETEROGENEITY DISENTANGLEMENT

We formalize homogeneity-heterogeneity disentanglement (H2D) based on UPaNT. To capture the homogeneous and heterogeneous neural representations, we first define a shared codebook $\mathbb{C}^S = \{(k, e^S(k))\}_{k \in [K^S]}$ to maintain the common neural embeddings with high-level semantic patterns extracted from all subjects under the same task. We also keep $m$ private codebooks $\{\mathbb{C}_i^P\}_{i=1}^m$ to encode the unique patterns of $m$ different subjects, where $\mathbb{C}_i^S = \{(k, e_i^P(k))\}_{k \in [K^P]}$. Given the $n^{th}$ neural recording sample $x_{n,i}$ from subject $i$, we first rank all tokens in the continuous encoded feature $z_{n,i}$ with the similarity between the tokens and the corresponding selected nearest code embeddings in the shared codebook $\mathbb{C}^S$, $R_{n,i} = \text{rank}(\{\|\mathcal{Z}_{n,i,j} - e^S(M_{n,i,j})\|\}_{j=1}^L)$, where $\text{rank}(\cdot)$ denotes the ascending ranking. Based on the ranking result, we split the tokens into the homogeneous group and the heterogeneous group, where the top-$\nu L$ tokens (the most similar $k^S$ tokens for $\mathbb{C}^S$) are quantized with the shared codebook, while the rest $\nu L$ tokens quantized using the corresponding private codebook of subject $i$. The partition factor $\nu \in [0, 1]$ to balance homogeneous and heterogeneous representations. The H2D quantization can be formulated as:

$$\hat{\mathcal{Z}}_{n,i,j} = \begin{cases} e^S(M_{n,i,j}), & R_{n,i}(j) \leq \nu L \\ e_i^P(M_{n,j}). & R_{n,i}(j) > \nu L \end{cases} \tag{4}$$

The shared codebook $\mathbb{C}^S$ and the private codebooks $\mathbb{C}_i^P$ are then updated with two different strategies. The shared codebook is updated by:

$$\hat{\mathcal{Z}}_{n,i,j} = (1 - \alpha)\mathcal{Z}_{n,i,j} + \alpha\hat{\mathcal{Z}}_{n,i,j}, \quad R_{n,i}(j) \leq \nu L, \tag{5}$$

where $\alpha$ is the momentum coefficient for the exponential moving average (EMA). Note that the EMA update of the codebook in Eq. (5) reduces the training instability caused by updating conflicts of the certain code from latent tokens of different subjects (Razavi et al., 2019). The private codebooks are updated as follows:

$$\mathcal{L}_{n,i}^{\text{pri}} = \sum_{j=1}^{L} \|\text{sg}[\mathcal{Z}_{n,i,j}] - \hat{\mathcal{Z}}_{n,i,j}\|_2^2, \quad R_{n,i}(j) > \nu L. \tag{6}$$

Note that $\mathcal{L}_{n,i}^{\text{pri}}$ is applied only to the rest $\nu L$ tokens. We use the commitment loss to align latent embeddings to the relevant codes as the same design as in VQVAE:

$$\mathcal{L}_{i,j}^{\text{comm}} = \sum_{j=1}^{L} \|\mathcal{Z}_{n,i,j} - \text{sg}[\hat{\mathcal{Z}}_{n,i,j}]\|_2^2. \tag{7}$$

Meanwhile, our proposed H2D adopts unsupervised autoencoding as the pretext task during neural representation learning using the reconstruction loss from Eq. (3). Overall, the learning objective of H2D is defined as:

$$\mathcal{L}_{\text{H2D}} = \sum_{i=1}^{m} \sum_{j=1}^{N_i} \mathcal{L}_{i,j}^{\text{rec}} + \mathcal{L}_{n,i}^{\text{pri}} + \beta\mathcal{L}_{i,j}^{\text{comm}}. \tag{8}$$

Table 1: Comparison with heterogeneous (supervised and self-supervised methods), homogeneous (UPaNT), and heterogeneity-homogeneity disentanglement approach (H2DiLR) for brain tone decoding. CL denotes contrastive learning, and MM denotes masked modeling on sEEG. Top-1 accuracy (%) for fine-tuning evaluations are reported. **Bold** and underline denote the best and second best.

| Paradigm | Method | Pre-training | Backbone | Subject 1 | Subject 2 | Subject 3 | Subject 4 | Avg. |
|---|---|---|---|---|---|---|---|---|
| Heterogeneous | SPaRCNet | - | CNN | 34.94±2.17 | 36.73±5.88 | 27.10±3.22 | 27.10±1.43 | 31.47±3.16 |
| | FFCL | - | CNN+LSTM | 37.47±2.40 | 37.71±5.42 | 26.61±3.18 | 29.47±1.69 | 32.82±3.17 |
| | TS-TCC | CL | CNN | 39.59 ±1.68 | 41.47 ±2.78 | 28.82 ±1.75 | 32.33 ±2.17 | 35.55 ±2.09 |
| | CoST | CL | CNN | 43.95 ±1.48 | 40.41 ±4.77 | 31.02 ±1.00 | 34.53 ±2.39 | 37.47 ±2.41 |
| | ST-Transformer | - | Transformer | 39.02 ±2.39 | 37.22 ±4.68 | 27.51 ±2.11 | 29.63 ±2.98 | 33.35 ±3.04 |
| | NeuroBERT | MM | Transformer | 42.20 ±1.86 | 43.10 ±2.76 | 29.80 ±1.98 | 32.65 ±2.89 | 36.94 ±2.37 |
| | **H2DiLR (ours)** | $\nu = 0$ | Transformer | 43.61 ±2.12 | 42.15 ±1.63 | 34.26 ±1.51 | 35.92 ±1.43 | 38.98 ±1.67 |
| Homogeneous | BIOT | MM | Transformer | 42.45 ±6.99 | 40.90 ±5.87 | 33.55 ±2.95 | 33.88 ±1.89 | 37.47 ±2.26 |
| | **H2DiLR (ours)** | **UPaNT only** | Transformer | 45.47 ±3.04 | 44.65 ±1.84 | 35.59 ±2.37 | 35.76 ±1.18 | 40.37 ±2.11 |
| Disentanglement | **H2DiLR (ours)** | **UPaNT+H2D** | Transformer | **49.06** ±**2.15** | **47.84** ±**1.81** | **39.18** ±**1.68** | **38.61** ±**1.49** | **43.67** ±**1.78** |

For simplicity, we set the homogeneous neural representation component to have the same number of tokens as the heterogeneous component, *i.e.*, $\nu = 0.5$. The size for the shared codebook $K^S$ is set to be $m \times K^P$, resulting in a total code size of $K = 2mK^P$ from all codebooks.

## 3.4 MODEL ARCHITECTURE

The architectures of VQ encoders, VQ decoders, and neural decoders can take on any arbitrary design, provided that they effectively accomplish the reconstruction and disentanglement tasks in the H2D stage, as well as the decoding task in the ND stage. In our work, we discover that very lightweight architectures can achieve promising results in lexical tone decoding. The VQ encoders consist of five 1-D convolution layers with a kernel size of four and a stride of two. Due to varying input channel numbers across different subjects, the channel dimension is first mapped to a uniform count of 64 by the stem layer and then progressively increased to 512 before being reduced back to 256. The VQ-decoder adopts a symmetrical design to the VQ-encoder, wherein 1-D convolution layers are substituted with transpose convolution layers. For the ND stage, we adopt a lightweight transformer as the neural decoder with the multi-head self-attention (MSA) structure with pre-normalization and residual connection as in ViT (Dosovitskiy et al., 2020). The patch embedding is performed by a 1-D convolution stem layer with a kernel size of five and stride five to ensure non-overlapping patch embedding. The output dimension of the stem layer is 128, and the hidden dimension of the Feed Forward Network (FFN) is set to 512. A fully connected layer is added to map the output to desired decoding output formats. A detailed description of network architectures is provided in Tab. A2.

## 4 EXPERIMENTS

### 4.1 EXPERIMENTAL SETUP

**Data acquisition for tone decoding.** We recruited four participants undergoing epilepsy monitoring with stereo electroencephalograph (sEEG) electrodes implanted in the Second Affiliated Hospital of Zhejiang University School of Medicine to participate in this study. The distribution of electrodes for all four participants is shown in Fig. 4. The experimental protocol was approved by the review board of the Second Affiliated Hospital of Zhejiang University School of Medicine. All participants gave their written, informed consent prior to testing. For each participant, we selected contacts related to speech and excluded those located in the visual cortex and white matter. All participants are asked to read 407 monosyllabic Mandarin characters, each with a unique tone, three times, covering all common pronunciations of Mandarin characters. To make the pronunciation process of the participants as similar as possible to normal speech, carrier words are added before and after each character to form a sentence. Thus, in each trial, participants are required to read a complete sentence containing the target syllable. The collected sEEG signals are downsampled to 1000 Hz with power line interference removed. See Appendix A for details.

**Evaluation Protocols.** We assess decoding performance using the top-1 accuracy (Acc). Data for each participant is divided into an 80% training set and a 20% testing set, with 20% of the training data further allocated for validation. We conducted the experiment five times using different random seeds and reported the mean and standard deviation. Refer to Appendix B for implementation details.

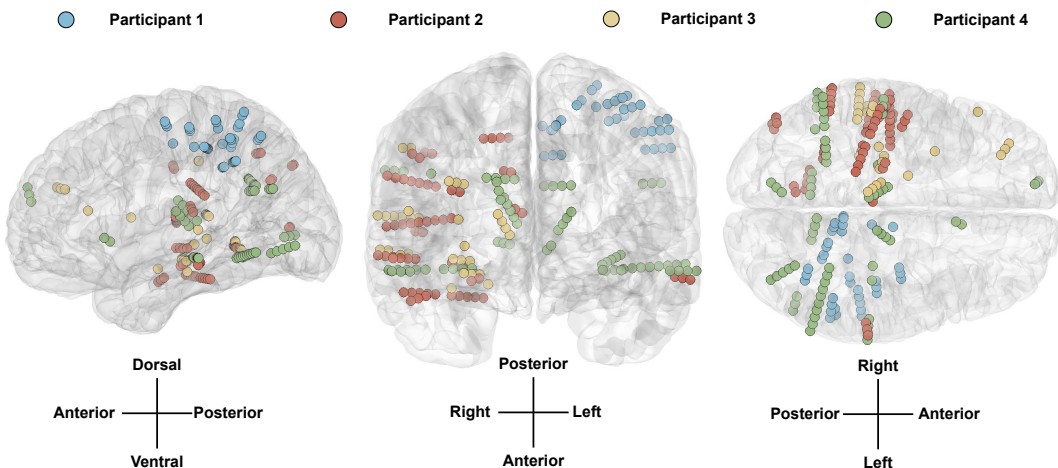

Figure 4: The anatomy of four participants mapped onto the standard Montreal Neurological Institute template brain, with directions indicated. All chosen contacts for Participant 1 are situated in the left hemisphere, while those for Participants 2 and 3 are in the right hemisphere. Participant Four's selected contacts are distributed across both hemispheres. The brain structures housing these chosen contacts cover most regions associated with speech, including the Superior Temporal Gyrus (STG), Middle Temporal Gyrus (MTG), ventral Sensorimotor Cortex (vSMC), Inferior Frontal Gyrus (IFG), Precentral Gyrus, and Postcentral Gyrus. Additionally, signals from several subcortical structures are recorded, such as the Thalamus, Hippocampus, Insula, and Amygdala.

## 4.2 DECODING PERFORMANCE COMPARISON

**Baseline.** We select representative approaches of both the heterogeneous learning and homogeneous learning paradigms as baselines in comparison to our proposed homogeneity-heterogeneity disentangled learning paradigm (H2DiLR). For the heterogeneous learning paradigm, we consider three supervised approaches featuring diverse backbone designs (Jing et al., 2023; Li et al., 2022; Song et al., 2021). Additionally, we include three methods utilizing contrastive and masked modeling pre-training (Woo et al., 2022; Eldele et al., 2021; Wu et al., 2022) for a fair comparison, as the H2D stage of our H2DiLR can be regarded as a pre-training stage. Furthermore, we consider H2DiLR with $\nu = 0$, which indicates no shared codebook, as another variant within the heterogeneous learning paradigm. For the homogeneous learning paradigm, we examine BIOT (Yang et al., 2023) and the UPaNT component of H2DiLR.

**Results comparison.** We compare the performance of H2DiLR with baselines in Tab. 1. It is observed that baselines of the heterogeneous training paradigm with pre-training outperform those with no pre-training. Heterogeneous approaches generally suffer from the scarcity of data from individual subjects, where self-supervised pre-training methods prove more effective in capturing neural representations and subsequently improving decoding performance. Although BIOT can integrate data from multiple individuals for homogeneous decoding, its performance does not significantly outperform heterogeneous approaches and even yields worse results compared to heterogeneous decoding methods with pre-training. This is due to the fact that BIOT eliminates data heterogeneity in terms of channel count, sampling rate, and data length but fails to explore and utilize the inherent heterogeneity and heterogeneity embedded in the brain recordings from multiple subjects. Compared to BIOT and heterogeneous baselines, our proposed UPaNT demonstrates a better decoding performance due to its pattern-aware feature extraction capabilities. With H2D, our proposed H2DiLR further improves the decoding performance on top of UPaNT by disentangling the homogeneous and heterogeneous neural representations, leading to a significant gain over existing approaches.

## 4.3 PRELIMINARY VERIFICATION OF H2D

**Homogeneous representations correlate with tonal decoding.** We first evaluate whether the homogeneous representations learned by our proposed H2D capture unified neural features across subjects in the context of tone decoding. Specifically, we visualize the neural codes learned in the

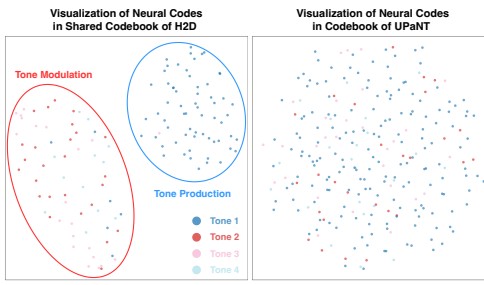

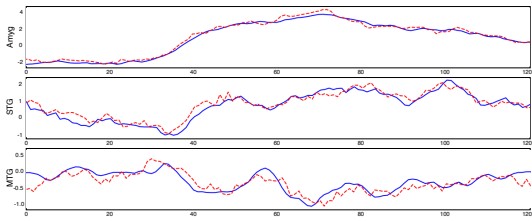

Figure 6: Reconstructed sEEG from the Amygdala (Amyg), Superior Temporal Gyrus (STG), and Middle Temporal Gyrus (MTG) in the H2D stage, showing that H2D achieves disentangled quantization while preserving heterogeneous information.

Figure 5: Comparison of UMAP visualization of neural codes learned by H2D and UPaNT w.r.t different tone classes.

shared codebook of H2D in comparison to the codes learned by UPaNT using UMAP (McInnes et al., 2018). Each neural code is assigned the tone class to which it is most frequently mapped during the quantization process. As shown in Fig. 5, The neural codes learned by UPaNT w.r.t. each tone class are scattered with no clear pattern, while the distribution of neural codes from the shared codebook of H2D demonstrates a clear separation between the Tone 1 and other tones. Since the H2D training stage is unsupervised training, we hypothesize that such clustering effect might correlate with the two key functions of the larynx: pitch voicing and modulation (lowering and rising pitch) (Lu et al., 2023). This is consistent with the fact that Tone 2,3 and 4 involve pitch change but not Tone 1. This finding and superior decoding performance, as seen in Tab. 1, prove that H2D better captures tone-related neural features by disentangling homogeneity from heterogeneity during neural representation learning.

**Heterogeneous representations capture subject-specific information.** To verify whether the heterogeneous representations learned by our proposed H2D capture the heterogeneity, we propose a subject classification task to determine whether the learned heterogeneous representation extracts sufficiently discriminative information to classify sEEG signals from different subjects. In particular, we adopt the same network architecture utilized for tone classification in this test and follow the same experimental setup as described in Sec. 4.1. As shown in Tab. 2, using heterogeneous representations yields a much higher subject classification performance but an inferior tone decoding performance than the homogeneous representation, suggesting that the heterogeneous representation extracts more subject-specific information rather than tone-related neural features. We also show in Fig. 6 that heterogeneous representations capture instrumental heterogeneity to reconstruct sEEG from different brain regions across subjects.

## 4.4 ABLATION STUDY

This section ablates three key designs and the scaling effect of the network parameters and subject count. The average top-1 accuracy for tone decoding on all subjects is reported, and subject classification tasks are designed using the same experimental setup as Sec. 4.1. Furthermore, additional experiments on diverse neural decoding tasks demonstrate the effectiveness of the proposed H2DiLR as a general neural representation learning framework, detailed in Sec. C.1.

**Ablation on $\nu$.** We first study how the heterogeneous and homogeneous representation partition ratio $\nu$ influences the learned representations in terms of tone decoding and subject classification tasks. By default, H2D sets $\nu = 0.5$, and a higher value of $\nu$ indicates more homogeneous information captured during representation learning. With a $\nu$ value of 1, H2D degenerates to the UPaNT. It is observed in Tab. 2 that a smaller value of $\nu$ leads to better subject classification performance with heterogeneous representation, indicating more heterogeneity captured. Also, a smaller value of $\nu$ leads to inferior tone decoding performances with homogeneous representation. Results show that $\nu = 0.5$ strikes a good balance for H2D.

**Ablation on codebook size and dimension.** By default, H2DiLR utilizes a codebook size and dimension of 256 and a fixed $\nu$ value of 0.5. We study how value changes in size and dimension affect the reconstruction performance in the H2D stage and the tone decoding performance in the ND stage. We use the mean squared error (MSE) as the reconstruction performance metric and report results in

Table 2: Ablation on H2D codebook partition ratio $\nu$. Note that 'Homo-' and 'Hetero-' denote using homogeneous and heterogeneous representation, respectively.

| $\nu$ | Subject | | Tone | | |
|---|---|---|---|---|---|
| | Homo- | Hetero- | Homo- | Hetero- | Homo+Hetero |
| 0.25 | 71.7 | **85.5** | 35.2 | 36.9 | 42.78 |
| 0.5 | 71.9 | 82.6 | 38.0 | **37.9** | 43.67 |
| 0.75 | 73.1 | 80.7 | 38.5 | 35.4 | 41.31 |
| 1.0 | **73.5** | - | **40.4** | - | 40.37 |

Table 3: Ablation study on the codebook dimension (dim.) and the total codebook size in the H2D stage of H2DiLR.

| Method | UPaNT | | | | UPaNT+H2D | | | |
|---|---|---|---|---|---|---|---|---|
| Value | Code dim. | | Code size | | Code dim. | | Code size | |
| | MSE | Acc. | MSE | Acc. | MSE | Acc. | MSE | Acc. |
| 64 | 0.096 | 38.86 | 0.103 | 38.37 | 0.092 | 39.76 | 0.094 | 39.27 |
| 128 | 0.081 | 39.76 | 0.079 | 39.10 | 0.079 | 42.78 | 0.077 | 42.37 |
| **256** | **0.079** | **40.37** | **0.079** | **40.37** | **0.076** | **43.67** | **0.076** | **43.67** |
| 512 | 0.078 | 40.08 | 0.071 | 40.41 | 0.073 | 43.35 | 0.071 | 43.59 |

Table 4: Scaling-up network parameters (embedding dimension and layers) for better performances.

| Method | Encoder | Transformer | Subject 1 | Subject 2 | Subject 3 | Subject 4 | Avg. |
|---|---|---|---|---|---|---|---|
| UPaNT+H2D | ConvNet-4-64 | Transformer-4-128 | 49.06 ±2.15 | 47.84 ±1.81 | 39.18 ±1.68 | 38.61 ±1.49 | 43.67 ±1.78 |
| UPaNT+H2D | ConvNet-4-128 | Transformer-4-128 | 50.38 ± 1.81 | 48.57 ± 1.69 | 40.04 ± 1.94 | 39.57 ± 1.68 | 44.64 ± 1.72 |
| UPaNT+H2D | ConvNet-4-64 | Transformer-8-256 | 52.12 ± 1.95 | 49.62 ± 1.54 | 41.23 ± 1.73 | 40.36 ± 1.25 | 45.82 ± 1.63 |
| UPaNT+H2D | **ConvNet-4-128** | **Transformer-8-256** | 52.27 ± 2.02 | 49.56 ± 1.87 | 41.39 ± 1.60 | 40.58 ± 1.54 | 45.95 ± 1.65 |

Table 5: Scaling-up participants count to verify the learned representations across participants.

| Participant Count | Participant 1 | Participant 2 | Participant 3 | Participant 4 |
|---|---|---|---|---|
| H2DiLR ($m = 2$) | 46.25 ±1.89 | 45.73 ±1.83 | - | - |
| H2DiLR ($m = 3$) | 48.16 ± 2.04 | 46.25 ± 1.61 | 38.22 ± 2.05 | - |
| H2DiLR ($m = 4$) | 49.06 ±2.15 | 47.84 ±1.81 | 39.18 ±1.68 | 38.61 ±1.49 |

Tab. 3 with the default settings greyed out. We observe that codebook sizes and dimensions larger than 128 lead to quite similar reconstruction performances, while a size and dimension of 64 yield much worse reconstruction performance due to the limited expressive capacity. It is worth noticing that a better reconstruction does not necessarily mean a better decoding performance. We hypothesize that neural codes that reconstruct signal the best might not be optimal for neural decoding.

**Ablation on scaling-up network parameters.** To verify the scaling-up effects of network parameters, we double the embedding dimension and the depth of network blocks. ConvNet-N-D represents a ConvNet encoder with N layers and a D-dimensional input embedding, with ConvNet-4-64 being the baseline setup as in Tab. 1. For the Transformer, Transformer-N-D indicates a model with N layers and a D-dimensional embedding, with Transformer-4-128 being the baseline. As demonstrated in Tab. 5, enlarging the Transformer model by both deepening and widening it (as in the Transformer-8-256 configuration) leads to more significant performance improvements compared to merely widening the encoder. Moreover, the combined scaling of both the encoder and the Transformer models to their larger configurations yields the highest performance enhancements across all participants tested.

**Ablation on scaling-up subject count.** As demonstrated in Tab. 1, we presented the decoding performance improvement achieved by H2DiLR using data from all four participants ($m = 4$). To provide a better understanding of how varying subject numbers affect model performance, we conduct an additional ablation study focusing on varying the number of participants (ranging from 2 to 4). Without loss of generality, we analyzed decoding performance with participants 1 and 2 for $m = 2$, and participants 1, 2, and 3 for $m = 3$. The preliminary results, as seen from Tab. 5, indicate a consistent trend: an increase in the number of participants leads to improved decoding performance.

## 5 CONCLUSION AND LIMITATION

This paper presents homogeneity-heterogeneity disentangled learning for neural representations (H2DiLR), which disentangles and models the homogeneity and heterogeneity from intracranial recordings of multiple subjects for neural decoding. Extensive tone decoding experiments on collected sEEG of multiple participants reading Mandarin suggest that H2DiLR enables unified tone decoding across subjects with superior performance compared to existing methods. We list three potential limitations of this work: (1) The generalization of the trained H2DiLR on unseen new subjects, which is of great practical value, remains to be explored. (2) Additional interpretability of the learned neural codes is required. Establishing mapping between learned neural codes with functionalities of different brain regions for better interpretability remains a promising future research direction. (3) Due to the complexity of intracranial recording data acquisition, the current constraints prevent us from expanding our subject pool further.

ACKNOWLEDGEMENT

This work was supported by STI2030-Major Projects (2022ZD0208805), the National Natural Science Foundation of China (Grant No. 623B2085), and the "Pioneer" and "Leading Goose" R&D Program of Zhejiang (2024C03002).

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

APPENDIX

## A  TONE DECODING DATA AQUISITION

This work proposes to use stereotactic electroencephalography (sEEG) as a means of collecting intracranial neurophysiological data for tone decoding. sEEG is a novel international technique that has emerged in recent years as a localization method for epileptic foci. This technique simultaneously records the brain electrical activity of epilepsy patients in different cranial structures, involving many brain networks associated with advanced cognitive functions, such as the hippocampus, frontal lobe, amygdala, cingulate gyrus, parietal lobe, and precuneus. Patients can undergo assessments and tests of advanced cognitive functions during the interictal period (when there are no symptoms of epilepsy, and the patient's behavior is indistinguishable from that of a normal, healthy person). Therefore, sEEG is currently recognized as an invasive method for studying advanced cognitive functions in the human brain and does not pose additional risks, such as cranial trauma for patients during the research process.

Table A1: Participant characteristics, including sex, age, education level, and handedness

| Subject Number | Subject 1 | Subject 2 | Subject 3 | Subject 4 |
|---|---|---|---|---|
| sex | male | male | female | female |
| age | 20 | 25 | 19 | 19 |
| education | Bachelor's | High school | Bachelor's | High school |
| handedness | right handed | right handed | right handed | right handed |

In this study, we recruited four participants undergoing epilepsy monitoring with stereo electroencephalograph (sEEG) electrodes implanted in the Second Affiliated Hospital Zhejiang University School of Medicine to participate in this study. The distribution of electrodes for all four participants is shown in Fig. 4. We also provide basic information on participants, including sex, age, education level, and handedness in Tab. A1. The experimental protocol was approved by the institutional review board of the Second Affiliated Hospital Zhejiang University School of Medicine. All participants gave their written, informed consent before testing. For each participant, we selected channels related to speech and excluded those located in the visual cortex and white matter. All participants are asked to read 407 monosyllabic Mandarin characters, each with a unique tone, three times, covering all common pronunciations of Mandarin characters. A comprehensive list of characters and their corresponding syllables is provided in Tab. A5 through Tab. A8. To make the pronunciation process of the participants as similar as possible to normal speech, carrier words are added before and after each character to form a sentence. Therefore, in each trial, participants must read a complete sentence containing the target syllable. For example, if the target syllable is 'ài', the sentence presented and to be read by the participant is "我读爱三遍" (I read love three times). Our reading material is carefully designed by Mandarin linguists to cover as many pronunciation phenomenons as possible to enable a brain decoding algorithm with generality. The reading material contains syllables of four tones subject to uniform distribution. It is worth noticing that there is an additional neutral tone, which has no specific pitch contour and is typically used on less emphasized syllables where the preceding syllable primarily influences its pitch. It is not considered a fifth tone in addition to the four tones but rather a special tonal variation of Tone 4, which physically manifests itself as a shortening of the length of the tone and a weakening of the strength of the tone. Consequently, we do not treat the neutral tone as an additional fifth class.

Neural signals were recorded using a multi-channel electrophysiological recording device, specifically the Neurofax EEG-1200 produced by Nihon Kohden Corporation, Japan, and were recorded at a sampling rate of 2000Hz. Each channel was subjected to visual and quantitative inspection for artifacts or excessive noise. We also record the audio signals of participants to provide time stamps for slicing the targeted syllable from the sentence being read. The neural signals were low-pass filtered at 300 Hz and notch filtered at 50 Hz, 100 Hz, 150 Hz, and 200 Hz to remove power line interference. The signals were then downsampled to 1000 Hz. Signal fragments are padded to the maximum length of 1000.

Table A2: Detailed architecture specifications for models utilized in the H2D stage and ND stage in this work. AvgPool stands for average pooling in ConvNet and rel. pos. denotes relative positional encoding in Transformers.

| | H2D Stage ConvNet | ND Stage Transformer |
|---|---|---|
| Stem | 4×1, 64, stride 2 | 5×1, 128, stride 5 |
| Block 1 | $\begin{bmatrix} 4\times1,\ 128 \\ \text{AvgPool, stride 2} \end{bmatrix} \times 1$ | $\begin{bmatrix} \text{MSA, 128, rel. pos.} \\ 1\times1,\ 512 \\ 1\times1,\ 128 \end{bmatrix} \times 1$ |
| Block 2 | $\begin{bmatrix} 4\times1,\ 256 \\ \text{AvgPool, stride 2} \end{bmatrix} \times 1$ | $\begin{bmatrix} \text{MSA, 128} \\ 1\times1,\ 512 \\ 1\times1,\ 128 \end{bmatrix} \times 1$ |
| Block 3 | $\begin{bmatrix} 4\times1,\ 512 \\ \text{AvgPool, stride 2} \end{bmatrix} \times 1$ | $\begin{bmatrix} \text{MSA, 128} \\ 1\times1,\ 512 \\ 1\times1,\ 128 \end{bmatrix} \times 1$ |
| Block 4 | $[4\times1,\ 256] \times 1$ | $\begin{bmatrix} \text{MSA, 128} \\ 1\times1,\ 512 \\ 1\times1,\ 128 \end{bmatrix} \times 1$ |
| # params. | $1.55 \times 10^6$ | $0.958 \times 10^6$ |

# B  IMPLEMENTATION DETAILS

All our experiments are implemented by PyTorch and conducted on workstations with NVIDIA A100 GPUs. For all baselines (Woo et al., 2022; Eldele et al., 2021; Wu et al., 2022; Yang et al., 2023) with open-source code, we reproduce results using the official code and setups provided by the authors. We use the implementation provided by BOIT (Yang et al., 2023) for baselines with no publicly available codes (Jing et al., 2023; Li et al., 2022; Song et al., 2021). For baselines with no pre-training involved, we train each model with a fixed epoch number of 40. We use AdamW as the optimizer with a base learning rate of 3e-4. Cosine annealing decay is adopted for learning rate scheduling. We set momentum factors $\beta_1, \beta_2 = 0.9, 0.999$ with a weight decay of 0.01. The batch size is set to 32. A dropout ratio of 0.1 is adopted. For the pre-training stage of baselines with pre-training, we follow the experimental setup provided by the authors. Next, we give the pre-training details for our proposed UPaNT and H2D. Following masked modeling methods (Wu et al., 2022; Li et al., 2023) with VQ tokenizers (Bao et al., 2022; Li et al., 2024), we use AdamW (Loshchilov & Hutter, 2019) optimizer with a base learning rate of 5e-5, $\beta_1, \beta_2 = 0.9, 0.999$, and the weight decay of 0.01. A cosine learning rate schedule is also adopted. We train a fixed number of 1000 epochs during pre-training with batch size 32. For UPaNT and H2D, we use $\beta = 0.25$ and $\nu = 0.5$ with a total codebook size of 256, *i.e.*, $K = 256$ for UPaNT only and $K = 32$ for H2D. For the fine-tuning stage of all approaches with pre-training, we adopt the same training setup as previously described for baselines with no pre-training, with the only difference being a learning rate of 5e-5.

# C  ADDITIONAL ABLATION STUDY

## C.1  VERIFICATION OF H2DiLR ON SEIZURE PREDICTION

Our proposed Homogeneity-Heterogeneity Disentangled Learning for Neural Representations (H2DiLR) is primarily inspired by the challenges of heterogeneous neural decoding from intracranial recordings, particularly in tasks involving the decoding of lexical tones where electrode placement and count may vary across subjects. To showcase its versatility, we extend the application of H2DiLR beyond its initial focus. As a proof of concept, we demonstrate its effectiveness as a general neural representation learning framework in a different neural decoding task—epilepsy seizure prediction. This demonstrates H2DiLR's adaptability and potential applicability across a range of complex neural decoding challenges.

**Dataset and Pre-processing.**    The publicly available CHB-MIT (Shoeb & Guttag, 2010) dataset includes 637 recordings from 23 epileptic patients. Signals were collected from 22 bi-polar electrodes placed according to the international 10-20 system of 256 Hz sampling frequency and 16-bit resolution. Clinical experts annotated the start and end times of seizures through visual inspection. Only patients

Table A3: Comparison with heterogeneous (supervised and self-supervised methods), homogeneous (UPaNT), and heterogeneity-homogeneity disentanglement approach (H2DiLR) for epileptic seizure prediction. CL denotes contrastive learning, and MM denotes masked modeling with sEEG signals. Top-1 accuracy (%) for fine-tuning evaluations are reported. **Bold** and underline denote the best and second best results.

| Paradigm | Method | Pre-training | Backbone | Acc(%) ↑ | Sen(%) ↑ | FPR(/h) ↓ |
|----------|--------|-------------|----------|----------|----------|-----------|
| Heterogeneous | SPaRCNet (Jing et al., 2023) | - | CNN | 84.33±3.26 | 86.74±2.94 | 0.18±0.07 |
| | FFCL (Li et al., 2022) | - | CNN+LSTM | 85.21±2.58 | 89.37±5.42 | 0.16±0.02 |
| | ST-Transformer (Song et al., 2021) | - | Transformer | 85.74 ±2.39 | 91.84 ±3.26 | 0.22 ±0.05 |
| | TS-TCC (Eldele et al., 2021) | CL | CNN | 86.16 ±1.68 | 90.42 ±2.78 | 0.20 ±0.04 |
| | CoST (Woo et al., 2022) | CL | CNN | 88.45 ±1.98 | 92.19 ±4.77 | 0.18 ±0.04 |
| | NeuroBERT (Wu et al., 2022) | MM | Transformer | 89.36 ±2.12 | 92.54 ±2.98 | 0.15 ±0.06 |
| Homogeneous | BIOT (Yang et al., 2023) | MM | Transformer | 87.22 ±2.41 | 90.53 ±1.87 | 0.17 ±0.04 |
| | **H2DiLR (Ours)** | **UPaNT only** | Transformer | 91.19 ±2.34 | 93.67 ±1.95 | 0.13 ±0.03 |
| Disentanglement | **H2DiLR (Ours)** | **UPaNT+H2D** | Transformer | **93.97** ±2.16 | **94.82** ±1.27 | **0.10** ±0.02 |

whose channel configurations remained consistent throughout data collection were included in our study. For seizure prediction, a 'lead seizure' is defined as one occurring after a seizure-free interval of at least four hours. We set the seizure prediction horizon (SPH) and preictal interval length (PIL) at 5 and 30 minutes, respectively. Interictal samples are generated using a 20-second non-overlapping sliding window, and preictal samples with a 20% overlapping 20-second window to address sample imbalance. All samples were normalized channel-wise by subtracting the mean and dividing by the standard deviation to standardize the input data for further analysis.

We follow the same experimental setup as described in Sec. B and report the averaged accuracy (Acc), sensitivity (Sen), and false positive rate (FPR) across all subjects over five random seeds in Tab. A3. The prediction performance observed for the baseline approaches mirrors the patterns seen in the lexical tone decoding task, where heterogeneous baselines with pre-training outperform those without, demonstrating enhanced feature extraction capabilities. BIOT does not significantly outperform the heterogeneous approaches and, in some cases, yields inferior results compared to pre-trained heterogeneous decoding methods. This can be attributed to BIOT's primary focus on addressing data heterogeneity in terms of channel count, sampling rate, and data length without adequately tackling the inherent heterogeneity in brain recordings across different subjects. This underscores a key insight: unlike in computer vision and natural language processing, merely aggregating neurological data into a larger dataset does not necessarily enhance performance. In contrast, our proposed UPaNT showcases superior prediction performance due to its pattern-aware feature extraction capabilities. Building on this, H2DiLR, through its ability to disentangle homogeneous and heterogeneous neural representations, significantly enhances prediction performance.

Table A4: Brain anatomical area contribution analysis in tone decoding for both cortical and subcortical issues. The contribution score is normalized to the range between zero and one.

| Brain region | Brain structure | Contribution |
|--------------|-----------------|--------------|
| Thalamus | Subcortical | 0.92 |
| Hippocampus | Subcortical | 0.85 |
| Amygdala | Subcortical | 0.83 |
| Precentral Gyrus | Cortical | 0.97 |
| Insula | Cortical | 0.85 |
| Postcentral Gyrus | Cortical | 0.94 |
| Inferior Frontal Gyrus | Cortical | 0.90 |
| Superior Temporal Gyrus | Cortical | 0.85 |
| Supramarginal Gyrus | Cortical | 0.60 |
| Angular Gyrus | Cortical | 0.66 |

## D ANATOMICAL REGION CONTRIBUTION

We study the contribution of different anatomical brain regions in the task of tone decoding.Due to the dual-stage training paradigm of our H2DiLR framework and the challenges of differentiating through quantization during backpropagation, we cannot directly analyze the contributions of specific regions during neural decoding. To address this, we conducted a separate analysis by combining

the homogeneity-heterogeneity disentanglement (H2D) pre-training and neural decoding stages into a single-stage model. We applied cross-entropy loss with a Transformer decoder to quantized representations, using a straight-through estimator (STE) to approximate gradients. Please note, however, that this setup may introduce training instability and slight performance degradation; it is used solely to analyze region contributions. We calculated channel saliency scores through class activation maps, normalized between 0 and 1, and aggregated them across brain regions linked to speech processing (e.g., Superior Temporal Gyrus, Precentral Gyrus, Postcentral Gyrus), as well as reference regions like the Angular Gyrus. Our findings are summarized in Tab. A4. Our analysis indicates that the Precentral and Postcentral Gyrus are key contributors among cortical regions, with subcortical regions like the Thalamus also playing a significant role in tone decoding.

## E  BROADER IMPACTS

The homogeneity-heterogeneity disentangled learning for neural representations (H2DiLR) presented in this paper aims to advance domains of both fundamental machine learning algorithms and neuroscience.

From the standpoint of algorithmic progress, H2DiLR marks a leap forward in the field of neural representation learning. By addressing the inherent data heterogeneity in neural recordings from multiple subjects, H2DiLR offers a novel approach to disentangling homogeneous and heterogeneous neural representations. This breakthrough has implications beyond the specific application domain of tonal language decoding, as it lays the groundwork for developing more generalized and adaptable neural representation learning algorithms. H2DiLR's ability to capture both homogeneous and heterogeneous neural patterns during representation learning provides researchers with a powerful tool for uncovering underlying neural structures and dynamics that may be obscured by data heterogeneity. Moreover, the framework's capacity to learn generalized neural representations across diverse subjects holds promise for realizing large models similar to LLM to facilitate a wide range of applications, including cognitive neuroscience, clinical diagnosis, and brain-computer interface design.

In terms of practical applications, H2DiLR extends to the field of neural prosthesis, particularly in the context of restoring communication abilities for speech-impaired individuals who speak tonal languages. The ability to decode lexical tones from intracranial recordings using deep learning algorithms has profound implications for speech prosthesis systems tailored to tonal language speakers. By leveraging H2DiLR's unified decoding paradigm, researchers and developers can design more robust and adaptive speech prosthesis systems that accommodate individual variations in neural activity while also facilitating more natural and efficient communication for users.

From the perspective of neuroscience, the proposed H2DiLR has the potential to enhance the interpretability and transferability of neural models, thereby contributing to the understanding of brain function and dynamics. The disentanglement of homogeneous and heterogeneous neural representations facilitated by H2DiLR provides researchers with a clearer insight into the underlying neural processes involved in various cognitive tasks and behaviors. By mapping learned neural codes and electrodes, H2DiLR helps elucidate how different brain regions encode information and interact with each other, shedding light on the complex mechanisms underlying cognitive functions such as language processing and motor control.

One concern with H2DiLR, as with any technology involving intracranial recordings, is the risk of privacy invasion. Decoding thoughts or speech could potentially access highly personal information without proper consent or awareness. If such data were misused or accessed by unauthorized parties, it could lead to significant privacy breaches.

Table A5: The first set of 407 Mandarin Chinese characters used as reading material, with corresponding syllables organized by initial phonetic order. For infrequently used characters, participants were presented directly with the syllable to pronounce.

| Characters | Syllables | Characters | Syllables | Characters | Syllables |
|---|---|---|---|---|---|
| 阿 | ā | 扯 | chě | 吊 | diào |
| 爱 | ài | 趁 | chèn | 叠 | dié |
| 安 | ān | 城 | chéng | 顶 | dǐng |
| 昂 | áng | 痴 | chī | 丢 | diū |
| 袄 | ǎo | 虫 | chóng | 东 | dōng |
| 拔 | bá | 愁 | chóu | 豆 | dòu |
| 白 | bái | 初 | chū | 读 | dú |
| 板 | bǎn | chuā | chuā | 短 | duǎn |
| 帮 | bāng | 揣 | chuāi | 对 | duì |
| 保 | bǎo | 穿 | chuān | 蹲 | dūn |
| 杯 | bēi | 床 | chuáng | 夺 | duó |
| 本 | běn | 吹 | chuī | 鹅 | é |
| 崩 | bēng | 春 | chūn | 恩 | ēn |
| 鼻 | bí | 戳 | chuō | ēng | ēng |
| 编 | biān | 次 | cì | 耳 | ěr |
| 标 | biāo | 葱 | cōng | 罚 | fá |
| 瘪 | biě | 凑 | còu | 反 | fǎn |
| 宾 | bīn | 粗 | cū | 方 | fāng |
| 病 | bìng | 窜 | cuàn | 肥 | féi |
| 伯 | bó | 脆 | cuì | 粉 | fěn |
| 补 | bǔ | 存 | cún | 风 | fēng |
| 擦 | cā | 搓 | cuō | 福 | fú |
| 财 | cái | 打 | dǎ | 否 | fǒu |
| 残 | cán | 带 | dài | 付 | fù |
| 仓 | cāng | 胆 | dǎn | 尬 | gà |
| 槽 | cáo | 党 | dǎng | 盖 | gài |
| 测 | cè | 到 | dào | 敢 | gǎn |
| 参 | cān | 德 | dé | 缸 | gāng |
| 层 | céng | 得 | děi | 告 | gào |
| 茶 | chá | �dèn | dèn | 割 | gē |
| 柴 | chái | 等 | děng | 给 | gěi |
| 产 | chǎn | 底 | dǐ | 根 | gēn |
| 唱 | chàng | 爹 | diē | 耕 | gēng |
| 超 | chāo | 电 | diàn | 共 | gòng |

Table A6: The second set of 407 Mandarin Chinese characters used as reading material, with corresponding syllables organized by initial phonetic order.

| Characters | Syllables | Characters | Syllables | Characters | Syllables |
|---|---|---|---|---|---|
| 狗 | gǒu | 金 | jīn | 冷 | lěng |
| 估 | gū | 镜 | jìng | 力 | lì |
| 瓜 | guā | 窘 | jiǒng | 俩 | liǎ |
| 怪 | guài | 酒 | jiǔ | 连 | lián |
| 关 | guān | 举 | jǔ | 良 | liáng |
| 广 | guǎng | 捐 | juān | 料 | liào |
| 贵 | guì | 决 | jué | 列 | liè |
| 滚 | gǔn | 俊 | jùn | 林 | lín |
| 裹 | guǒ | 卡 | kǎ | 领 | lǐng |
| 哈 | hā | 开 | kāi | 柳 | liǔ |
| 孩 | hái | 砍 | kǎn | 龙 | lóng |
| 寒 | hán | 糠 | kāng | 楼 | lóu |
| 航 | háng | 靠 | kào | 路 | lù |
| 耗 | hào | 科 | kē | 卵 | luǎn |
| 河 | hé | 克 | kè | 轮 | lún |
| 黑 | hēi | 肯 | kěn | 罗 | luó |
| 恨 | hèn | 坑 | kēng | 滤 | lü |
| 恒 | héng | 孔 | kǒng | 略 | lüè |
| 烘 | hōng | 口 | kǒu | 马 | mǎ |
| 吼 | hǒu | 库 | kù | 买 | mǎi |
| 虎 | hǔ | 夸 | kuā | 慢 | màn |
| 画 | huà | 快 | kuài | 忙 | máng |
| 坏 | huài | 款 | kuǎn | 毛 | máo |
| 换 | huàn | 狂 | kuáng | 美 | měi |
| 慌 | huāng | 亏 | kuī | 门 | mén |
| 悔 | huǐ | 捆 | kǔn | 猛 | měng |
| 昏 | hūn | 阔 | kuò | 米 | mǐ |
| 货 | huò | 拉 | lā | 面 | miàn |
| 机 | jī | 来 | lái | 苗 | miáo |
| 价 | jià | 懒 | lǎn | 灭 | miè |
| 尖 | jiān | 浪 | làng | 民 | mín |
| 桨 | jiǎng | 老 | lǎo | 命 | mìng |
| 交 | jiāo | 勒 | lēi | 谬 | miù |
| 姐 | jiě | 雷 | léi | 魔 | mó |

Table A7: The third set of 407 Mandarin Chinese characters used as reading material, with corresponding syllables organized by initial phonetic order. For infrequently used characters, participants were presented directly with the syllable to pronounce.

| Characters | Syllables | Characters | Syllables | Characters | Syllables |
|---|---|---|---|---|---|
| 谋 | móu | 盆 | pén | 乳 | rǔ |
| 木 | mù | 碰 | pèng | 挼 | ruá |
| 拿 | ná | 皮 | pí | 软 | ruǎn |
| 奶 | nǎi | 偏 | piān | 锐 | ruì |
| 男 | nán | 票 | piào | 润 | rùn |
| 囊 | náng | 撇 | piē | 弱 | ruò |
| 闹 | nào | 品 | pǐn | 洒 | sǎ |
| 讷 | nè | 瓶 | píng | 赛 | sài |
| 内 | nèi | 破 | pò | 伞 | sǎn |
| 嫩 | nèn | 剖 | pōu | 嗓 | sǎng |
| 能 | néng | 普 | pǔ | 骚 | sāo |
| 泥 | ní | 骑 | qí | 涩 | sè |
| 年 | nián | 掐 | qiā | 森 | sēn |
| 娘 | niáng | 浅 | qiǎn | 僧 | sēng |
| 尿 | niào | 枪 | qiāng | 沙 | shā |
| 捏 | niē | 桥 | qiáo | 筛 | shāi |
| 您 | nín | 窃 | qiè | 闪 | shǎn |
| 凝 | níng | 琴 | qín | 伤 | shāng |
| 牛 | niú | 情 | qíng | 烧 | shāo |
| 农 | nóng | 穷 | qióng | 赊 | shē |
| 耨 | nòu | 球 | qiú | 谁 | shuí |
| 奴 | nú | 取 | qǔ | 神 | shén |
| 暖 | nuǎn | 劝 | quàn | 绳 | shéng |
| 挪 | nuó | 缺 | quē | 石 | shí |
| 女 | nü | 群 | qún | 手 | shǒu |
| 虐 | nüè | 然 | rán | 书 | shū |
| 哦 | ó | 让 | ràng | 刷 | shuā |
| 藕 | ǒu | 饶 | ráo | 帅 | shuài |
| 爬 | pá | 热 | rè | 栓 | shuān |
| 牌 | pái | 忍 | rěn | 爽 | shuǎng |
| 盘 | pán | 扔 | rēng | 水 | shuǐ |
| 旁 | páng | 日 | rì | 顺 | shùn |
| 抛 | pāo | 容 | róng | 硕 | shuò |
| 赔 | péi | 肉 | ròu | 死 | sǐ |

Table A8: The fourth set of 407 Mandarin Chinese characters used as reading material, with corresponding syllables organized by initial phonetic order.

| Characters | Syllables | Characters | Syllables | Characters | Syllables |
|---|---|---|---|---|---|
| 松 | sōng | 午 | wǔ | 早 | zǎo |
| 搜 | sōu | 洗 | xǐ | 则 | zé |
| 酥 | sū | 霞 | xiá | 贼 | zéi |
| 算 | suàn | 险 | xiǎn | 怎 | zěn |
| 岁 | suì | 向 | xiàng | 增 | zēng |
| 损 | sǔn | 笑 | xiào | 渣 | zhā |
| 锁 | suǒ | 写 | xiě | 债 | zhài |
| 塔 | tǎ | 心 | xīn | 展 | zhǎn |
| 抬 | tái | 形 | xíng | 涨 | zhǎng |
| 谈 | tán | 熊 | xióng | 找 | zhǎo |
| 躺 | tǎng | 修 | xiū | 遮 | zhē |
| 讨 | tǎo | 许 | xǔ | 这 | zhè |
| 特 | tè | 选 | xuǎn | 针 | zhēn |
| 藤 | téng | 学 | xué | 蒸 | zhēng |
| 替 | tì | 寻 | xún | 直 | zhí |
| 田 | tián | 芽 | yá | 肿 | zhǒng |
| 条 | tiáo | 烟 | yān | 州 | zhōu |
| 铁 | tiě | 养 | yǎng | 煮 | zhǔ |
| 停 | tíng | 药 | yào | 抓 | zhuā |
| 桶 | tǒng | 野 | yě | 拽 | zhuāi |
| 偷 | tōu | 衣 | yī | 砖 | zhuān |
| 图 | tú | 银 | yín | 撞 | zhuàng |
| 团 | tuán | 鹰 | yīng | 追 | zhuī |
| 腿 | tuǐ | 哟 | yō | 准 | zhǔn |
| 吞 | tūn | 永 | yǒng | 捉 | zhuō |
| 拖 | tuō | 油 | yóu | 字 | zì |
| 挖 | wā | 雨 | yǔ | 总 | zǒng |
| 外 | wài | 元 | yuán | 走 | zǒu |
| 万 | wàn | 月 | yuè | 组 | zǔ |
| 忘 | wàng | 云 | yún | 钻 | zuān |
| 围 | wéi | 杂 | zá | 醉 | zuì |
| 文 | wén | 栽 | zāi | 尊 | zūn |
| 翁 | wēng | 暂 | zàn | 左 | zuǒ |
| 窝 | wō | 葬 | zàng |  |  |

