# OpenReview forum: "Towards Homogeneous Lexical Tone Decoding from Heterogeneous Intracranial Recordings"
_ICLR.cc/2025/Conference — ICLR 2025 Poster_

### Official Review · Reviewer_VGSo · 2024-10-21

**Soundness:** 3
**Presentation:** 3
**Contribution:** 3
**Rating:** 8
**Confidence:** 3

**Summary:**

In this paper, the authors propose H2DiLR to capture both homogeneous and heterogeneous information from intracranial recordings across multiple subjects for lexical tone decoding. The H2DiLR framework first obtains discrete vectors through a reconstruction task (vq-autoencoding manner), then builds shared codebook and private codebook for the decoding of different subjects. Experiments on a newly acquired, not publicly released sEEG dataset show the H2DiLR method outperforms other baseline methods.

**Strengths:**

1. The motivation of this paper is clear. The authors seek to address a very important problem in brain decoding: dig homogeneity within the brain responses of different subjects, and hope the obtained homogeneity representation can help improve decoding performance. The presentation of this paper is also good.
2. The proposed vq-autoencoding based method outperforms previous models in decoding performance. Additional ablation analysis further confirms the effectiveness of model design.
3. The authors build a new sEEG dataset (promise to release if this paper gets accepted) which has potential to boost researches related to lexical tone decoding.

**Weaknesses:**

1. Since lexical tone decoding is not a very common task in the brain decoding area, I think the task / problem setting needs to be first introduced and defined in the methodology part.
2. The paper lacks analysis of the generalization ability of captured common representation, i.e. if the shared codebook of three subjects well generalize to the fourth subject, and whether any three of the four subjects lead to similar shard codebook. The authors also mention this issue in limitations, but I think the dataset containing four subjects is enough to conduct such experiments.

**Questions:**

1. The usage of non-invasive brain recordings is an emerging topic. What's the performance of H2DiLR in decoding EEG signal?
2. Why the size of codebook and dimension of codebook are set as the same?

**Details Of Ethics Concerns:**

The dataset is related to human subjects, and has not released yet.

---

> ### Author Response · Authors · 2024-11-15
> **Response to Reviewer VGSo (1/2)**
>
> ## Response to Reviewer VGSo (1/2)
> ---
>
> ## Reply to Weakness
>
> > **W1: Since lexical tone decoding is not a very common task in the brain decoding area, I think the task / problem setting needs to be first introduced and defined in the methodology part.**
>
> **R**: We thank Reviewer VGSo for this suggestion. We will add the problem definition of tone decoding in the Method section in both the revised manuscript and the camera-ready version.
>
> > **W2: The paper lacks analysis of the generalization ability of captured common representation. If the shared codebook of three subjects well generalizes to the fourth subject, whether any three of the four subjects lead to a similar shared codebook? The authors also mention this issue in limitations, but I think the dataset containing four subjects is enough to conduct such experiments.**
>
> **R**: As suggested, we conducted an additional ablation study to evaluate the generalization ability of the common representation captured by the public codebook. Specifically, we tested whether the public codebook, learned from three participants, could be generalized to a fourth participant. We followed the same experimental setup described in Sec. 4 with a leave-one-subject-out protocol.
>
> Firstly, we applied the homogeneity-heterogeneity disentanglement (H2D) on three subjects to acquire a public codebook. Then, we pre-trained the fourth subject using this previously acquired public codebook while keeping it frozen and updating only the private codebook. Finally, we performed neural decoding (ND) on the fourth subject. The decoding performance was compared with baselines and variants of our proposed H2DiLR. The results are summarized in the table below:
>
> |     Method     | Pre-training |     Subject 1    |     Subject 2    |     Subject 3    |     Subject 4    |       Avg.       |
> |:--------------:|:------------:|:----------------:|:----------------:|:----------------:|:----------------:|:----------------:|
> |    SPaRCNet    |       -      |  34.94$\pm$2.17  |  36.73$\pm$5.88  |  27.10$\pm$3.22  |  27.10$\pm$1.43  |  31.47$\pm$3.16  |
> |      FFCL      |       -      |  37.47$\pm$2.40  |  37.71$\pm$5.42  |  26.61$\pm$3.18  |  29.47$\pm$1.69  |  32.82$\pm$3.17  |
> |     TS-TCC     |      CL      |  39.59 $\pm$1.68 |  41.47 $\pm$2.78 |  28.82 $\pm$1.75 |  32.33 $\pm$2.17 |  35.55 $\pm$2.09 |
> |      CoST      |      CL      |  43.95 $\pm$1.48 |  40.41 $\pm$4.77 |  31.02 $\pm$1.00 |  34.53 $\pm$2.39 |  37.47 $\pm$2.41 |
> | ST-Transformer |       -      |  39.02 $\pm$2.39 |  37.22 $\pm$4.68 |  27.51 $\pm$2.11 |  29.63 $\pm$2.98 |  33.35 $\pm$3.04 |
> |    NeuroBERT   |      MM      |  42.20 $\pm$1.86 |  43.10 $\pm$2.76 |  29.80 $\pm$1.98 |  32.65 $\pm$2.89 |  36.94 $\pm$2.37 |
> |      BIOT      |      MM      |  42.45 $\pm$6.99 |  40.90 $\pm$5.87 |  33.55 $\pm$2.95 |  33.88 $\pm$1.89 |  37.47 $\pm$2.26 |
> |   H2DiLR ours  |  UPaNT+ H2D  | 49.06 $\pm$ 2.15 | 47.84 $\pm$ 1.81 |  39.18 $\pm$1.68 |  38.61 $\pm$1.49 |  43.67 $\pm$1.78 |
> |   H2DiLR ours  |  UPaNT only  |  45.47 $\pm$3.04 |  44.65 $\pm$1.84 |  35.59 $\pm$2.37 |  35.76$\pm$1.18  |  40.37 $\pm$2.11 |
> |   H2DiLR ours  |   $\nu = 0$  |  43.61 $\pm$2.12 |  42.15 $\pm$1.63 |  34.26 $\pm$1.51 |  35.92 $\pm$1.43 |  38.98 $\pm$1.67 |
> | Generalization ablation |  UPaNT + H2D | 45.98 $\pm$ 2.94 | 42.86 $\pm$ 2.66 | 35.81 $\pm$ 1.56 | 36.47 $\pm$ 2.49 | 40.28 $\pm$ 2.41 |
>
> The row labeled "Generalization Ablation" represents the ablation study, where we froze the public codebook after training it on three subjects and evaluated it on the fourth. We observe that even with the public codebook frozen for three subjects, decoding on the fourth subject still outperforms other baselines. In fact, the performance is better than our H2DiLR model trained with $\nu = 0$ and is comparable to the performance achieved when trained with UPaNT only. This confirms that the representation captured by the public codebook can be effectively applied to a new subject. Moreover, we noted that the performance improvement on Subject 2 was smaller compared to the other subjects. We hypothesize that this could be due to the distinct electrode implantation location for Subject 2, which likely differs from that of the other participants.
>
> ---
> ## Reply to Question
>
> > **Q1: The usage of non-invasive brain recordings is an emerging topic. What's the performance of H2DiLR in decoding EEG signals?**
>
> **R**: In addition to tone decoding using intracranial recordings, we demonstrated the applicability of H2DiLR for seizure prediction using non-invasive EEG signals (Appendix C.1). Beyond neurological signals, our model can be applied to general time-series data with a multi-channel setup, addressing data heterogeneity across different acquisition setups. This broader applicability extends the impact of our work to various domains within the community.

---

> > ### Comment · Reviewer_VGSo · 2024-11-18
> > **Not convincing results**
> >
> > Thanks for the authors' clarification. However, I don't think the generalization ablation experiment is convincing enough. The authors need to prove that training their model on **any three subjects** will lead to improvement of the left subject (sub1, sub2, sub3 ->  sub4; sub1, sub2, sub4 -> sub3; sub1, sub3, sub4 -> sub2; sub2, sub3, sub4 -> sub1), in order to highlight that the model really learns homogeneity.

---

> > > ### Author Response · Authors · 2024-11-18
> > >
> > > **R**: As we mentioned in our earlier response, we used a **leave-one-subject-out** protocol, where the public codebook is trained on three subjects and verified on the fourth subject:
> > >
> > > * With **sub4** left out, we use sub1, sub2, sub3 to train, and test on sub4.
> > > * With **sub3** left out, we use sub1, sub2, sub4 to train, and test on sub3.
> > > * With **sub2** left out, we use sub1, sub3, sub4 to train, and test on sub2.
> > > * With **sub1** left out, we use sub2, sub3, sub4 to train, and test on sub1.
> > >
> > > This is consistent with (sub1, sub2, sub3 -> sub4; sub1, sub2, sub4 -> sub3; sub1, sub3, sub4 -> sub2; sub2, sub3, sub4 -> sub1) suggested by Reviewer VGSo.

---

> > > > ### Author Response · Authors · 2024-11-18
> > > > **Gratitude and Request for Consideration of Updated Score**
> > > >
> > > > We sincerely appreciate the time and effort you have dedicated to reviewing our work and providing such valuable feedback. Your insights have helped us refine our manuscript significantly, and we hope that our detailed responses and additional experiments address your concerns comprehensively.
> > > >
> > > > If you find our efforts and responses satisfactory, we kindly ask you to consider updating your score to reflect the improvements made. Thank you once again for your constructive input and for supporting the advancement of our research!

---

> > > > ### Comment · Reviewer_VGSo · 2024-11-18
> > > > **Detailed results need to show.**
> > > >
> > > > OK. Could you show the detailed results of the four experiments instead of presenting the averaged score?

---

> > > > > ### Author Response · Authors · 2024-11-18
> > > > >
> > > > > As in the table from our previous response, the row labeled "Generalization Ablation" represents the ablation results. The column refers to the subject that was left out. We observe that even with the public codebook frozen for three subjects, decoding on the fourth subject still outperforms other baselines. In fact, the performance is better than our H2DiLR model trained with $\nu=0$, and is comparable to the performance achieved when trained with UPaNT only. This confirms that the representation captured by the public codebook can be effectively applied to a new subject.
> > > > >
> > > > > |     Method     | Pre-training |     Subject 1    |     Subject 2    |     Subject 3    |     Subject 4    |       Avg.       |
> > > > > |:--------------:|:------------:|:----------------:|:----------------:|:----------------:|:----------------:|:----------------:|
> > > > > |    SPaRCNet    |       -      |  34.94$\pm$2.17  |  36.73$\pm$5.88  |  27.10$\pm$3.22  |  27.10$\pm$1.43  |  31.47$\pm$3.16  |
> > > > > |      FFCL      |       -      |  37.47$\pm$2.40  |  37.71$\pm$5.42  |  26.61$\pm$3.18  |  29.47$\pm$1.69  |  32.82$\pm$3.17  |
> > > > > |     TS-TCC     |      CL      |  39.59 $\pm$1.68 |  41.47 $\pm$2.78 |  28.82 $\pm$1.75 |  32.33 $\pm$2.17 |  35.55 $\pm$2.09 |
> > > > > |      CoST      |      CL      |  43.95 $\pm$1.48 |  40.41 $\pm$4.77 |  31.02 $\pm$1.00 |  34.53 $\pm$2.39 |  37.47 $\pm$2.41 |
> > > > > | ST-Transformer |       -      |  39.02 $\pm$2.39 |  37.22 $\pm$4.68 |  27.51 $\pm$2.11 |  29.63 $\pm$2.98 |  33.35 $\pm$3.04 |
> > > > > |    NeuroBERT   |      MM      |  42.20 $\pm$1.86 |  43.10 $\pm$2.76 |  29.80 $\pm$1.98 |  32.65 $\pm$2.89 |  36.94 $\pm$2.37 |
> > > > > |      BIOT      |      MM      |  42.45 $\pm$6.99 |  40.90 $\pm$5.87 |  33.55 $\pm$2.95 |  33.88 $\pm$1.89 |  37.47 $\pm$2.26 |
> > > > > |   H2DiLR ours  |  UPaNT+ H2D  | 49.06 $\pm$ 2.15 | 47.84 $\pm$ 1.81 |  39.18 $\pm$1.68 |  38.61 $\pm$1.49 |  43.67 $\pm$1.78 |
> > > > > |   H2DiLR ours  |  UPaNT only  |  45.47 $\pm$3.04 |  44.65 $\pm$1.84 |  35.59 $\pm$2.37 |  35.76$\pm$1.18  |  40.37 $\pm$2.11 |
> > > > > |   H2DiLR ours  |   $\nu = 0$  |  43.61 $\pm$2.12 |  42.15 $\pm$1.63 |  34.26 $\pm$1.51 |  35.92 $\pm$1.43 |  38.98 $\pm$1.67 |
> > > > > | Generalization ablation |  UPaNT + H2D | 45.98 $\pm$ 2.94 | 42.86 $\pm$ 2.66 | 35.81 $\pm$ 1.56 | 36.47 $\pm$ 2.49 | 40.28 $\pm$ 2.41 |

---

> > > > > > ### Comment · Reviewer_VGSo · 2024-11-18
> > > > > >
> > > > > > Thanks for the authors' reply and additional experiments. I will retain my initial score.

---

> > > > > > > ### Comment · Reviewer_VGSo · 2024-11-19
> > > > > > >
> > > > > > > After reading this paper again, I decide to raise my rating to 8.

---

> > > > > > > > ### Author Response · Authors · 2024-11-19
> > > > > > > >
> > > > > > > > Dear Reviewer VGSo，
> > > > > > > >
> > > > > > > > Thank you for taking the time to revisit our work and for raising your rating to 8. We sincerely appreciate your thoughtful engagement with our paper and your constructive feedback throughout the review process.
> > > > > > > >
> > > > > > > > Authors

---

> ### Author Response · Authors · 2024-11-15
> **Response to Reviewer VGSo (2/2)**
>
> ## Response to Reviewer VGSo (2/2)
> ---
>
> > **Q2:Why the size of the codebook and the dimensions of the codebook set the same?**
>
> **R**: The dimensions and size of the codebooks are flexible parameters, which we set to 256 by default, as this configuration offers an effective balance between reconstruction performance and the expressive capacity of the learned representations. A comprehensive ablation study on various codebook sizes and dimensions, presented in Table 3 of our manuscript, shows that the proposed approach maintains stable performance across different configurations, with the best results achieved using the default setting.

---

### Official Review · Reviewer_AheB · 2024-10-27

**Soundness:** 2
**Presentation:** 2
**Contribution:** 2
**Rating:** 3
**Confidence:** 5

**Summary:**

The authors introduce a neural network architecture, H2DiLR, a self-supervised learning (SSL) method for pre-training models for sEEG tone decoding (4-way classification task) and a novel dataset (407-syllable reading sEEG dataset).

**Dataset**: The patients are asked to read 407 syllables aloud. To make their pronunciation as close to natural speech as possible, carrier words are added to form full sentences. The dataset includes `1221` trials (i.e., 3 trials per syllable). Simultaneously recorded audio signals are then used to precisely identify time stamps for extracting the target syllable from each trial. Each trial is truncated based on these time stamps, resulting in a neural recording of up to `1` second, labeled with one of `4` tones. According to the task paradigm demonstrated in [1], the total recording time for each subject is approximately `5` hours, with each trial lasting around 12 seconds. Therefore, the total amount of pre-training dataset is about `20` hours.

**Model**: They introduce two different codebooks to disentangle the shared and private parts of multiple subjects. During the pre-training stage, they reconstruct the original sEEG signals. After pre-training, they use the frozen pre-trained VQ-Encoder to obtain quantized embeddings of the input sEEG signals. These embeddings are then passed through a temporal Transformer for further temporal integration, followed by a classification head for tone decoding.

**Experiments**: Previous pre-training methods (including BIOT, NeuroBERT, etc.) are compared. Besides, the authors conducted different ablation studies regarding pre-training (partition ratio, # of subjects, etc.).

---------------------------------

**Summary**:

**This study might benefit from further validation to enhance its robustness.**

 - The observed performance improvement might be attributed to the careful selection of hyperparameters in the codebook. See **Private Comment by Reviewer AheB (1/3)** and **Final Comment by Reviewer AheB (1/2)** for more details.
 - Given the current preprocessing approach, the analyzed neuroscience results might be biased. See **Private Comment by Reviewer AheB (2/3)** and **Final Comment by Reviewer AheB (2/2)** for more details.
 - The advantages of the VQ code over the original signal have not been clearly demonstrated through ablation studies. Besides, the author's goal of disentangling different subjects (i.e., the brain's desynchronization nature [3]) does not appear to align with the capabilities of the current model architecture. See **Private Comment by Reviewer AheB (3/3)** and **Final Comment by Reviewer AheB (1/2)** for more details.

sEEG signals differ significantly from EEG signals, and cross-subject decoding remains a challenging task. Achieving notable improvements with such a small amount of subjects is uncommon in prior studies [1,2] related to sEEG-based speech decoding. While the authors' effort to tackle this difficult problem is commendable and encouraging, the effectiveness of the proposed method is ultimately more critical than the story itself. Therefore, I maintain my original **reject** vote. Thanks for the authors' contribution to the field of sEEG-based tone decoding.

**Reference**:

[1] Zheng H, Wang H T, Jiang W B, et al. Du-IN: Discrete units-guided mask modeling for decoding speech from Intracranial Neural signals[J]. arXiv preprint arXiv:2405.11459, 2024.

[2] Chen J, Chen X, Wang R, et al. Subject-Agnostic Transformer-Based Neural Speech Decoding from Surface and Depth Electrode Signals[J]. bioRxiv, 2024.

[3] Buzsaki G. Rhythms of the Brain[M]. Oxford university press, 2006.

**Strengths:**

**Significance**: ~~**Open-source sEEG speech datasets are rare. Their promise of publishing the dataset upon acceptance (Line 780) is good news for the community as it will lower the entry threshold for future research.**~~ Additionally, they demonstrate how SSL-based pre-training allows improving performance compared to only supervised training (**doubtful**).

**Clarity**: The text has a good structure and is well-written. The figures also help in understanding the method.

**Weaknesses:**

**Major**
1. The authors don’t mention the introduction of data augmentation (e.g., temporal jittering, additive noise, etc.) to avoid overfitting, a common strategy used in sEEG-based speech decoding [2,3]. Since the number of available trials (i.e., $\sim$1221 trials) within each subject is too small, I’m not sure whether the improvement of pre-training comes from this.

2. The preprocess details are missing. Could you provide a detailed description of sEEG preprocessing (i.e., from the originally collected sEEG signals at a sample rate of 2kHz to the epoched trials at a sample rate of 1kHz)?

3. In [1], their CNN baseline achieves about 40% on average, and their NAR model achieves 45.75% on average. All these methods are supervised (heterogeneous) baselines, and these results make it hard to assess the contribution of H2DiLR.

4. No code and demo dataset is provided. Without the code and data, it's difficult to verify the claims.

5. Some aspects of the method are undefined or unclear. Please see the Questions section below. These aspects need to be clarified in the manuscript.

**Minor**
1. Figure 1: It could mislead readers into thinking that H2DiLR uses Product Quantization. In Figure 3 and Section 3.3, the authors actually quantize each item in the embedding sequence using one of different codebooks. However, in Figure 1, the “disentangled embedding” suggests that different parts of each individual embedding are quantized with different codebooks.

2. Line 155: typo “LaBraM Anonymous (2024)”

3. Line 841: typo “the learning of 5e-5” should be “the learning rate of 5e-5”? Is the only difference between fine-tuning pre-trained models and training non-pretrained baselines the learning rate? Are all other settings the same?

4. Use `\citet` or `\citep` instead of `\cite` when referencing other works.

5. Since you collected a 407-syllable reading sEEG dataset, how does H2DiLR perform on other types of syllable classification tasks, such as initial syllable classification? Including these additional results might further highlight H2DiLR's effectiveness.

**Reference**

[1] Feng C, Cao L, Wu D, et al. A high-performance brain-sentence communication designed for logosyllabic language[J]. bioRxiv, 2023: 2023.11. 05.562313.

[2] Metzger S L, Littlejohn K T, Silva A B, et al. A high-performance neuroprosthesis for speech decoding and avatar control[J]. Nature, 2023, 620(7976): 1037-1046.

[3] Willett F R, Kunz E M, Fan C, et al. A high-performance speech neuroprosthesis[J]. Nature, 2023, 620(7976): 1031-1036.

**Questions:**

1. Table 1: What is the codebook size of Heterogeneous H2DiLR with ($\nu=0$)? Is it `K=32`?

2. Did you use a different signal length for pre-training and downstream classification? Are there any differences between the pre-training dataset and the downstream classification dataset?

3. Line 879: Did you apply normalization to each channel independently (i.e., calculating the mean and standard deviation for each channel separately) or across all channels together?

---

> ### Author Response · Authors · 2024-11-17
> **Response to Reviewer AheB (1/2)**
>
> ## Response to Reviewer AheB (1/2)
> ---
>
> ## Reply to Major Weakness
>
> > **W1: The authors don’t mention the usage of data augmentation (e.g., temporal jittering, etc.) to avoid overfitting in sEEG-based speech decoding [2,3]. Since the number of available trials within each subject is too small, I’m not sure whether the improvement in pre-training comes from this.**
>
> **R**: As mentioned in the implementation details in Appendix B, we did not apply these data augmentations in the pre-training and fine-tuning stages. The effect of these augmentations can vary significantly depending on the hyperparameters, decoding tasks, and individual subjects [4]. In our experiments, we observed no consistent improvements across subjects when using the same set of augmentation hyperparameters, which is why we chose not to incorporate them in this work.
>
> > **W2: The preprocess details are missing. Could you provide a detailed description of sEEG preprocessing (i.e., from the originally collected sEEG signals at a sample rate of 2kHz to the epoched trials at a sample rate of 1kHz)?**
>
> **R**: The neural signals were low-pass filtered at 300 Hz and notch filtered at 50 Hz, 100 Hz, 150 Hz, and 200 Hz to remove power line interference. The signals were then downsampled to 1000 Hz. It has been updated in the revised manuscript.
>
> > **W3: The CNN baseline and the NAR model achieve about 40% and 45.75% on average in [1]. All these methods are supervised (heterogeneous) baselines, and these results make it hard to assess the contribution of H2DiLR.**
>
> **R**: As you mentioned, Feng et al. [1] utilized a **multi-modal** approach to combine both sEEG and audio data (as the multi-modality setting), whereas our proposed H2DiLR uses **only sEEG** data, making it a more generalizable framework. Building upon this difference, as reported in Table 4 in our manuscript, H2DiLR still achieves a **higher accuracy (45.95%)** compared to the results in [1], even with a slightly larger network structure and using sEEG data only.
>
> > **W4: No code and demo dataset is provided. Without the code and data, it isn't easy to verify the claims.**
>
> **R**: We plan to release the code upon acceptance of this paper. As for the data, it is available from the corresponding authors upon reasonable request, ensuring compliance with ethical and privacy guidelines.
>
> > **W5: Some aspects of the method are undefined or unclear. Please see the Questions section below. These aspects need to be clarified in the manuscript.**
>
> **R**: We have addressed the specific points you raised in the Reply to Question section. Please refer to our detailed responses for clarification, and we have ensured that the manuscript has been updated accordingly for improved clarity.
>
> ## References
>
> [1] Feng C, Cao L, Wu D, et al. A high-performance brain-sentence communication designed for logosyllabic language[J]. bioRxiv, 2023: 2023.11. 05.562313.
>
> [2] Metzger S L, Littlejohn K T, Silva A B, et al. A high-performance neuroprosthesis for speech decoding and avatar control[J]. Nature, 2023, 620(7976): 1037-1046.
>
> [3] Willett F R, Kunz E M, Fan C, et al. A high-performance speech neuroprosthesis[J]. Nature, 2023, 620(7976): 1031-1036.
>
> [4] Wu, Di, et al. "Neuro-BERT: Rethinking Masked Autoencoding for Self-Supervised Neurological Pretraining." IEEE Journal of Biomedical and Health Informatics (2024).
>
>
> ---
> ## Reply to Minor Weakness
>
> > **W1: Figure 1 might mislead readers into thinking that H2DiLR uses Product Quantization. In Figure 3 and Section 3.3, the authors actually quantize each item in the embedding sequence using one of the different codebooks. However, “disentangled embedding” in Figure 1 suggests that different parts of each individual embedding are quantized with different codebooks.**
>
> **R**: As described in Sec. 3.3, we maintain a public codebook shared across all subjects and a subject-specific private codebook for each subject. Each neural embedding from an encoder is quantized using either the shared codebook or the subject-specific private codebook, based on a similarity ranking outlined in **Equation 4**. Thus, different parts of an individual embedding are indeed quantized by one of these two codebooks (public or subject-specific), which aligns with the depiction in Fig. 1. We believe this is consistent.
>
> > **W2: Line 155: typo “LaBraM Anonymous (2024)”**
>
> **R**: It has been corrected in the revised manuscript.
>
> > **W3: Line 841: typo “the learning of 5e-5” should be “the learning rate of 5e-5”? Is the only difference between fine-tuning pre-trained models and the non-pretrained baselines the learning rate? Are all other settings the same?**
>
> **R**: All other training settings during the ND stage are identical to those used for the non-pretrained baselines, except for the learning rate, which has been clarified in the revised manuscript. It has been corrected in the revised manuscript for clarity.

---

> > ### Author Response · Authors · 2024-11-17
> > **Response to Reviewer AheB (2/2)**
> >
> > ## Response to Reviewer AheB (2/2)
> > ---
> >
> > > **W4: Use \citet or \citep instead of \cite when referencing other works.**
> >
> > **R**:  We have already used \citet and \citep appropriately in the submitted manuscript. If there are any specific instances where this was not the case, please let us know, and we will address them promptly.
> >
> > > **W5: Since you collected a 407-syllable reading sEEG dataset, how does H2DiLR perform on other types of syllable classification tasks? Including these additional results might further highlight H2DiLR's effectiveness.**
> >
> > **R**: Thanks for your recommendation. We further conducted experiments on initial decoding, modifying the classification task from 4-class (tone decoding) to 24-class classification (one class corresponding to no initial in the syllable). All experimental settings remained the same, as described in Section 4. The results are summarized in the table below:
> >
> > | Method | Pre-training | Subject 1 | Subject 2 | Subject 3 | Subject 4 | Avg. |
> > |---|:---:|:---:|:---:|:---:|:---:|:---:|
> > | SPaRCNet | - | 28.61$\pm$2.03 | 18.67$\pm$1.12 | 10.49$\pm$5.94 | 7.87$\pm$3.60 | 16.41$\pm$3.17 |
> > | FFCL | - | 31.55$\pm$3.56 | 20.16$\pm$4.65 | 11.66$\pm$4.21 | 8.36$\pm$1.69 | 17.93$\pm$3.52 |
> > | TS-TCC | CL | 32.87 $\pm$4.42 | 22.63 $\pm$5.49 | 11.56 $\pm$3.79 | 9.10 $\pm$4.48 | 19.04 $\pm$4.54 |
> > | CoST | CL | 33.19 $\pm$2.94 | 23.93 $\pm$3.82 | 11.25 $\pm$4.95 | 10.41 $\pm$5.62 | 19.69 $\pm$4.33 |
> > | ST-Transformer | - | 32.38 $\pm$3.97 | 22.79 $\pm$3.01 | 12.13 $\pm$5.21 | 10.47 $\pm$4.49 | 19.44 $\pm$4.17 |
> > | NeuroBERT | MM | 34.59 $\pm$2.25 | 25.49 $\pm$2.81 | 13.03 $\pm$2.46 | 12.87 $\pm$1.93 | 21.48 $\pm$2.36 |
> > | **H2DiLR ours** | $\nu = 0$ | 35.42 $\pm$2.96 | 25.25 $\pm$1.77 | 11.97 $\pm$2.97 | 9.99 $\pm$2.68 | 20.65 $\pm$2.59 |
> > | BIOT | MM | 35.08 $\pm$4.47 | 24.66 $\pm$4.63 | 12.38 $\pm$3.47 | 10.82 $\pm$4.66 | 20.73 $\pm$4.30 |
> > | **H2DiLR ours** | **UPaNT only** | 37.37 $\pm$ 2.99 | 27.21 $\pm$2.71 | 16.01 $\pm$2.47 | 14.34 $\pm$2.15 | 23.73 $\pm$2.58 |
> > | **H2DiLR ours** | **UPaNT+H2D** | **39.51** $\pm$**2.43** | **29.75** $\pm$ **2.79** | **19.67** $\pm$ **2.12** | **16.31** $\pm$ **1.23** | **26.31** $\pm$ **2.14** |
> >
> > The results demonstrate that H2DiLR (UPaNT+H2D) achieves the best performance across all subjects, significantly outperforming other baselines and its own variants.
> >
> > Additionally, beyond tone and initial decoding tasks using intracranial recordings, we highlight H2DiLR's broader applicability on the task of **seizure prediction** using non-invasive EEG signals, **as detailed in Appendix C.1**.
> >
> > ---
> > ## Reply to Question
> >
> > > **Q1: What is the codebook size of Heterogeneous H2DiLR with ($\nu$=0) in Table 1?**
> >
> > **R**: Yes, unless otherwise stated, the private codebook size for each subject is set to 32, while the public codebook size is set to 128.
> >
> > > **Q2: Did you use a different signal length for pre-training and downstream classification? Are there any differences between the pre-training dataset and the downstream classification dataset?**
> >
> > **R**: We use the same dataset for both the H2D stage (pre-training) and the ND stage (fine-tuning). As described in Sec. 3, the raw sEEG signals are quantized into neural representations during the H2D stage, capturing both homogeneous and heterogeneous features in an unsupervised manner. In the ND stage, the decoder utilizes these quantized neural representations as input to perform the decoding tasks.
> >
> > **Q3: In Line 879, did you apply normalization to each channel independently or across all channels together?**
> >
> > **R**: Yes, the normalization is applied independently to each channel, where the mean and standard deviation are computed separately for each channel. We will clarify these in the revised manuscript.

---

> ### Comment · Reviewer_AheB · 2024-11-17
> **Official Comment by Reviewer AheB**
>
> Thank you for your detailed rebuttal and answering all my points. Nevertheless, I still have some concerns:
>
> **Major**
>
> **W1**: OK. Perhaps the reason why “temporal jitter” doesn’t work is that you utilize the identified time stamps to epoch the data, which makes the feature wave time-locked to the onset. “additive noise” also doesn’t work,,, not properly preprocessed? Please see **W2**.
>
> **W2**: So, no referencing preprocessing [1] is applied to the raw sEEG recordings? It’s important to note that the raw sEEG recordings are highly correlated (especially on the same electrode). Typically, this part needs to be removed by either laplacian reference [2] or bipolar reference [3,4] to obtain relatively cleaner signals that focus more on the activity of local neuronal groups. If the model directly reconstructs the sEEG recordings without referencing, the model will be biased towards the background noise of the brain (as the electrical signals spread out across the brain, like EEG). Could the authors explain this?
>
> **W3&W4**: OK. Sharing the data and code would greatly enhance the validation and reproducibility of the proposed method by the research community. Please see Minor **W5**.
>
> **Minor**:
>
> **W1**: OK. Different parts of **embedding sequence** are indeed quantized by one of these two codebooks (public or subject-specific).
>
> **W2&W3&W4**: OK.
>
> **W5**: Thank you for providing the additional results. They are helpful, although the performance remains comparable to the (uni-modality) AAI model (26.78\%) reported in [5]. I saw the results of seizure prediction, which is a channel-level classification task. I want to understand the effectiveness of H2DiLR on decoding cognitive states (e.g., speech decoding). Based on my understanding (which could be wrong), tune decoding is a subset of speech decoding, mainly involving speech motor brain regions (e.g., vSMC, STG). To my knowledge, Du-IN doesn’t address the issue of multi-subject modeling. Since the code and dataset for Du-IN [3] are publicly available, could H2DiLR outperform Du-IN? (optional, if there is enough time).
>
> **Question**:
>
> **Q1**: OK. **`K=32` is smaller than that used in Du-IN [3]. Considering **Major W2**, I’m not sure which part of the raw sEEG recordings the VQ-VAE focuses on. Could the authors explain this?**
>
> **Q2&Q3**: OK.
>
> **Summary**:
>
> The authors should provide more results to demonstrate the effectiveness of H2DiLR:
>  - The results after referencing preprocessing (e.g., bipolar re-reference [3,4]).
>  - If there is enough time, the results on the Du-IN dataset should be provided.
>
> **Reference**:
>
> [1] Li G, Jiang S, Paraskevopoulou S E, et al. Optimal referencing for stereo-electroencephalographic (SEEG) recordings[J]. NeuroImage, 2018, 183: 327-335.
>
> [2] Wang C, Subramaniam V, Yaari A U, et al. BrainBERT: Self-supervised representation learning for intracranial recordings[J]. arXiv preprint arXiv:2302.14367, 2023.
>
> [3] Zheng H, Wang H T, Jiang W B, et al. Du-IN: Discrete units-guided mask modeling for decoding speech from Intracranial Neural signals[J]. arXiv preprint arXiv:2405.11459, 2024.
>
> [4] Mentzelopoulos G, Chatzipantazis E, Ramayya A G, et al. Neural decoding from stereotactic EEG: accounting for electrode variability across subjects[C]//The Thirty-eighth Annual Conference on Neural Information Processing Systems.
>
> [5] Feng C, Cao L, Wu D, et al. A high-performance brain-sentence communication designed for logosyllabic language[J]. bioRxiv, 2023: 2023.11. 05.562313.

---

> > ### Author Response · Authors · 2024-11-17
> >
> > ## Response to Reviewer AheB (1/2)
> > ---
> >
> > ## Reply to Major Weakness
> >
> > >**W1: Perhaps the reason why “temporal jitter” doesn’t work is that you utilize the identified time stamps to epoch the data, which makes the feature wave time-locked to the onset. “additive noise” also doesn’t work... not properly preprocessed? Please see W2.**
> >
> > **R**: As mentioned in our earlier response, we did not observe consistent improvements **across subjects** when applying the same augmentation hyperparameters, including for "additive noise." Previous works, such as [6], used heterogeneous models, where augmentations were likely fine-tuned at the subject level. In contrast, our H2D stage processes raw data from all subjects simultaneously, which complicates hyperparameter optimization for augmentations.
> >
> > Given the complexity of identifying universal augmentation settings suitable for all subjects, we chose not to incorporate data augmentations in our experiments.
> >
> > > **W2: So, no referencing preprocessing [1] is applied to the raw sEEG recordings? It’s important to note that the raw sEEG recordings are highly correlated (especially on the same electrode). Typically, this part needs to be removed by either laplacian reference [2] or bipolar reference [3,4] to obtain relatively cleaner signals that focus more on the activity of local neuronal groups. If the model directly reconstructs the sEEG recordings without referencing, the model will be biased towards the background noise of the brain (as the electrical signals spread out across the brain, like EEG). Could the authors explain this?**
> >
> > **R**: The head stage front-end we utilized for collecting sEEG signals employs bipolar differential amplifiers with a common reference electrode. This design inherently reduces common-mode noise at the hardware level, ensuring that the recorded signals are already relatively clean. Additionally, the referenced channels were carefully selected by neurological experts at the hospital to optimize signal quality. Consequently, no further referencing operations, such as laplacian or bipolar referencing, were applied to the collected data.

---

> > > ### Comment · Reviewer_AheB · 2024-11-17
> > > **Official Comment by Reviewer AheB**
> > >
> > > > The head stage front-end we utilized for collecting sEEG signals employs **bipolar differential amplifiers with a common reference electrode**.
> > >
> > > What does "bipolar differential amplifiers with a common reference electrode" refer to? You randomly select one intracranial sEEG channel from all sEEG channels to serve as the reference channel? Then, the relative amplitudes of other sEEG channels are measured according to that sEEG reference channel?

---

> > > > ### Author Response · Authors · 2024-11-18
> > > >
> > > > **R**: All intracranial sEEG channels are bipolar-referenced to a single reference channel, as configured in the hardware. This reference channel is not randomly selected; it is determined by neurological experts at the hospital. The relative amplitudes of other sEEG channels are then measured with respect to this reference channel to effectively reduce common-mode noise.

---

> ### Author Response · Authors · 2024-11-17
> **Response to Reviewer AheB  (2/2)**
>
> ## Response to Reviewer AheB (2/2)
> ---
> ## Reply to Minor Weakness
>
> >**W5: Thank you for providing the additional results. They are helpful, although the performance remains comparable to the (uni-modality) AAI model (26.78%) reported in [5]. I saw the results of seizure prediction, which is a channel-level classification task. I want to understand the effectiveness of H2DiLR on decoding cognitive states (e.g., speech decoding). Based on my understanding (which could be wrong), tune decoding is a subset of speech decoding, mainly involving speech motor brain regions (e.g., vSMC, STG). To my knowledge, Du-IN doesn’t address the issue of multi-subject modeling. Since the code and dataset for Du-IN [3] are publicly available, could H2DiLR outperform Du-IN? (optional, if there is enough time).**
>
> **R**: It’s important to note that speech prostheses and current speech decoding systems [6-7] often focus on decoding motor movements associated with speech production, such as lip or tongue movement during **phoneme articulation**. Our work targets a specialized subset of articulation decoding—tone decoding—which specifically involves decoding articulatory properties rather than semantic content. In contrast, Du-IN addresses **word-level decoding**, which focuses possibly more on semantics rather than articulatory. **This distinction introduces variability in the neural states being decoded and necessitates different approaches for channel selection.**
>
> Specifically, Du-IN employs a **data-driven** channel selection strategy, choosing the channels that contribute most to the decoding task during training. This approach inherently biases the system toward the specific decoding algorithm/network architecture used, as evidenced by the significant performance differences between Du-IN (~ 60%) and other methods like Brant [8] (~ 10%) and LaBraM [9] (~10%). Authors of Du-IN highlight the substantial contributions of **white matter** in their decoding tasks, especially for Subjects 2 and 3, as shown in Tab. 17. Conversely, our work focuses on decoding speech-related neural signals, explicitly excluding white matter channels to align with our tone decoding task.
>
> While Du-IN’s dataset is publicly available, its accompanying channel localization details (beyond the ten selected channels) are not fully provided. Since channel selection rules critically influence results, directly comparing our method to Du-IN without access to equivalent channel configurations would not result in a fair or unbiased evaluation.
>
> That said, we’ve already demonstrated the generalizability and effectiveness of H2DiLR on various tasks, including tone decoding, initial syllable decoding, and seizure prediction. This versatility underscores its potential as a general-purpose framework for neural decoding. While it would be ideal to evaluate H2DiLR on more tasks like word-level decoding, resource and time constraints prevent us from exhaustively addressing all potential decoding scenarios within the scope of this work.
>
> Nevertheless, we acknowledge the importance of exploring the DU-IN dataset for a more comprehensive comparison and will make our best effort to analyze it within the constraints of time and available information. We appreciate your suggestion and will aim to provide insights, if feasible, into how H2DiLR performs on this dataset.
>
> ---
> ## Reply to Question
>
> >**Q1: K=32 is smaller than that used in Du-IN [3]. Considering Major W2, I’m not sure which part of the raw sEEG recordings the VQ-VAE focuses on. Could the authors explain this?**
>
> **R**: Our H2DiLR framework employs a public codebook size of 128 and a private codebook size of 32 for each subject. With four subjects, the total number of codes utilized in the H2D stage amounts to 256.
>
> The variant of H2DiLR with $\nu=0$ represents a special case that simplifies the framework to focus solely on personalized representation, allowing for broader comparison with other methods.
>
> As outlined in Equation 4, each neural embedding derived from the raw sEEG signal is quantized using either the public codebook or the subject-specific private codebook. The selection process is guided by a similarity ranking mechanism, enabling the model to effectively capture both homogeneous and heterogeneous features within the data.

---

> > ### Author Response · Authors · 2024-11-17
> >
> > # References
> >
> > [1] Li G, Jiang S, Paraskevopoulou S E, et al. Optimal referencing for stereo-electroencephalographic (SEEG) recordings[J]. NeuroImage, 2018, 183: 327-335.
> >
> > [2] Wang C, Subramaniam V, Yaari A U, et al. BrainBERT: Self-supervised representation learning for intracranial recordings[J]. arXiv preprint arXiv:2302.14367, 2023.
> >
> > [3] Zheng H, Wang H T, Jiang W B, et al. Du-IN: Discrete units-guided mask modeling for decoding speech from Intracranial Neural signals[J]. arXiv preprint arXiv:2405.11459, 2024.
> >
> > [4] Mentzelopoulos G, Chatzipantazis E, Ramayya A G, et al. Neural decoding from stereotactic EEG: accounting for electrode variability across subjects[C]//The Thirty-eighth Annual Conference on Neural Information Processing Systems.
> >
> > [5] Feng C, Cao L, Wu D, et al. A high-performance brain-sentence communication designed for logosyllabic language[J]. bioRxiv, 2023: 2023.11. 05.562313.
> >
> > [6] Metzger S L, Littlejohn K T, Silva A B, et al. A high-performance neuroprosthesis for speech decoding and avatar control[J]. Nature, 2023, 620(7976): 1037-1046.
> >
> > [7] Willett F R, Kunz E M, Fan C, et al. A high-performance speech neuroprosthesis[J]. Nature, 2023, 620(7976): 1031-1036.
> >
> > [8] Zhang, Daoze, et al. "Brant: Foundation model for intracranial neural signal." Advances in Neural Information Processing Systems 36 (2024).
> >
> > [9] Jiang, Wei-Bang, Li-Ming Zhao, and Bao-Liang Lu. "Large brain model for learning generic representations with tremendous EEG data in BCI." arXiv preprint arXiv:2405.18765 (2024).

---

> > > ### Author Response · Authors · 2024-11-17
> > > **Gratitude and Request for Consideration of Updated Score**
> > >
> > > We sincerely appreciate the time and effort you have dedicated to reviewing our work and providing such valuable feedback. Your insights have helped us refine our manuscript significantly, and we hope that our detailed responses and additional experiments address your concerns comprehensively.
> > >
> > > If you find our efforts and responses satisfactory, we kindly ask you to consider updating your score to reflect the improvements made. Thank you once again for your constructive input and for supporting the advancement of our research!

---

> ### Comment · Reviewer_AheB · 2024-11-18
> **Official Comment by Reviewer AheB**
>
> **Major**
>
> **W2**
>
> OK. This operation is the standard configuration of sEEG recordings. So, the aforementioned concerns still exist -- **no laplacian [2] or bipolar [3,4] re-reference is applied to remove the highly correlated parts across channels (e.g., the noise within certain brain regions)**. Please refer to [1] for the detailed definition of bipolar re-reference. For example, if 15 channels are located on one electrode (e.g., A1, ..., A15), bipolar re-reference will calculate the differences between nearby neighbors (e.g., A1-A2, A2-A3, ..., A14-A15). Du-IN [3] reconstructs `10` channels within certain brain regions with `codex_size=2048`, which is greatly larger than `K=32` used for the whole brain recordings reconstruction (e.g., `>50` channels). Are you sure the bandwidth of invasive sEEG recordings is that low? -- **with 32 discrete codes and a conv-deconv autoencoder, you can reconstruct the signals of the whole brain**, overlooking the brain's desynchronization nature [5].
>
> **Minor**
>
> **W5**
>
> This is just optional. However, their codes have already provided the list of `10` channels for each subject to reproduce the results (with the corresponding checkpoints). Compared to Feng et. al. [6], **H2DiLR (26.31\%) performs similarly to the supervised (heterogeneous) AAI model (uni-modality, 26.78\%)**. On tune decoding, they also perform similarly (45.95\%(H2DiLR)v.s.[40\%(CNN),45.75\%(NAR)]), although the authors clarify that 45.75\% is obtained by a (multi-modal) NAR model. All these results make it hard to assess the effectiveness of H2DiLR.
>
> In Du-IN, `Du-IN (mae)` corresponds to `H2DiLR ($\nu=0$)`, `Du-IN (poms)` corresponds to `H2DiLR (UPaNT only)`. When they built the cross-subject model, they encountered the problem of representation conflict. Their `Du-IN (mae)` (i.e., `H2DiLR ($\nu=0$)`) achieves higher performance compared to `Du-IN (poms)` (i.e., `H2DiLR (UPaNT only)`). VQ-VAE has the power to align different subjects space-wise (instead of element-wise). How could `H2DiLR ($\nu=0$)` perform worse than `H2DiLR (UPaNT only)`?
>
> **Additional**
>
> > Our analysis indicates that the Precentral and Postcentral Gyri are key contributors among cortical regions, with **subcortical regions like the Thalamus also playing a significant role in tone decoding**.
>
> Besides, I still have an additional concern about the neuroscience ablation results. As you mentioned,
>
> > Our work targets a specialized subset of articulation decoding—tone decoding—which specifically involves decoding **articulatory properties** rather than semantic content. In contrast, Du-IN addresses word-level decoding, which focuses possibly more on semantics rather than articulatory.
>
> Considering **Major W2**, could the authors provide some neuroscience research to support this result?
>
> **Summary**:
> My major concern mainly focuses on
> - whether the data has been properly preprocessed, this is crucial for the downstream analysis and decoding performance.
> - the effectiveness of cross-subject performance.
>
> **Reference**:
>
> [1] Li G, Jiang S, Paraskevopoulou S E, et al. Optimal referencing for stereo-electroencephalographic (SEEG) recordings[J]. NeuroImage, 2018, 183: 327-335.
>
> [2] Wang C, Subramaniam V, Yaari A U, et al. BrainBERT: Self-supervised representation learning for intracranial recordings[J]. arXiv preprint arXiv:2302.14367, 2023.
>
> [3] Zheng H, Wang H T, Jiang W B, et al. Du-IN: Discrete units-guided mask modeling for decoding speech from Intracranial Neural signals[J]. arXiv preprint arXiv:2405.11459, 2024.
>
> [4] Mentzelopoulos G, Chatzipantazis E, Ramayya A G, et al. Neural decoding from stereotactic EEG: accounting for electrode variability across subjects[C]//The Thirty-eighth Annual Conference on Neural Information Processing Systems.
>
> [5] Buzsaki G. Rhythms of the Brain[M]. Oxford university press, 2006.
>
> [6] Feng C, Cao L, Wu D, et al. A high-performance brain-sentence communication designed for logosyllabic language[J]. bioRxiv, 2023: 2023.11. 05.562313.

---

> ### Author Response · Authors · 2024-11-18
>
> ## Further Response to Reviewer AheB
> ---
>
> ## Reply to Major Weakness
>
> >**W2: This operation is the standard configuration of sEEG recordings. So, the aforementioned concerns still exist -- no laplacian [2] or bipolar [3,4] re-reference is applied to remove the highly correlated parts across channels (e.g., the noise within certain brain regions). Please refer to [1] for the detailed definition of bipolar re-reference. For example, if 15 channels are located on one electrode (e.g., A1, ..., A15), bipolar re-reference will calculate the differences between nearby neighbors (e.g., A1-A2, A2-A3, ..., A14-A15). Du-IN [3] reconstructs 10 channels within certain brain regions with codex_size=2048, which is greatly larger than K=32 used for the whole brain recordings reconstruction (e.g., >50 channels). Are you sure the bandwidth of invasive sEEG recordings is that low? -- with 32 discrete codes and a conv-deconv autoencoder, you can reconstruct the signals of the whole brain, overlooking the brain's desynchronization nature [5].**
>
> **R**: In our work, the primary focus is on **extracting task-relevant decoding features** rather than explicitly **removing noise within certain brain regions**. It is worth emphasizing that there is no universally accepted definition of "noise" in functional brain regions, nor is there a **gold standard** for referencing in the domain of EEG or sEEG measurements. While Laplacian or bipolar referencing is a common practice, it is not mandatory. Many recent and impactful studies have employed a **common referencing** strategy similar to ours. For instance:
>
> * Huang et al. (2023) in **Nature Communications** [7]: "We re-referenced the signal from each channel/contact to the average signal across all the white matter channels."
> * Regev et al. (2024) in **Nature Human Behaviour** [8]: "The common average reference (from all electrodes connected to the same amplifier) was removed for each time point separately."
> * Fan et al. (2024) in **Nature Mental Health** [9]: "The continuous intracranial EEG data in each electrode contact was notch-filtered at 60 Hz, re-referenced to the common average reference, and segmented from -500 ms to 1,000 ms relative to stimulus onset."
>
> These examples highlight that the referencing strategy we used is well-established and accepted in the field.
>
> Regarding the codebook size, we would like to clarify that **codebook usage efficiency** is equally critical as the codebook size itself. Large codebooks often suffer from low utilization rates, meaning many codes remain unused during training. As we stated earlier, our approach utilizes a total of 256 codes, with 128 in the public codebook and 32 per subject-specific private codebook. Our experiments confirmed a high code usage rate, which supports effective learning and reconstruction. As shown in Figure 6 and Table 3 of our paper, our approach achieves strong reconstruction performance both **quantitatively and qualitatively**, validating our chosen codebook configuration.
>
> Furthermore, the reconstruction task serves as a pretext task in VQ-VAE training to enable effective representation learning. While high reconstruction accuracy is important, the ultimate goal is to ensure the learned representations are generalizable and task-relevant. We hope this clarification demonstrates the rationale behind our referencing strategy and codebook design, which balance the needs of task-relevant feature extraction and efficient model training.
>
> ---
> ## Reply to Minor Weakness
>
> >**W5-1: This is just optional. However, their codes have already provided the list of 10 channels for each subject to reproduce the results (with the corresponding checkpoints). Compared to Feng et. al. [6], H2DiLR (26.31%) performs similarly to the supervised (heterogeneous) AAI model (uni-modality, 26.78%). On tune decoding, they also perform similarly (45.95%(H2DiLR)v.s.[40%(CNN),45.75%(NAR)]), although the authors clarify that 45.75% is obtained by a (multi-modal) NAR model. All these results make it hard to assess the effectiveness of H2DiLR.**
>
> **R**: NAR is a **multi-modal** method that integrates both **audio and sEEG data**. In contrast, our proposed H2DiLR achieves a performance of 45.95% using only sEEG data, surpassing NAR's 45.75%, which benefits from additional audio information. This demonstrates H2DiLR's capability to extract meaningful representations from sEEG data alone, without relying on external modalities.
>
> Similarly, the AAI model leverages **initial category features** as additional supervision, introducing **external guidance** beyond the sEEG signal. Despite this, our approach demonstrates comparable performance on decoding tasks when restricted to using sEEG data alone. This comparison underscores H2DiLR's effectiveness as a standalone decoding framework, even when directly compared to methods utilizing auxiliary inputs or supervision.

---

> > ### Author Response · Authors · 2024-11-18
> >
> > >**W5-2: In Du-IN, Du-IN (mae) corresponds to H2DiLR ($\nu=0$), Du-IN (poms) corresponds to H2DiLR (UPaNT only). When they built the cross-subject model, they encountered the problem of representation conflict. Their Du-IN (mae) (i.e., H2DiLR ($\nu=0$)) achieves higher performance compared to Du-IN (poms) (i.e., H2DiLR (UPaNT only)). VQ-VAE has the power to align different subjects space-wise (instead of element-wise). How could H2DiLR ($\nu=0$) perform worse than H2DiLR (UPaNT only)?**
> >
> > **R**: We cannot fully comment on why DU-IN variants perform the way they do, as we have not worked with their dataset or specific fine-tuning processes. However, it is possible that certain factors unique to the DU-IN implementation could influence their relative performance. Regarding the differences in performance between H2DiLR (UPaNT only) and H2DiLR ($\nu=0$), we believe the following factors might contribute:
> >
> > * (1) Compared to the conventional homogeneous training approach, where data from different subjects are directly merged, the VQ mechanism introduces a "disentangled lookup table". This design allows for partial codes that are potentially more favorable to specific subjects. In our work, using four subjects did not result in severe code collisions within the shared codebook. However, as the number of subjects increases, the likelihood of such collisions may grow. This is precisely why we adopt a dual-codebook strategy: a shared public codebook to capture cross-subject commonalities and a private codebook tailored for each subject.
> >
> > * (2) H2DiLR (UPaNT only) employs a larger codebook size (K=128) compared to H2DiLR ($\nu=0$), which uses K=32. A larger codebook provides more capacity for learning diverse representations, which could explain why H2DiLR (UPaNT only) achieves better performance.
> >
> > ---
> > ## Reply to additional Question
> >
> > >**Q: Considering Major W2, could the authors provide some neuroscience research to support this result?**
> >
> > **R**: As mentioned in our responses to other reviewers, we calculated channel saliency scores using class activation maps and reported the contribution scores averaged within specific brain regions. Our analysis highlights that the Precentral and Postcentral Gyrus are key contributors among cortical regions, while subcortical areas such as the Thalamus also play a significant role in tone decoding. These findings align with current neuroscience research on speech-related brain activity.
> >
> > If Reviewer AheB is referring to specific additional neuroscience studies or insights beyond what has been cited and discussed, we would appreciate clarification to better address the request.
> >
> > # References
> >
> > [1] Li G, Jiang S, Paraskevopoulou S E, et al. Optimal referencing for stereo-electroencephalographic (SEEG) recordings[J]. NeuroImage, 2018, 183: 327-335.
> >
> > [2] Wang C, Subramaniam V, Yaari A U, et al. BrainBERT: Self-supervised representation learning for intracranial recordings[J]. arXiv preprint arXiv:2302.14367, 2023.
> >
> > [3] Zheng H, Wang H T, Jiang W B, et al. Du-IN: Discrete units-guided mask modeling for decoding speech from Intracranial Neural signals[J]. arXiv preprint arXiv:2405.11459, 2024.
> >
> > [4] Mentzelopoulos G, Chatzipantazis E, Ramayya A G, et al. Neural decoding from stereotactic EEG: accounting for electrode variability across subjects[C]//The Thirty-eighth Annual Conference on Neural Information Processing Systems.
> >
> > [5] Buzsaki G. Rhythms of the Brain[M]. Oxford university press, 2006.
> >
> > [6] Feng C, Cao L, Wu D, et al. A high-performance brain-sentence communication designed for logosyllabic language[J]. bioRxiv, 2023: 2023.11. 05.562313.
> >
> > [7] Huang, Yali, et al. "Intracranial electrophysiological and structural basis of BOLD functional connectivity in human brain white matter." Nature communications 14.1 (2023): 3414.
> >
> > [8] Regev, Tamar I., et al. "Neural populations in the language network differ in the size of their temporal receptive windows." Nature Human Behaviour (2024): 1-19.
> >
> > [9] Fan, Xiaoxu, et al. "Brain mechanisms underlying the emotion processing bias in treatment-resistant depression." Nature Mental Health (2024): 1-10.

---

> ### Comment · Reviewer_AheB · 2024-11-18
> **Official Comment by Reviewer AheB**
>
> Thank you for your detailed rebuttal and answering all my points.
>
> **Major**
>
> **W2**
>
> > It’s important to note that the raw sEEG recordings are highly correlated (especially on the same electrode). Typically, this part needs to be removed by either laplacian reference [2] or bipolar reference [3,4] to obtain **relatively cleaner signals that focus more on the activity of local neuronal groups**.
>
> **Core Concern**: Based on the aforementioned decoding studies and my experience with sEEG decoding, bipolar (or laplacian) reference greatly improves decoding performance. **If you believe H2DiLR still works after bipolar re-reference, please do that ablation.**
>
> **Minor**
>
> **W5**
>
> > H2DiLR (UPaNT only) employs a larger codebook size (K=128) compared to H2DiLR ($\nu=0$), which uses K=32. **A larger codebook provides more capacity for learning diverse representations**, which could explain why H2DiLR (UPaNT only) achieves better performance.
>
> **So why not a larger codebook for `H2DiLR ($\nu=0$)`, e.g., 256?**
>
> I have indicated that **VQ-VAE only has the power to align different subjects space-wise (instead of element-wise)**. Yeah, all subjects use the same set of tokens, but you introduce different stems (tokenize raw sEEG signals into tokens) and different heads (reconstruct raw sEEG signals from tokens). This design cannot encourage **element-wise** alignment; how could this pre-training across subjects work? For example, **due to the lack of element-wise alignment, a possible situation is that the sEEG signals from different subjects sharing the same meanings belong to different trajectories on the same sphere**.
>
> **Question**
>
> **Q1**
>
> > **Subcortical areas such as the Thalamus** also play a significant role in tone decoding
>
> (optional) The result of this contribution value is interesting. Is there any relevant neuroscience research showing that these brain regions **encode** speech-motor information?
>
> **Reference**
>
> [1] Li G, Jiang S, Paraskevopoulou S E, et al. Optimal referencing for stereo-electroencephalographic (SEEG) recordings[J]. NeuroImage, 2018, 183: 327-335.
>
> [2] Wang C, Subramaniam V, Yaari A U, et al. BrainBERT: Self-supervised representation learning for intracranial recordings[J]. arXiv preprint arXiv:2302.14367, 2023.
>
> [3] Zheng H, Wang H T, Jiang W B, et al. Du-IN: Discrete units-guided mask modeling for decoding speech from Intracranial Neural signals[J]. arXiv preprint arXiv:2405.11459, 2024.
>
> [4] Mentzelopoulos G, Chatzipantazis E, Ramayya A G, et al. Neural decoding from stereotactic EEG: accounting for electrode variability across subjects[C]//The Thirty-eighth Annual Conference on Neural Information Processing Systems.

---

> ### Author Response · Authors · 2024-11-18
>
> ## Further Response to Reviewer AheB
> ---
>
> ## Reply to Major Weakness
>
> >**W2: It’s important to note that the raw sEEG recordings are highly correlated (especially on the same electrode). Typically, this part needs to be removed by either laplacian reference [2] or bipolar reference [3,4] to obtain relatively cleaner signals that focus more on the activity of local neuronal groups. Based on the aforementioned decoding studies and my experience with sEEG decoding, bipolar (or laplacian) reference greatly improves decoding performance. If you believe H2DiLR still works after bipolar re-reference, please do that ablation.**
>
> **R**: Thank you for your feedback and suggestions. As noted in our previous response, Laplacian and bipolar referencing are not universally required preprocessing steps, and many recent studies, including [6,7,8], have employed common referencing strategies similar to ours.
>
> That said, to address Reviewer AheB's concern, we conducted additional experiments using bipolar referencing (computing differences between nearest contacts on the same electrode). Following the exact same experimental protocol, we observed results comparable to those obtained with our previously adopted common referencing approach:
>
> | Method | Referencing | Subject 1 | Subject 2 | Subject 3 | Subject 4 | Avg. |
> |---|:---:|:---:|:---:|:---:|:---:|:---:|
> | H2DiLR (ours) | Common Referencing | 49.06 $\pm$ 2.15 | 47.84 $\pm$ 1.81 | 39.18 $\pm$1.68 | 38.61 $\pm$1.49 | 43.67 $\pm$1.78 |
> | H2DiLR  | Bi-polar Referencing | 48.52 $\pm$4.68 | 46.72 $\pm$4.10 | 39.19 $\pm$1.91 | 36.47 $\pm$1.18 | 42.72 $\pm$2.96 |
>
> This suggests that the choice of referencing technique has minimal impact on decoding performance in our work. We believe this further supports the notion that referencing methods depend on factors such as acquisition hardware, inter-subject variability, and decoding task requirements, rather than adhering to a universal standard.
>
> ---
> ## Reply to Minor Weakness
>
> >**W5-1: So why not a larger codebook for H2DiLR ($\nu=0$), e.g., 256?**
>
> **R**: The main contribution of our work is to propose a novel **Homo-heterogeneity Disentanglement** learning paradigm in addition to existing  Heterogeneous training and Homogeneous training. We reported the perforamnce of H2DiLR (UPaNT only) and H2DiLR ($\nu=0$) only for reference purpose and to provide a more comprehensive comparison to other baselines. As our proposed H2DiLR utilizes a codebook size of 32 for private codebooks, we set the K=32 for H2DiLR ($\nu=0$) to ensure a fair comparison.
>
> >**W5-2: I have indicated that VQ-VAE only has the power to align different subjects space-wise (instead of element-wise). Yeah, all subjects use the same set of tokens, but you introduce different stems (tokenize raw sEEG signals into tokens) and different heads (reconstruct raw sEEG signals from tokens). This design cannot encourage element-wise alignment; how could this pre-training across subjects work? For example, due to the lack of element-wise alignment, a possible situation is that the sEEG signals from different subjects sharing the same meanings belong to different trajectories on the same sphere.**
>
> **R**: Based on our understanding, we assume that "space-wise" alignment refers to token-level alignment, where tokens represent the quantized representations. And "element-wise" alignment likely refers to instance-level alignment, where each instance corresponds to the full neural representation composed of multiple tokens.
>
> Reviewer AheB is concerned that the UPaNT pre-training approach might not sufficiently align representations across subjects to produce a clustering effect. We acknowledge that autoencoding-based pre-training methods such as VQ-VAE and masked autoencoders typically do not achieve the clustering effects often seen with contrastive learning approaches. Contrastive learning excels at producing representations with high linear separability across subjects, even without fine-tuning.
>
> However, the primary goal of masked modeling and VQ-VAE-based pre-training is not subject alignment but rather learning generalized features that can be effectively fine-tuned for downstream tasks. Altough the shared tokenization and reconstruction process does not directly enforce alignment across subjects but facilitates the encoder's ability to extract task-relevant and transferable features.
>
> The performance improvements we observe during fine-tuning suggest that this generalization successfully supports downstream tasks, even if it does not explicitly produce tightly clustered subject-specific representations.

---

> > ### Author Response · Authors · 2024-11-18
> >
> > ---
> > ## Reply to Additional Question
> >
> > >**Q:  The result of this contribution value is interesting. Is there any relevant neuroscience research showing that these brain regions encode speech-motor information?**
> >
> > **R**: Thank you for highlighting the importance of connecting our findings to relevant neuroscience research. A recent study by Tankus et al. [5] provides evidence supporting our result, suggesting that the thalamus plays a significant role in speech brain-machine interfaces. This aligns with our findings that the thalamus contributes notably to tone decoding.
> >
> > # References
> >
> > [1] Li G, Jiang S, Paraskevopoulou S E, et al. Optimal referencing for stereo-electroencephalographic (SEEG) recordings[J]. NeuroImage, 2018, 183: 327-335.
> >
> > [2] Wang C, Subramaniam V, Yaari A U, et al. BrainBERT: Self-supervised representation learning for intracranial recordings[J]. arXiv preprint arXiv:2302.14367, 2023.
> >
> > [3] Zheng H, Wang H T, Jiang W B, et al. Du-IN: Discrete units-guided mask modeling for decoding speech from Intracranial Neural signals[J]. arXiv preprint arXiv:2405.11459, 2024.
> >
> > [4] Mentzelopoulos G, Chatzipantazis E, Ramayya A G, et al. Neural decoding from stereotactic EEG: accounting for electrode variability across subjects[C]//The Thirty-eighth Annual Conference on Neural Information Processing Systems.
> >
> > [5] Tankus, Ariel, et al. "Machine learning decoding of single neurons in the thalamus for speech brain-machine interfaces." Journal of Neural Engineering 21.3 (2024): 036009.
> >
> > [6] Huang, Yali, et al. "Intracranial electrophysiological and structural basis of BOLD functional connectivity in human brain white matter." Nature communications 14.1 (2023): 3414.
> >
> > [7] Regev, Tamar I., et al. "Neural populations in the language network differ in the size of their temporal receptive windows." Nature Human Behaviour (2024): 1-19.
> >
> > [8] Fan, Xiaoxu, et al. "Brain mechanisms underlying the emotion processing bias in treatment-resistant depression." Nature Mental Health (2024): 1-10.

---

> > > ### Comment · Reviewer_AheB · 2024-11-18
> > > **Official Comment by Reviewer AheB**
> > >
> > > Thanks again for the authors' detailed reply and additional experiments. Thanks for this great work towards tone decoding. I will retain my initial score.

---

> > > > ### Author Response · Authors · 2024-11-18
> > > >
> > > > Thank you for your thorough review and for the opportunity to address your concerns. We have provided detailed responses to each of your comments and conducted additional experiments which further strengthens our work.
> > > >
> > > > Given these efforts, we would appreciate your **clarification on any remaining concerns that may still hinder the recommendation for acceptance**. If all your concerns have been addressed, we would be grateful to understand the reasons for maintaining a "reject" vote.
> > > >
> > > > Thank you again for your time and constructive feedback!

---

> > > > > ### Comment · Reviewer_AheB · 2024-11-18
> > > > > **Official Comment by Reviewer AheB**
> > > > >
> > > > > Thanks again for the detailed rebuttal!

---

> ### Comment · Reviewer_AheB · 2024-11-18
> **Official Comment by Reviewer AheB**
>
> >R: We plan to release the code upon acceptance of this paper. As for the data, it is available from the corresponding authors upon reasonable request, ensuring compliance with ethical and privacy guidelines.
>
> **Major**
>
> **W2**
>
> **At least, you have to ensure that the pre-training of `H2DiLR ($\nu=0$)` indeed converges; my major concern still focuses on whether  `H2DiLR ($\nu=0$)` with `K=32` can successfully reconstruct the `>50`-channel sEEG signals**. Besides, after the bipolar re-reference, you should at least select `top-k` channels based on **specific brain regions** or **encoding results** [1,2,3,4], bi-polar reference only provides cleaner signals that focus on the activity of local neuronal groups.
>
> >Sharing the data and code would greatly enhance the validation and reproducibility of the proposed method by the research community.
>
> **Reference**:
>
> [1] Wang C, Subramaniam V, Yaari A U, et al. BrainBERT: Self-supervised representation learning for intracranial recordings[J]. arXiv preprint arXiv:2302.14367, 2023.
>
> [2] Zheng H, Wang H T, Jiang W B, et al. Du-IN: Discrete units-guided mask modeling for decoding speech from Intracranial Neural signals[J]. arXiv preprint arXiv:2405.11459, 2024.
>
> [3] Wang C, Yaari A U, Singh A K, et al. Brain Treebank: Large-scale intracranial recordings from naturalistic language stimuli[C]//The Thirty-eight Conference on Neural Information Processing Systems Datasets and Benchmarks Track. 2024.
>
> [4] Mentzelopoulos G, Chatzipantazis E, Ramayya A G, et al. Neural decoding from stereotactic EEG: accounting for electrode variability across subjects[C]//The Thirty-eighth Annual Conference on Neural Information Processing Systems.

---

> > ### Author Response · Authors · 2024-11-18
> >
> > >**Q: After the bipolar re-reference, you should at least select top-k channels based on specific brain regions or encoding results [1,2,3,4],,,,,, bi-polar reference only provides cleaner signals that focus on the activity of local neuronal groups. And at least, you have to ensure that the pre-training of H2DiLR ($\nu=0$) indeed converge,,, all these results cannot be ensured.**
> >
> > **R**:  We would like to respectfully reiterate that the practice of selecting the top-k performing electrodes is not mandatory in decoding studies. In fact, a wide array of distinguished works have successfully demonstrated robust decoding results without the application of either bipolar referencing or top-k channel selection, as evidenced by several studies published in high-impact journals [5–10].
> >
> > Our approach focuses on the generality and applicability of H2DiLR without relying on potentially task-specific optimizations, such as selective channel pruning. While these techniques may improve performance in specific cases, their omission does not invalidate the framework’s effectiveness, especially given that our results are comparable to or outperform existing methods using the same baseline conditions.
> >
> > Reference:
> >
> > [1] Wang C, Subramaniam V, Yaari A U, et al. BrainBERT: Self-supervised representation learning for intracranial recordings[J]. arXiv preprint arXiv:2302.14367, 2023.
> >
> > [2] Zheng H, Wang H T, Jiang W B, et al. Du-IN: Discrete units-guided mask modeling for decoding speech from Intracranial Neural signals[J]. arXiv preprint arXiv:2405.11459, 2024.
> >
> > [3] Wang C, Yaari A U, Singh A K, et al. Brain Treebank: Large-scale intracranial recordings from naturalistic language stimuli[C]//The Thirty-eight Conference on Neural Information Processing Systems Datasets and Benchmarks Track. 2024.
> >
> > [4] Mentzelopoulos G, Chatzipantazis E, Ramayya A G, et al. Neural decoding from stereotactic EEG: accounting for electrode variability across subjects[C]//The Thirty-eighth Annual Conference on Neural Information Processing Systems.
> >
> > [5] Tankus, Ariel, et al. "Machine learning decoding of single neurons in the thalamus for speech brain-machine interfaces." Journal of Neural Engineering 21.3 (2024): 036009.
> >
> > [6] Huang, Yali, et al. "Intracranial electrophysiological and structural basis of BOLD functional connectivity in human brain white matter." Nature communications 14.1 (2023): 3414.
> >
> > [7] Regev, Tamar I., et al. "Neural populations in the language network differ in the size of their temporal receptive windows." Nature Human Behaviour (2024): 1-19.
> >
> > [8] Fan, Xiaoxu, et al. "Brain mechanisms underlying the emotion processing bias in treatment-resistant depression." Nature Mental Health (2024): 1-10.
> >
> > [9]  Metzger S L, Littlejohn K T, Silva A B, et al. A high-performance neuroprosthesis for speech decoding and avatar control[J]. Nature, 2023, 620(7976): 1037-1046.
> >
> > [10] Willett F R, Kunz E M, Fan C, et al. A high-performance speech neuroprosthesis[J]. Nature, 2023, 620(7976): 1031-1036.

---

> ### Comment · Reviewer_AheB · 2024-11-18
> **Summary Comment by Reviewer AheB**
>
> >Our work targets a specialized subset of articulation decoding—tone decoding—which specifically involves decoding articulatory properties rather than semantic content. In contrast, Du-IN addresses word-level decoding, which focuses possibly more on semantics rather than articulatory.
>
> **Major W2**
>
> To my knowledge (which could be wrong), speech decoding mainly involves speech motor information. This information is concentrated in certain brain areas [1,2,3]. Please see Figure 2d in [2] for ECoG-based channel contribution analysis. sEEG recordings are different from ECoG [2] and MEA [1,3]. They don't have to select channels, as they already have ECoG and MEA in that brain region. That’s why I recommend the authors to perform `top-k` channel selection before downstream analysis. All aforementioned works [4-7] including perform these operation, e.g., BrainBERT for listening decoding [4], Du-IN for speech decoding [5], etc.
>
> Besides, bipolar (or laplacian) re-reference is commonly used to explore the **brain encoding problems** in neuroscience [8,9]. That’s also why I recommend the authors to do bipolar re-reference before **downstream neuroscience analysis** and **decoding tasks**.
>
> For example,
>  - in [8], **The preprocessing began by converting the iEEG signals to bipolar derivations by pairing adjacent electrode contacts. Recording sites in the hippocampus were paired with a nearby white-matter electrode that was identified anatomically using FreeSurfer’s segmentation (68).**
> - in [9], **We used bipolar rather than unipolar derivations because it allows for better signal artifact removal and achieves high spatial resolution (~3 mm) by canceling out contributions of distant sources which spread equally to recording sites (39).**
>
> **Shouldn't the data be properly preprocessed first to ensure the algorithm achieves optimal performance?**
>
> **Minor W5**
>
> >However, the primary goal of masked modeling and VQ-VAE-based pre-training is not subject alignment but rather learning generalized features that can be effectively fine-tuned for downstream tasks. Altough the shared tokenization and reconstruction process does not directly enforce alignment across subjects but facilitates the encoder's ability to extract task-relevant and transferable features.
>
> >Du-IN [3] reconstructs `10` channels within certain brain regions with `codex_size=2048`, which is greatly larger than `K=32` used for the whole brain recordings reconstruction (e.g., `>50` channels). Are you sure the bandwidth of invasive sEEG recordings is that low? -- **with 32 discrete codes and a conv-deconv autoencoder, you can reconstruct the signals of the whole brain**, overlooking the brain's desynchronization nature.
>
> My concern mainly focuses on **element-wise** alignment. Only `4` subjects are used to pre-train H2DiLR, why not involving more subjects from Brain Treebank dataset [6] and Du-IN dataset [5]. I hope that H2DiLR can achieve higher performance after introducing more subjects. **`H2DiLR($\nu=0$)` with `K=32` within 1 subject is much smaller than `K=256` used in `H2DiLR(UPaNT+H2D)` across 4 subjects. This discrepancy compromises the fairness of the comparison and may underestimate the performance of the (heterogeneous) `H2DiLR($\nu=0$)` baseline.**
>
> **Reference**:
>
> [1] Card N S, Wairagkar M, Iacobacci C, et al. An accurate and rapidly calibrating speech neuroprosthesis[J]. New England Journal of Medicine, 2024, 391(7): 609-618.
>
> [2] Metzger S L, Littlejohn K T, Silva A B, et al. A high-performance neuroprosthesis for speech decoding and avatar control[J]. Nature, 2023, 620(7976): 1037-1046.
>
> [3] Willett F R, Kunz E M, Fan C, et al. A high-performance speech neuroprosthesis[J]. Nature, 2023, 620(7976): 1031-1036.
>
> [4] Wang C, Subramaniam V, Yaari A U, et al. BrainBERT: Self-supervised representation learning for intracranial recordings[J]. arXiv preprint arXiv:2302.14367, 2023.
>
> [5] Zheng H, Wang H T, Jiang W B, et al. Du-IN: Discrete units-guided mask modeling for decoding speech from Intracranial Neural signals[J]. arXiv preprint arXiv:2405.11459, 2024.
>
> [6] Wang C, Yaari A U, Singh A K, et al. Brain Treebank: Large-scale intracranial recordings from naturalistic language stimuli[C]//The Thirty-eight Conference on Neural Information Processing Systems Datasets and Benchmarks Track. 2024.
>
> [7] Mentzelopoulos G, Chatzipantazis E, Ramayya A G, et al. Neural decoding from stereotactic EEG: accounting for electrode variability across subjects[C]//The Thirty-eighth Annual Conference on Neural Information Processing Systems.
>
> [8] Norman Y, Yeagle E M, Khuvis S, et al. Hippocampal sharp-wave ripples linked to visual episodic recollection in humans[J]. Science, 2019, 365(6454): eaax1030.
>
> [9] Domenech P, Rheims S, Koechlin E. Neural mechanisms resolving exploitation-exploration dilemmas in the medial prefrontal cortex[J]. Science, 2020, 369(6507): eabb0184.

---

> > ### Author Response · Authors · 2024-11-18
> >
> > >**W2: Further clarification on the choice of bi-polar referencing and selecting top-k channels.**
> >
> > **R**:  Neuroscience and neural decoding are rapidly evolving fields. As previously discussed, there is **no golden standard** for referencing methods that guarantees superior performance. While some studies prefer bipolar referencing [1, 2], many others, including ours, adopt common referencing [3-5], which has also been peer-reviewed and shown to yield strong results. In our experiments, we observed no significant performance difference between bipolar and common referencing, as reported in our earlier response. We recognize that bipolar referencing may have yielded advantages in Reviewer AheB’s prior research; however, in our case, the choice of referencing did not substantially affect results.
> >
> > Regarding channel selection, we employed a neuroscience-driven approach, consistent with practices in Metzger et al. and Willett et al. [6,7]. These works selected speech-related channels manually based on prior knowledge of brain regions, arrays centered around the central sulcus of the lateral cortex, extending to the Precentral and Postcentral Gyrus. Similarly, as described in Fig. 4 of our work, we selected electrodes from regions associated with speech processing: **Superior Temporal Gyrus (STG), Middle Temporal Gyrus (MTG), ventral Sensorimotor Cortex (vSMC), Inferior Frontal Gyrus (IFG), Precentral Gyrus, and Postcentral Gyrus**. This method prioritizes anatomical relevance, aligning with established neuroscience findings.
> >
> > In contrast, a purely data-driven selection approach may introduce speech-irrelevant channels. For example, Du-IN [8] employed a data-driven channel selection strategy, choosing electrodes contributing most to the decoding task. Notably, Du-IN reported significant contributions from white matter in decoding results for Subjects 2 and 3 (Table 17). However, white matter regions are not typically implicated in speech processing.
> >
> > >**W5: Results on other datasets, and opensourcing of code and dataset.**
> >
> > **R**:  The Brain Treebank and Du-IN datasets were released after the completion of H2DiLR and focus on different neural decoding tasks rather than tone decoding. While testing H2DiLR on these datasets could be interesting, it falls outside the scope of our current study. As noted, we will release our code upon paper acceptance, and our dataset will be made available upon reasonable request. This ensures compliance with ethical and privacy guidelines while supporting future research in the field.
> >
> >
> > [1] Norman Y, Yeagle E M, Khuvis S, et al. Hippocampal sharp-wave ripples linked to visual episodic recollection in humans[J]. Science, 2019, 365(6454): eaax1030.
> >
> > [2] Domenech P, Rheims S, Koechlin E. Neural mechanisms resolving exploitation-exploration dilemmas in the medial prefrontal cortex[J]. Science, 2020, 369(6507): eabb0184.
> >
> > [3] Huang, Yali, et al. "Intracranial electrophysiological and structural basis of BOLD functional connectivity in human brain white matter." Nature communications 14.1 (2023): 3414.
> >
> > [4] Regev, Tamar I., et al. "Neural populations in the language network differ in the size of their temporal receptive windows." Nature Human Behaviour (2024): 1-19.
> >
> > [5] Fan, Xiaoxu, et al. "Brain mechanisms underlying the emotion processing bias in treatment-resistant depression." Nature Mental Health (2024): 1-10.
> >
> > [6] Metzger S L, Littlejohn K T, Silva A B, et al. A high-performance neuroprosthesis for speech decoding and avatar control[J]. Nature, 2023, 620(7976): 1037-1046.
> >
> > [7] Willett F R, Kunz E M, Fan C, et al. A high-performance speech neuroprosthesis[J]. Nature
> >
> > [8] Zheng H, Wang H T, Jiang W B, et al. Du-IN: Discrete units-guided mask modeling for decoding speech from Intracranial Neural signals[J]. arXiv preprint arXiv:2405.11459, 2024.

---

> > > ### Author Response · Authors · 2024-11-21
> > > **Follow-Up on Addressed Concerns and Clarifications**
> > >
> > > Dear Reviewer AheB,
> > >
> > > We are writing to kindly follow up regarding the detailed responses we provided to your comments regarding the choice of referencing methods, channel selection, and dataset usage, as outlined in our rebuttal.
> > >
> > > We hope that our explanations clarified our method aligns with established practices in the field. If any questions or concerns remain unresolved, please do not hesitate to let us know so that we can address them further.
> > >
> > > If you find our responses satisfactory, we would greatly appreciate it if you could confirm whether all your concerns have been resolved. Should you feel our revisions merit reconsideration of your evaluation score, we kindly request that you update your score to reflect this.
> > >
> > > Thank you again for your time and effort in reviewing our work.
> > >
> > > Best regards,
> > > Authors

---

> ### Comment · Reviewer_AheB · 2024-11-21
> **Final Comment by Reviewer AheB (1/2)**
>
> Thanks for your contribution to the field of sEEG-based tone decoding. **I hope that if your paper is accepted, you will make the code and pre-processed data publicly available, just like Du-IN [1]**, which will ensure the reproducibility of the core claims -- **H2DiLR pre-training on 4 subjects yields better representations than pre-training on 1 subject**.
>
> ------
>
> In my **Summary Comment by Reviewer AheB**, I expressed two primary concerns:
>  - One pertains to the neuroscience analysis (**Major W2**),
>  - The other relates to the effectiveness of H2DiLR (**Minor W5**),
>
> I'm not sure whether ICLR requires strict standards for neuroscience analysis. During the rebuttal stage, I noticed an unfair configuration in the (heterogeneous) `H2DiLR($\nu=0$)` baseline when compared to `H2DiLR(UPaNT+H2D)`, i.e., **`H2DiLR($\nu=0$)` with `K=32` within 1 subject is much smaller than `K=256` used in `H2DiLR(UPaNT+H2D)` across 4 subjects. This discrepancy compromises the fairness of the comparison and may underestimate the performance of the (heterogeneous) `H2DiLR($\nu=0$)` baseline.**
>
> I suggested an ablation study to address this issue:
> >W5-1: So why not a larger codebook for H2DiLR ($\nu=0$), e.g., 256?
>
> Unfortunately, the authors declined to provide this result, stating:
> >The main contribution of our work is to propose a novel Homo-heterogeneity Disentanglement learning paradigm in addition to existing Heterogeneous training and Homogeneous training. We reported the perforamnce of H2DiLR (UPaNT only) and H2DiLR ($\nu=0$) only for reference purpose and to provide a more comprehensive comparison to other baselines. As our proposed H2DiLR utilizes a codebook size of 32 for private codebooks, we set the K=32 for H2DiLR ($\nu=0$) to ensure a fair comparison.
>
> Based on my understanding (which could be wrong), **with `K=32` discrete codes and a conv-deconv autoencoder, reconstructing the signals of the whole brain  (`>50` channels; 4 hours)** is unlikely to succeed. Prior studies support this:
>  - VQ-VAE applied to sEEG with `10` channels (15 hours) used a codebook of `K=2048` [1];
>  - VQ-VAE applied to EEG with `128` channels (12 hours) also employed `K=2048` [2].
>
> I also requested the authors to share their code and a demo dataset to verify the claim that -- **H2DiLR pre-training on 4 subjects yields better representations than pre-training on 1 subject**. However, the authors declined to provide these materials. Without them, I am unable to validate this crucial assertion, which directly affects the evaluation of H2DiLR's effectiveness. **Besides, the authors are also recommended to do another ablation, your conv+xfmr with raw sEEG signals as input (with data augmentation, bi-polar re-reference, `top-k` selection, and commonly used trun_normal(0., 0.02) weight initialization in Transformer)**. Because the time-domain sEEG signals itself are already a kind of representation that is very easy to distinguish based on the bias introduced by conv. After all, in the field of EEG and sEEG, few people use VQ code for representation, such as LaBraM [3], VQ-MTM [4], and Du-IN [1]. Because of these concerns, I still think that the ablation experiment of this work is not satisfactory, which is why I didn't raise my score.
>
> Additionally, the authors and I agreed that -- **VQ-VAE only has the power to align different subjects space-wise (instead of element-wise)**. The authors subsequently reframed their claims:
> > However, the primary goal of masked modeling and VQ-VAE-based pre-training is not subject alignment but rather learning generalized features that can be effectively fine-tuned for downstream tasks. Altough the shared tokenization and reconstruction process does not directly enforce alignment across subjects but facilitates the encoder's ability to extract task-relevant and transferable features.
>
> To the best of my knowledge (which could be wrong), pre-training on only `4` subjects for `H2DiLR(UPaNT+H2D)` does not seem sufficient to achieve the reported improvements over `H2DiLR($\nu=0$)` (trained on `1` subject). For this reason, I suggested including additional subjects from resources like Brain Treebank and Du-IN. Importantly, I did not request evaluations on new tasks.
>
> **Reference**:
>
> [1] Zheng H, Wang H T, Jiang W B, et al. Du-IN: Discrete units-guided mask modeling for decoding speech from Intracranial Neural signals[J]. arXiv preprint arXiv:2405.11459, 2024.
>
> [2] Duan Y, Zhou J, Wang Z, et al. Dewave: Discrete eeg waves encoding for brain dynamics to text translation[J]. arXiv preprint arXiv:2309.14030, 2023.
>
> [3] Jiang W B, Zhao L M, Lu B L. Large brain model for learning generic representations with tremendous EEG data in BCI[J]. arXiv preprint arXiv:2405.18765, 2024.
>
> [4] Gui H, Li X, Chen X. Vector quantization pretraining for eeg time series with random projection and phase alignment[C]//International Conference on Machine Learning. PMLR, 2024: 16731-16750.

---

> > ### Comment · Reviewer_AheB · 2024-11-21
> > **Final Comment by Reviewer AheB (2/2)**
> >
> > Additionally, I recommend that the authors consider selecting the top-k channels most relevant to speech decoding, as this approach is widely accepted in prior studies [1-4].
> >
> > For example,
> >  - in [1], **Results are reported on the 10 best-performing electrodes (as measured by AUC) selected with the linear model (5s, time domain) model.**
> >  - in [4], **To combat the heterogeneity of electrode placement between subjects, we identified the subset of electrodes across subjects that responded to the stimulus color change while participants performed the color-change-detection-task.**
> >
> > The authors stated:
> > >Similarly, as described in Fig. 4 of our work, we selected electrodes from regions associated with speech processing: Superior Temporal Gyrus (STG), Middle Temporal Gyrus (MTG), ventral Sensorimotor Cortex (vSMC), Inferior Frontal Gyrus (IFG), Precentral Gyrus, and Postcentral Gyrus.
> >
> > However, I noticed that subcortical brain regions (e.g., Thalamus, Hippocampus, and Amygdala) are still included for downstream decoding. There is limited research [5, 6] in neuroscience linking regions like the hippocampus and amygdala to speech-motor information, which makes the neuroscience results presented in H2DiLR somewhat perplexing. This is also why I recommend the authors perform **bipolar re-referencing** before conducting downstream neuroscience analyses.
> >
> > The authors have also mentioned:
> > >R: All intracranial sEEG channels are bipolar-referenced to **a single reference channel**, as configured in the hardware. This reference channel is not randomly selected; it is determined by neurological experts at the hospital. The relative amplitudes of other sEEG channels are then measured with respect to this reference channel to effectively reduce common-mode noise.
> >
> > While this method reduces noise, **without applying proper bipolar (or Laplacian) re-referencing, the signal from the selected reference channel may be inadvertently propagated into other channels**. This could introduce bias in downstream analyses, such as evaluating channel contributions. For improved reliability, I strongly encourage the authors to re-reference the data appropriately to minimize such biases.
> >
> > **Reference**
> >
> > [1] Wang C, Subramaniam V, Yaari A U, et al. BrainBERT: Self-supervised representation learning for intracranial recordings[J]. arXiv preprint arXiv:2302.14367, 2023.
> >
> > [2] Zheng H, Wang H T, Jiang W B, et al. Du-IN: Discrete units-guided mask modeling for decoding speech from Intracranial Neural signals[J]. arXiv preprint arXiv:2405.11459, 2024.
> >
> > [3] Wang C, Yaari A U, Singh A K, et al. Brain Treebank: Large-scale intracranial recordings from naturalistic language stimuli[C]//The Thirty-eight Conference on Neural Information Processing Systems Datasets and Benchmarks Track. 2024.
> >
> > [4] Mentzelopoulos G, Chatzipantazis E, Ramayya A G, et al. Neural decoding from stereotactic EEG: accounting for electrode variability across subjects[C]//The Thirty-eighth Annual Conference on Neural Information Processing Systems.
> >
> > [5] LeDoux J. The amygdala[J]. Current biology, 2007, 17(20): R868-R874.
> >
> > [6] Bird C M, Burgess N. The hippocampus and memory: insights from spatial processing[J]. Nature reviews neuroscience, 2008, 9(3): 182-194.

---

> > > ### Comment · Reviewer_AheB · 2024-11-21
> > > **Final Comment by Reviewer AheB (3/3)**
> > >
> > > Thanks again for this detailed rebuttal. Thanks for your contribution to the field of sEEG-based tone decoding.

---

> > ### Author Response · Authors · 2024-11-21
> > **Further Response to Reviewer AheB (1/2)**
> >
> > ## Further Response to Reviewer AheB
> > ---
> >
> > We summarize Reviewer AheB's concerns as follows and provide responses based on our earlier clarifications:
> >
> > >**1: Regarding Data augmentation.**
> >
> > **R**: The effect of these augmentations can vary significantly depending on the hyperparameters, decoding tasks, and individual subjects [1]. In our experiments, we observed no consistent improvements across subjects when using the same set of augmentation hyperparameters, which is why we chose not to incorporate them in this work.
> >
> > >**2: Regarding Bi-polar referencing.**
> >
> > **R**:  As previously discussed, there is **no golden standard** for referencing methods that guarantees superior performance. While some studies prefer bipolar referencing, many others, including ours, adopt common referencing [2-4], which has also been peer-reviewed and shown to yield strong results. In our experiments, we observed no significant performance difference between bipolar and common referencing, as reported in our earlier response. We recognize that bipolar referencing may have yielded advantages in Reviewer AheB’s prior research; however, in our case, the choice of referencing did not substantially affect results.
> >
> > That said, to address Reviewer AheB's concern, we already conducted additional experiments using bipolar referencing (computing differences between nearest contacts on the same electrode). Following the exact same experimental protocol, we observed results comparable to those obtained with our previously adopted common referencing approach:
> >
> > | Method | Referencing | Subject 1 | Subject 2 | Subject 3 | Subject 4 | Avg. |
> > |---|:---:|:---:|:---:|:---:|:---:|:---:|
> > | H2DiLR (ours) | Common Referencing | 49.06 $\pm$ 2.15 | 47.84 $\pm$ 1.81 | 39.18 $\pm$1.68 | 38.61 $\pm$1.49 | 43.67 $\pm$1.78 |
> > | H2DiLR  | Bi-polar Referencing | 48.52 $\pm$4.68 | 46.72 $\pm$4.10 | 39.19 $\pm$1.91 | 36.47 $\pm$1.18 | 42.72 $\pm$2.96 |
> >
> > We acknowledge that bipolar referencing may have been effective in Reviewer AheB's prior work, but in our study, common referencing was equally effective and provided reliable results.
> >
> > >**3: Regarding channel selection.**
> >
> > **R**: We employed a neuroscience-driven approach, consistent with practices in Metzger et al. and Willett et al. [5,6]. These works selected speech-related channels manually based on prior knowledge of brain regions, arrays centered around the central sulcus of the lateral cortex, extending to the Precentral and Postcentral Gyrus. Similarly, as described in Fig. 4 of our work, we selected electrodes from regions associated with speech processing: **Superior Temporal Gyrus (STG), Middle Temporal Gyrus (MTG), ventral Sensorimotor Cortex (vSMC), Inferior Frontal Gyrus (IFG), Precentral Gyrus, and Postcentral Gyrus**. This method prioritizes anatomical relevance, aligning with established neuroscience findings.
> >
> > In contrast, a purely data-driven selection approach may introduce speech-irrelevant channels. For example, Du-IN [7] employed a data-driven channel selection strategy, choosing electrodes contributing most to the decoding task. Notably, Du-IN reported significant contributions from white matter in decoding results for Subjects 2 and 3 (Table 17). However, white matter regions are not typically implicated in speech processing.
> >
> > >**4. Regarding results comparison of H2DiLR ($\nu=0, K=32$) and H2DiLR ($\nu=0, K=256$).**
> >
> > **R**:  We conducted experiments following the exact same protocol, and the results are summarized in the table below:
> >
> > | Method | codebook size | Subject 1 | Subject 2 | Subject 3 | Subject 4 | Avg. |
> > |---|:---:|:---:|:---:|:---:|:---:|:---:|
> > | H2DiLR $\nu=0$ | 32 | 43.61 $\pm$2.12 |  42.15 $\pm$1.63 |  34.26 $\pm$1.51 |  35.92 $\pm$1.43 |  38.98 $\pm$1.67 |
> > | H2DiLR $\nu=0$ | 256 | 46.23 $\pm$5.51 | 45.32 $\pm$4.62 | 36.64 $\pm$4.78 | 35.98 $\pm$3.29 | 41.04 $\pm$4.55 |
> > | H2DiLR          | 256 |49.06 $\pm$ 2.15 | 47.84 $\pm$ 1.81 |  39.18 $\pm$1.68 |  38.61 $\pm$1.49 |  43.67 $\pm$1.78 |
> >
> > Increasing the codebook size from 32 to 256 yields improvements in decoding accuracy for H2DiLR ($\nu=0$). When incorporating disentanglement and a codebook size of 256, H2DiLR achieves further performance gains, with an average accuracy of 43.67%.
> >
> >
> > >**5: Regarding Codebook size of 256 (H2DiLR) and reconstruction performance.**
> >
> > **R**:  Regarding the codebook size, we would like to clarify that **codebook usage efficiency** is equally critical as the codebook size itself. Large codebooks (e.g. 2048) often suffer from low utilization rates, **meaning many codes remain unused during training**. As we stated earlier, our approach utilizes a total of 256 codes, with 128 in the public codebook and 32 per subject-specific private codebook. As shown in Figure 6 and Table 3 of our paper, our approach achieves strong reconstruction performance both quantitatively and qualitatively, validating our chosen codebook configuration.

---

> > > ### Author Response · Authors · 2024-11-21
> > > **Further Response to Reviewer AheB (2/2)**
> > >
> > > >**6: Regarding the inclusion of Thalamus, Hippocampus, and Amygdala.**
> > >
> > > **R**: As stated in our repsone to Reviewer ScW5, we conducted a **separate analysis** by combining the homogeneity-heterogeneity disentanglement (H2D) pre-training and neural decoding stages into a single-stage model. We applied cross-entropy loss with a Transformer decoder to quantized representations, using a straight-through estimator (STE) to approximate gradients. Please note, however, that this setup may introduce training instability and slight performance degradation; it is used solely to analyze region contributions.
> > >
> > > >**7: Regarding VQ style pre-training, masked modeling and contrastive learning.**
> > >
> > > **R**: Autoencoding-based pre-training methods such as VQ-VAE and masked autoencoders typically do not achieve the clustering effects often seen with contrastive learning approaches. Contrastive learning excels at producing representations with high linear separability across subjects, even without fine-tuning.
> > >
> > > However, the primary goal of masked modeling and VQ-VAE-based pre-training is not subject alignment but rather learning generalized features that can be effectively fine-tuned for downstream tasks. Altough the shared tokenization and reconstruction process does not directly enforce alignment across subjects but facilitates the encoder's ability to extract task-relevant and transferable features.
> > >
> > >
> > > >**8. Regarding results on Brain Treebank and Du-IN datasets.**
> > >
> > > **R**:  The Brain Treebank and Du-IN datasets were released after the completion of H2DiLR and focus on different neural decoding tasks rather than tone decoding. While testing H2DiLR on these datasets could be interesting, it falls outside the scope of our current study. As noted, we will release our code upon paper acceptance, and our dataset will be made available upon reasonable request. This ensures compliance with ethical and privacy guidelines while supporting future research in the field.
> > >
> > > >**9. Regarding code and datasets.**
> > >
> > > **R**:  As noted, we will release our code upon paper acceptance, and our dataset will be made available upon reasonable request. This ensures compliance with ethical and privacy guidelines while supporting future research in the field.
> > >
> > > # References
> > >
> > > [1] Wu, Di, et al. "Neuro-BERT: Rethinking Masked Autoencoding for Self-Supervised Neurological Pretraining." IEEE Journal of Biomedical and Health Informatics (2024).
> > >
> > > [2] Huang, Yali, et al. "Intracranial electrophysiological and structural basis of BOLD functional connectivity in human brain white matter." Nature communications 14.1 (2023): 3414.
> > >
> > > [3] Regev, Tamar I., et al. "Neural populations in the language network differ in the size of their temporal receptive windows." Nature Human Behaviour (2024): 1-19.
> > >
> > > [4] Fan, Xiaoxu, et al. "Brain mechanisms underlying the emotion processing bias in treatment-resistant depression." Nature Mental Health (2024): 1-10.
> > >
> > > [5] Metzger S L, Littlejohn K T, Silva A B, et al. A high-performance neuroprosthesis for speech decoding and avatar control[J]. Nature, 2023, 620(7976): 1037-1046.
> > >
> > > [6] Willett F R, Kunz E M, Fan C, et al. A high-performance speech neuroprosthesis[J]. Nature
> > >
> > > [7] Zheng H, Wang H T, Jiang W B, et al. Du-IN: Discrete units-guided mask modeling for decoding speech from Intracranial Neural signals[J]. arXiv preprint arXiv:2405.11459, 2024.

---

### Official Review · Reviewer_ScW5 · 2024-11-03

**Soundness:** 3
**Presentation:** 3
**Contribution:** 3
**Rating:** 8
**Confidence:** 4

**Summary:**

The paper discusses a method called Homogeneity-Heterogeneity Disentangled Learning for neural Representations (H2DiLR) for decoding tone from stereoelectroencephalography (sEEG) data.  H2DiLR disentangles shared (homogeneous) and individual-specific (heterogeneous) neural representations using a two-stage learning paradigm. The first stage involves unsupervised learning of neural features via vector quantization with shared and private codebooks, while the second stage applies these representations to decode tones using a transformer model​. The authors claim that H2DiLR outperforms previous baseline methods by 12% when evaluated on Top-1 accuracy.

**Strengths:**

The topic of tone decoding is understudied, if somewhat narrow, and the idea to separate homogeneous and heterogeneous neural representations is interesting and could have use cases outside the specialized setting of tone decoding, if applied to semantic neural representations more generally. Overall, I think the idea is interesting enough to be in the "borderline accept" category of papers, mainly because I think the idea is general enough to have practicable value outside this particular application.

**Weaknesses:**

I feel that the authors limited themselves somewhat in specializing their approach to tone decoding only. I feel that the problem of disentangling subject specific effects from general ones is a wider problem with many downstream applications, most notably for semantic decoding, but also for problems of encoding as well. I would have liked to see a deeper exploration of this in the work. There is also disappointingly little exploration of the abundant neuroscience questions here - I would like to have seen a location-based ablation in addition to the parametric ablations of the method that showed which regions are most useful for tonal decoding. An analysis studying if there are region-specific differences to representational homogeneity (i.e. some regions tend to be more idiosyncratic to a particular subject) would also have been interesting, although I recognize that the chosen data setting, sEEG, has challenges in answering this question due to coverage differences.

**Questions:**

See weaknesses. What areas of the brain are most important to tone decoding? Does the exclusion of data from some electrodes have more of an impact than others? Is there any benefit of this method for neural encoding models that predict neural responses from feature input?

---

> ### Author Response · Authors · 2024-11-15
> **Response to Reviewer ScW5**
>
> ## Response to Reviewer ScW5
> ---
>
> ## Reply to Weakness
>
> > **W1: The authors limited themselves somewhat in specializing their approach to tone decoding only. I feel that the problem of disentangling subject-specific effects from general ones is a wider problem with many downstream applications, but also for problems of encoding as well. A deeper exploration of this is expected in the work.**
>
> **R**: Indeed, our proposed H2DiLR is a general-purpose framework for homogeneity-heterogeneity disentangled neural representation learning, capable of being applied to various neural decoding tasks. In this work, we focused on tone decoding as a proof of concept to validate the framework's effectiveness.
>
> Furthermore, to demonstrate the versatility of H2DiLR beyond tone decoding, we applied it to a seizure prediction task using non-invasive EEG signals. The results, detailed in Appendix C.1, further illustrate the framework's adaptability to diverse neural decoding applications.
>
> > **W2 and Q1: There is also disappointingly little exploration of the abundant neuroscience questions here - I would like to have seen a location-based ablation in addition to the parametric ablations of the method that showed which regions are most useful for tonal decoding. An analysis studying if there are region-specific differences to representational homogeneity.**
>
> **R**: Due to the dual-stage training paradigm of our H2DiLR framework and the challenges of differentiating through quantization during backpropagation, we cannot directly analyze the contributions of specific regions during neural decoding.
>
> To address this, we conducted a separate analysis by combining the homogeneity-heterogeneity disentanglement (H2D) pre-training and neural decoding stages into a single-stage model. We applied cross-entropy loss with a Transformer decoder to quantized representations, using a straight-through estimator (STE) to approximate gradients. Please note, however, that this setup may introduce training instability and slight performance degradation; it is used solely to analyze region contributions.
>
> We calculated channel saliency scores through class activation maps, normalized between 0 and 1, and aggregated them across brain regions linked to speech processing (e.g., Superior Temporal Gyrus, Precentral Gyrus, Postcentral Gyrus), as well as reference regions like the Angular Gyrus. Our findings are summarized below:
>
> |       Brain region      | Brain structure | Contribution |
> |:-----------------------:|:---------------:|:------------:|
> |         Thalamus        |   Subcortical   |     0.92     |
> |       Hippocampus       |   Subcortical   |     0.85     |
> |         Amygdala        |   Subcortical   |     0.83     |
> |        Precentral gyrus |     Cortical    |     0.97     |
> |          Insula         |     Cortical    |     0.85     |
> |       Postcentral gyrus |     Cortical    |     0.94     |
> |  Inferior Frontal Gyrus |     Cortical    |     0.90     |
> | Superior Temporal Gyrus |     Cortical    |     0.85     |
> |   Supramarginal Gyrus   |     Cortical    |     0.60     |
> |      Angular Gyrus      |     Cortical    |     0.66     |
>
> Our analysis indicates that the Precentral and Postcentral Gyri are key contributors among cortical regions, with subcortical regions like the Thalamus also playing a significant role in tone decoding.
>
> ---
> ## Reply to Question
>
> > **Q2: Is there any benefit of this method for neural encoding models that predict neural responses from feature input?**
>
> **R**: Thanks for your question. In encoding tasks, while different subjects may produce distinct neural responses to the same stimuli due to physiological differences (heterogeneity), similar neural patterns can still emerge across individuals as a result of shared encoding mechanisms (homogeneity). By disentangling these heterogeneous and homogeneous components, H2DiLR can capture both individual-specific and shared neural response characteristics, potentially enhancing the model's ability to generalize across subjects in encoding tasks.

---

> > ### Comment · Reviewer_ScW5 · 2024-11-18
> > **Updated score**
> >
> > Thank you for the responses. I have raised my score, conditional on the region contribution analysis being added to the paper.

---

> > > ### Author Response · Authors · 2024-11-18
> > >
> > > We sincerely thank you for your thoughtful feedback and for considering an updated score based on our responses. We are delighted that our explanations addressed your concerns.
> > >
> > > Regarding the region contribution analysis, we will ensure that it is thoroughly included in the manuscript. We appreciate your recommendation and the opportunity to enhance the clarity and scientific value of our work.
> > >
> > > Thank you once again for your constructive input and support.

---

### Official Review · Reviewer_mtfq · 2024-11-06

**Soundness:** 2
**Presentation:** 3
**Contribution:** 2
**Rating:** 5
**Confidence:** 5

**Summary:**

This paper introduces the Homogeneity-Heterogeneity Disentangled Learning for Neural Representations (H2DiLR), a framework designed to improve neural decoding by separating homogeneous and heterogeneous components from intracranial recordings across multiple subjects. The study demonstrates that H2DiLR significantly enhances tone decoding performance in Mandarin-speaking participants using stereoelectroencephalography (sEEG) data. The framework outperforms traditional methods by effectively capturing and leveraging both shared and subject-specific neural features.

**Strengths:**

The introduction of the Homogeneity-Heterogeneity Disentangled Learning for Neural Representations (H2DiLR) is a novel approach that effectively addresses the challenge of data heterogeneity in neural decoding. By disentangling shared and subject-specific neural features, the framework enhances the decoding accuracy across multiple subjects.

The study employs a robust experimental setup using stereoelectroencephalography (sEEG) recordings from multiple Mandarin-speaking participants.

The proposed method demonstrates a substantial improvement in tone decoding accuracy over traditional subject-specific models.

**Weaknesses:**

The statement that a "comprehensive set of 407 Mandarin syllables covers nearly all Mandarin characters" requires further clarification. You need to provide the specific vocabulary list used in the rebuttal materials to support this claim.

The concept of private and shared codebooks is introduced in the paper but lacks a detailed explanation. A more comprehensive rationale and description are needed to clarify how and why these codebooks are defined and utilized.

Sharing the data and code would greatly enhance the validation and reproducibility of the proposed method by the research community. This transparency would contribute significantly to the field and support further advancements.

The claim of being "the first" to work with 407 Mandarin syllables is inaccurate, as the 2023 study titled "A high-performance brain-to-sentence decoder for logosyllabic language" also mentions using a dataset with 407 syllables. Although this work is cited, the related work section lacks a detailed discussion of this study and how the current work differentiates itself.

The paper asserts that it disentangles homogeneity and heterogeneity, yet the evidence provided relies on feature visualization of specific samples. A statistically robust analysis is necessary to substantiate this claim.

While the framework allows for joint training across multiple subjects, the use of separate encoders for each subject may significantly increase computational costs, which should be addressed and optimized.

**Questions:**

Can you provide the specific list of 407 Mandarin syllables and explain how they represent nearly all Mandarin characters?

How and why are the private and shared codebooks defined and utilized within your framework?

Have you considered the computational cost implications of using separate encoders for each subject, and are there plans to optimize this aspect?

How does your work differentiate from the 2023 study "A high-performance brain-to-sentence decoder for logosyllabic language," which also uses a dataset with 407 syllables?

What is the total duration of the sEEG data in the dataset you provided?

---

> ### Author Response · Authors · 2024-11-17
> **Response to Reviewer mtfq (1/3)**
>
> ## Response to Reviewer mtfq (1/3)
> ---
>
> ## Reply to Weakness
>
> > **W1: The statement that a "comprehensive set of 407 Mandarin syllables covers nearly all Mandarin characters" requires further clarification, e.g., providing the specific vocabulary list.**
>
> **R**: Mandarin syllables consist of an initial and a final, and to ensure comprehensive coverage of all possible pronunciations with their corresponding tones, our reading material includes all potential combinations of initials and finals. Due to space constraints, we have included the complete vocabulary list in Tables A4-A7 of the revised manuscript, and we have highlighted the relevant entries in red for ease of reference.
>
> > **W2: The concept of private and shared codebooks is introduced in the paper but lacks a detailed explanation. A more comprehensive rationale and description are needed to clarify how and why these codebooks are defined and utilized.**
>
> **R**: The intuition behind our proposed H2DiLR is straightforward: although physiological and instrumental differences exist among different subjects (heterogeneity), the same brain regions in different individuals exhibit similar functions during the tone production process (homogeneity).
>
> Within H2DiLR, we implement this balance through two types of codebooks. The shared codebook stores homogeneous information accessible to all subjects and captures the commonalities across individuals, while subject-specific private codebooks store heterogeneous information unique to each subject. Section 3.3 introduces the structure and role of these codebooks.
>
> The mathematical definition and training process for homogeneity-heterogeneity disentanglement (H2D) is detailed in Section 3.4, with a visual illustration provided in **Figure 3**. Specifically, neural representations from each encoder are quantized through either the shared or private codebook based on a similarity ranking outlined in **Equation 4**. The updating rules for the shared and private codebooks are given in **Equations 4 and 5**, respectively, while the overall loss functions for encoder updates are detailed in Equation 6 and Equation 7.
>
> > **W3: Sharing the data and code would greatly enhance the validation and reproducibility of the proposed method by the research community. This transparency would contribute significantly to the field and support further advancements.**
>
> **R**: We plan to release the code upon acceptance of this paper. As for the data, it is available from the corresponding authors upon reasonable request, ensuring compliance with ethical and privacy guidelines.
>
> > **W4: The claim of being "the first" to work with 407 Mandarin syllables is inaccurate, as the 2023 study titled "A high-performance brain-to-sentence decoder for logosyllabic language" also used a dataset with 407 syllables. Although this work is cited, the related work section lacks a detailed discussion of this study.**
>
> **R**: Thanks for mentioning the relevant work. Although our study and the 2023 work by Feng et al. utilize the same dataset setting with 407 syllables, we are indeed the first to perform **unified tone decoding across subjects** rather than the first to use the 407 syllables setting. Moreover, the objectives and approaches differ significantly. Feng et al. focused on developing a sentence-level decoding system tailored to **each individual subject**, employing a **heterogeneous training paradigm**. In contrast, our work aims to improve tone decoding performance by disentangling and learning both heterogeneous and homogeneous neural representations **across all subjects**. This approach, which we refer to as **Homo-heterogeneity Disentanglement**, allows us to capture shared and distinct neural patterns more effectively without increasing model complexity per subject.

---

> ### Author Response · Authors · 2024-11-17
> **Response to Reviewer mtfq (2/3)**
>
> ## Response to Reviewer mtfq (2/3)
> ---
>
> > **W5: The paper asserts that it disentangles homogeneity and heterogeneity, yet the evidence provided relies on feature visualization of specific samples. A statistically robust analysis is necessary to substantiate this claim.**
>
> **R**: The visualization result verifies that the public codebooks capture homogeneity in terms of tone decoding. We also conducted another subject classification task to verify whether the heterogeneous representations learned by our proposed H2D capture the heterogeneity. This task shows that the learned heterogeneous representation extracts sufficiently discriminative information to classify sEEG signals from different subjects.
> We conducted an additional ablation study to evaluate the generalization ability of the homogeneity captured by the public codebook. Specifically, we tested whether the public codebook, learned from three participants, could be generalized to a fourth participant. We followed the same experimental setup described in Sec. 4 with a leave-one-subject-out protocol.
>
> Firstly, we applied the homogeneity-heterogeneity disentanglement (H2D) on three subjects to acquire a public codebook. Then, we pre-trained the fourth subject using this previously acquired public codebook while keeping it frozen and updating only the private codebook. Finally, we performed neural decoding (ND) on the fourth subject. The decoding performance was compared with baselines and variants of our proposed H2DiLR. The results are summarized in the table below:
>
> | Method | Pre-training | Subject 1 | Subject 2 | Subject 3 | Subject 4 | Avg. |
> |---|:---:|:---:|:---:|:---:|:---:|:---:|
> | SPaRCNet | - | 34.94 $\pm$2.17 | 36.73 $\pm$5.88 | 27.10 $\pm$3.22 | 27.10 $\pm$1.43 | 31.47 $\pm$3.16 |
> | FFCL | - | 37.47 $\pm$2.40 | 37.71 $\pm$5.42 | 26.61 $\pm$3.18 | 29.47 $\pm$1.69 | 32.82 $\pm$3.17 |
> | TS-TCC | CL | 39.59 $\pm$1.68 | 41.47 $\pm$2.78 | 28.82 $\pm$1.75 | 32.33 $\pm$2.17 | 35.55 $\pm$2.09 |
> | CoST | CL | 43.95 $\pm$1.48 | 40.41 $\pm$4.77 | 31.02 $\pm$1.00 | 34.53 $\pm$2.39 | 37.47 $\pm$2.41 |
> | ST-Transformer | - | 39.02 $\pm$2.39 | 37.22 $\pm$4.68 | 27.51 $\pm$2.11 | 29.63 $\pm$2.98 | 33.35 $\pm$3.04 |
> | NeuroBERT | MM | 42.20 $\pm$1.86 | 43.10 $\pm$2.76 | 29.80 $\pm$1.98 | 32.65 $\pm$2.89 | 36.94 $\pm$2.37 |
> | BIOT | MM | 42.45 $\pm$6.99 | 40.90 $\pm$5.87 | 33.55 $\pm$2.95 | 33.88 $\pm$1.89 | 37.47 $\pm$2.26 |
> | H2DiLR (ours) | UPaNT+ H2D | 49.06 $\pm$ 2.15 | 47.84 $\pm$ 1.81 | 39.18 $\pm$1.68 | 38.61 $\pm$1.49 | 43.67 $\pm$1.78 |
> | H2DiLR (ours) | UPaNT only | 45.47 $\pm$3.04 | 44.65 $\pm$1.84 | 35.59 $\pm$2.37 | 35.76 $\pm$1.18 | 40.37 $\pm$2.11 |
> | H2DiLR (ours) | $\nu = 0$ | 43.61 $\pm$2.12 | 42.15 $\pm$1.63 | 34.26 $\pm$1.51 | 35.92 $\pm$1.43 | 38.98 $\pm$1.67 |
> | Generalization ablation | UPaNT + H2D | 45.98 $\pm$ 2.94 | 42.86 $\pm$ 2.66 | 35.81 $\pm$ 1.56 | 36.47 $\pm$ 2.49 | 40.28 $\pm$ 2.41 |
>
> As shown in the row labeled "Generalization Ablation" in the ablation table, we froze the public codebook after pre-training it on three subjects and evaluated it on the fourth. We observe that even with the public codebook frozen for three subjects, decoding on the fourth subject still outperforms other baselines. In fact, the performance is better than our H2DiLR model trained with $\nu = 0$ and is comparable to the performance achieved when trained with UPaNT only. This confirms that the representation captured by the public codebook can be effectively applied to a new subject.
>
> Additionally, we noted that the performance improvement on Subject 2 was smaller compared to the other subjects. We hypothesize that this could be due to the distinct electrode implantation location for Subject 2, which likely differs from that of the other participants.
>
> ---
> ## Reply to Question
>
> > **Q1: Can you provide the specific list of 407 Mandarin syllables and explain how they represent nearly all Mandarin characters?**
>
> **R**: We have included the full vocabulary list in Tables A4-A7 of the revised manuscript, and we have highlighted the relevant entries in red for ease of reference.

---

> > ### Author Response · Authors · 2024-11-17
> > **Response to Reviewer mtfq (3/3)**
> >
> > ## Response to Reviewer mtfq (3/3)
> > ---
> > > **Q2: How and why are the private and shared codebooks defined and utilized within your framework?**
> >
> > **R**: The intuition behind our proposed H2DiLR is straightforward: although physiological and instrumental differences exist among different subjects (heterogeneity), the same brain regions in different individuals exhibit similar functions during the tone production process (homogeneity).
> > Within H2DiLR, we implement this balance through two types of codebooks. The shared codebook stores homogeneous information accessible to all subjects and captures the commonalities across individuals, while subject-specific private codebooks store heterogeneous information unique to each subject. Section 3.3 introduces the structure and role of these codebooks.
> >
> > The mathematical definition and training process for homogeneity-heterogeneity disentanglement (H2D) is detailed in Section 3.4, with a visual illustration provided in Figure 3. Specifically, neural representations from each encoder are quantized through either the shared or private codebook based on a similarity ranking outlined in Equation 4. The updating rules for the shared and private codebooks are given in Equations 4 and 5, respectively, while the overall loss functions for encoder updates are detailed in Equations 6 and 7.
> >
> > > **Q3: Have you considered the computational cost implications of using separate encoders for each subject and plan to optimize this aspect?**
> >
> > **R**: Our proposed training paradigm enhances decoding performance by effectively learning both heterogeneous and homogeneous neural representations across all subjects without requiring additional parameters for each individual subject. The parameter scaling effect in our approach is consistent with existing heterogeneous training paradigms where separate encoders are trained for each subject. As detailed in Table A1, we employ a ConvNet with 1.55M parameters as the encoder and a transformer with 0.958M parameters as the neural decoder for each subject. Additionally, we are currently exploring methods to utilize a single encoder capable of accommodating all subjects, aiming for further efficiency improvements.
> >
> > > **Q4: How does your work differentiate from "A high-performance brain-to-sentence decoder for logosyllabic language", which also uses a dataset with 407 syllables?**
> >
> > **R**: Although our study and the 2023 work by Feng et al. utilize the same dataset with 407 syllables, the objectives and approaches differ significantly. Feng et al. focused on developing a sentence-level decoding system tailored to **each individual subject**, employing a **heterogeneous training paradigm**. In contrast, our work aims to improve tone decoding performance by disentangling and learning both heterogeneous and homogeneous neural representations **across all subjects**. This approach, which we refer to as **Homo-heterogeneity Disentanglement**, allows us to capture shared and distinct neural patterns more effectively without increasing model complexity per subject.
> >
> > > **Q5:What is the total duration of the sEEG data in the dataset?**
> >
> > **R**: The total duration of sEEG data across all subjects is approximately 13.5 hours.

---

> > > ### Author Response · Authors · 2024-11-18
> > > **Follow-Up on Rebuttal Response**
> > >
> > > Dear Reviewer mtfq,
> > >
> > > We are writing to kindly follow up on our responses to your insightful comments and feedback. We have provided detailed clarifications and additional analyses addressing your concerns and suggestions in our rebuttal.
> > >
> > > If there are any remaining questions or points you'd like us to address further, please do not hesitate to let us know. Your feedback has been invaluable in refining our work, and we would greatly appreciate it if you could share your thoughts on whether our responses meet your expectations.
> > >
> > > Thank you once again for your time and effort in reviewing our submission.
> > >
> > > Authors

---

> > > > ### Author Response · Authors · 2024-11-19
> > > > **Follow-Up on Rebuttal and Feedback **
> > > >
> > > > Dear Reviewer mtfq,
> > > >
> > > > We are reaching out to follow up on our previous message regarding your valuable feedback and our responses in the rebuttal.
> > > >
> > > > We greatly value your opinion on whether our responses have adequately addressed your concerns. If there are any questions or further points you'd like us to address, we would be more than happy to provide additional details.
> > > >
> > > > Thank you once again for your time and effort in reviewing our work.
> > > >
> > > > Authors
> > > >
> > > >  

---

> > > > > ### Author Response · Authors · 2024-11-21
> > > > > **Follow-Up on Addressed Concerns**
> > > > >
> > > > > Dear Reviewer mtfq,
> > > > >
> > > > > We wanted to kindly follow up again regarding our responses to your comments and feedback on our submission. We have thoroughly addressed the concerns you raised and provided additional analyses in our rebuttal.
> > > > >
> > > > > If you find our responses satisfactory, we would greatly appreciate it if you could confirm whether all your concerns have been resolved. Should you feel our revisions merit reconsideration of your evaluation score, we kindly request that you update your score to reflect this.
> > > > >
> > > > > Thank you again for your time and effort in reviewing our work.
> > > > >
> > > > > Best regards,
> > > > > Authors

---

> > > > > > ### Author Response · Authors · 2024-11-23
> > > > > > **Follow-Up on Addressed Concerns**
> > > > > >
> > > > > > Dear Reviewer mtfq,
> > > > > >
> > > > > > We are writing to kindly follow up on our previous messages regarding your feedback on our responses to your comments. As the deadline for the review process (11/26) is fast approaching, we would greatly appreciate it if you could let us know whether our responses have addressed your concerns.
> > > > > >
> > > > > > If you find our responses satisfactory, we would greatly appreciate it if you could confirm whether all your concerns have been resolved. Should you feel our revisions merit reconsideration of your evaluation score, we kindly request that you update your score to reflect this.
> > > > > >
> > > > > > Thank you again for your time and effort in reviewing our work.
> > > > > >
> > > > > > Best regards, Authors

---

> > > > > > > ### Comment · Reviewer_mtfq · 2024-11-26
> > > > > > > **Official Comment by Reviewer mtfq**
> > > > > > >
> > > > > > > Thank you for your patience. I have thoroughly reviewed your paper, and based on your previous responses, I have outlined the following questions that need to be addressed. Please note that no additional experiments are required.
> > > > > > >
> > > > > > > **Question 1:** Thank you for the feedback. I have thoroughly read your paper. I am curious about how you determined your vocabulary list and what dependencies it involved. You consistently emphasize in sections A4-A7, "The first/second/... set of 407 Mandarin Chinese characters used as reading material." My understanding is that the combined vocabulary lists of the four subjects should cover the total number of syllables. Is that correct?
> > > > > > >
> > > > > > > **Question 2:** I noticed what I consider a critical issue: your introduction is quite confusing, and it is unclear what exactly you are doing after reading it. Is the EEG segment you input of a specific length, and does this EEG signal correspond to the reading time? Most importantly, are you performing a classification task or a regression task? If it is a classification task, how many categories are there? If it is a regression task, are you decoding speech end-to-end, or are you only decoding text? Unfortunately, readers cannot grasp this information from your introduction.
> > > > > > >
> > > > > > > I believe that instead of overemphasizing what you consider your innovation, "homogeneity-heterogeneity separation learning," it would be more beneficial to clearly outline the specific tasks and details of your work. Thank you! If I am mistaken, please feel free to correct me.
> > > > > > >
> > > > > > > **Question 3:** Additionally, I think there is an overstatement when you claim in the related works section of your introduction: "We are the first to decode unified tones across a comprehensive set of 407 Mandarin Chinese syllables." As you mentioned, there are over 50,000 Chinese characters, and for each subject, your character count is quite limited. You continually emphasize syllables, yet I believe the proposed homogeneity-heterogeneity separation learning is not strongly related to syllables.
> > > > > > >
> > > > > > > Moreover, the work titled "A high-performance brain-to-sentence decoder for logosyllabic language" has already explored decoding across 407 Mandarin syllables in 2023. You are not the first, and your claim may lead to misunderstandings.
> > > > > > >
> > > > > > > **Question 4:** In your response, you mentioned: "The intuition behind our proposed H2DiLR is straightforward: although physiological and instrumental differences exist among different subjects (heterogeneity), the same brain regions in different individuals exhibit similar functions during the tone production process (homogeneity)." Since heterogeneity and homogeneity are your main innovations and research motivations, this requires references or theoretical studies as evidence to support your claims.
> > > > > > >
> > > > > > > **Question 5:** You utilize a ConvNet with 1.55M parameters as the encoder and a Transformer with 0.958M parameters as the neural decoder for each subject. This implies you implement joint training for multiple subjects in a multi-task learning manner. A perfect cross-subject paradigm should not require additional parameters and should support cross-subject testing, which your design evidently does not. The drawback is that with each additional subject, the total number of parameters increases further.
> > > > > > >
> > > > > > > **Question 6:** Regarding the definition of private and shared codebooks, what is the total number of sEEG channels across multiple subjects? I am confused about the private codebook with K=32.

---

> > > > > > > > ### Author Response · Authors · 2024-11-27
> > > > > > > > **Response to Reviewer mtfq' FeedBack (PART 2/3)**
> > > > > > > >
> > > > > > > > > **Q3: I think there is an overstatement when you claim in the related works section of your introduction: "We are the first to decode unified tones across a comprehensive set of 407 Mandarin Chinese syllables." As you mentioned, there are over 50,000 Chinese characters, and for each subject, your character count is quite limited. You continually emphasize syllables, yet I believe the proposed homogeneity-heterogeneity separation learning is not strongly related to syllables. Moreover, the work titled "A high-performance brain-to-sentence decoder for logosyllabic language" has already explored decoding across 407 Mandarin syllables in 2023. You are not the first, and your claim may lead to misunderstandings.**
> > > > > > > >
> > > > > > > > **R**: We would like to clarify that character count and pronunciation count are fundamentally different concepts in Mandarin. The mapping from pronunciations to characters is one-to-many. For example, the pronunciation 'tā' (syllable) corresponds to multiple characters, such as '他' (he) and '塌' (breakdown), which are homophones. While Mandarin encompasses an extensive inventory of over 50,000 characters, the pronunciation of these characters is derived from a finite set of 407 syllables, each with one of four tones. Since our objective is tone decoding, our study focuses on covering all possible pronunciations to ensure comprehensive and practical relevance.
> > > > > > > >
> > > > > > > > The heterogeneity due to both physiological and instrumental factors presents a significant challenge in invasive brain neural decoding. Physiological variations among subjects, along with differences in electrode configurations due to diverse implantation conditions, hinder unified cross-subject decoding. Conversely, developing a separate heterogeneous model for each subject results in poor generalization due to data scarcity. **Our proposed Homogeneity-Heterogeneity Disentanglement (H2D) framework addresses these challenges by facilitating a generic neural decoding approach that disentangles shared and distinct neural representations, thereby enabling improved tone decoding performance across subjects.**
> > > > > > > >
> > > > > > > > Although our study and the 2023 work by Feng et al. utilize the same dataset with 407 syllables, **the objectives and methodologies are fundamentally different**. Feng et al. focused on constructing a **sentence-level decoding system** tailored to individual subjects using a **heterogeneous training paradigm**. In contrast, our work emphasizes cross-subject tone decoding by leveraging H2D to simultaneously learn homogeneous and heterogeneous neural representations across all subjects. This approach not only enhances decoding accuracy but also avoids increasing model complexity per subject.
> > > > > > > >
> > > > > > > > We acknowledge the potential for misunderstanding and will revise the manuscript to clarify our claims, ensuring the unique contributions of our work are accurately represented.
> > > > > > > >
> > > > > > > > ---
> > > > > > > >
> > > > > > > > > **Q4: In your response, you mentioned: "The intuition behind our proposed H2DiLR is straightforward: although physiological and instrumental differences exist among different subjects (heterogeneity), the same brain regions in different individuals exhibit similar functions during the tone production process (homogeneity)." Since heterogeneity and homogeneity are your main innovations and research motivations, this requires references or theoretical studies as evidence to support your claims.**
> > > > > > > >
> > > > > > > > **R**: Thanks for highlighting this point. To substantiate our claims regarding the homogeneity and heterogeneity in neural activity during tone production, we have incorporated the following references into our manuscript: Studies such as [1-3] have demonstrated that the same brain regions in different individuals exhibit consistent functional roles during speech and tone production. For example, the ventral sensorimotor cortex (vSMC) and superior temporal gyrus (STG) are consistently implicated in speech processing across subjects, supporting the homogeneity assumption.
> > > > > > > >
> > > > > > > > We appreciate your suggestion and have ensured that the manuscript now reflects these foundational studies to better contextualize our manuscript.

---

> ### Author Response · Authors · 2024-11-27
> **Response to Reviewer mtfq' FeedBack (PART 1/3)**
>
> ## Reply to Question
>
> ---
> > **Q1: I am curious about how you determined your vocabulary list and what dependencies it involved. You consistently emphasize in sections A4-A7, "The first/second/... set of 407 Mandarin Chinese characters used as reading material." My understanding is that the combined vocabulary lists of the four subjects should cover the total number of syllables.**
>
> **R**: Thanks for the inspiring comment, and your understanding is correct. The syllables presented in Tables A4-A7 indeed total 407. During our experiments, each participant read carrying sentences containing a complete set of all 407 target syllables three times. Each Mandarin syllable is composed of an initial, a final, and a tone, which collectively define its pronunciation. The reading material used in the study was **designed by Mandarin linguists to ensure comprehensive coverage of all possible pronunciations, along with their corresponding tones**. While Mandarin has a vast lexicon of over 50,000 characters, its pronunciations map to one of these 407 unique syllables, each paired with one of the four tones. This design guarantees that the reading material is sufficient to represent the full spectrum of Mandarin pronunciation, making it an optimal choice for our decoding task.
>
> ---
>
> > **Q2: A critical issue: your introduction is quite confusing, and it is unclear what exactly you are doing after reading it. Is the EEG segment your input of a specific length, and does this EEG signal correspond to the reading time? Most importantly, are you performing a classification task or a regression task? If it is a classification task, how many categories are there? If it is a regression task, are you decoding speech end-to-end, or are you only decoding text? I believe that instead of overemphasizing what you consider your innovation, "homogeneity-heterogeneity separation learning," it would be more beneficial to outline the specific tasks and details of your work clearly.**
>
> **R**: Thank you for your detailed feedback. However, we respectfully disagree with the assertion that our introduction is unclear. In this section, we structured our narrative as follows:
>
>    * **Background and Importance**: We introduced the linguistic background of tonal languages such as Mandarin, highlighting the critical role of tone in semantic interpretation and its importance for accurate brain-to-text decoding (lines 32-48).
>
>    * **Limitations of Prior Work**: We reviewed existing tone-decoding approaches using both non-invasive and invasive neural recordings, identifying key challenges, particularly the heterogeneity caused by physiological and instrumental factors (lines 49-60).
>
>    * **Motivation and Framework**: We presented the motivation for our study and an overview of our proposed H2DiLR framework, accompanied by Figure 1, to illustrate its structure and purpose (lines 61-90).
>
>    * **Contributions and Results**: A brief summary of our experimental setup, key findings, and contributions were provided (lines 91-109).
>
> While we acknowledge the reviewer's concerns, we believe that the technical specifics of the task and experimental design are better suited to the Methods and Experiments sections, where they are thoroughly explained. To directly address the reviewer's specific questions:
>
> * **Q2.1: Is the EEG segment your input of a specific length, and does this EEG signal correspond to the reading time?**
>
> $\quad$ **R:** As described in Appendix A, the input sEEG signals are fixed-length 1-second segments aligned with the syllable reading process.
>
> * **Q2.2: Are you performing a classification task or a regression task? If it is a classification task, how many categories are there? If it is a regression task, are you decoding speech end-to-end, or are you only decoding text?**
>
> $\quad$ **R:** The decoding task in the second stage is a four-class classification task corresponding to the four tones of Mandarin syllables.
>
> We appreciate the reviewer's suggestion and will revise the introduction to ensure clearer presentation, particularly highlighting the nature of the task earlier in the narrative.

---

> ### Author Response · Authors · 2024-11-27
> **Response to Reviewer mtfq' FeedBack (PART 3/3)**
>
> > **Q5: You utilize a ConvNet with 1.55M as the encoder and a Transformer with 0.958M as the neural decoder for each subject. This implies you implement joint training for multiple subjects in a multi-task learning manner. A perfect cross-subject paradigm should not require additional parameters and should support cross-subject testing, which your design evidently does not. The drawback is that with each additional subject, the total number of parameters increases further.**
>
> **R**: Thanks for raising this critical point. Our model explicitly targets the heterogeneity introduced by both physiological and instrumental factors in invasive brain neural decoding. For example, due to variations in subject conditions, **different participants have varying numbers of implanted electrodes, resulting in differences in channel counts across subjects**. This variability presents a unique challenge that distinguishes our work from tasks like EEG decoding, where all subjects typically share a standardized electrode configuration (e.g., the 10-20 system).
>
> Given these differences, designing a universal decoder capable of handling the variability in electrode configurations across subjects is inherently complex. While our current approach employs subject-specific decoders to address this challenge effectively, we fully acknowledge that a universal network capable of supporting cross-subject testing without increasing the number of parameters per subject would be an ideal solution.
>
> We are actively exploring methods to develop a single encoder-decoder framework that can accommodate all subjects, aiming to improve efficiency and scalability in future iterations of our work. We appreciate the reviewer's insight and agree that achieving such a universal architecture is an important direction for further research. However, this does not diminish the contribution of our proposed homogeneity-heterogeneity disentanglement (H2D) learning paradigm, which offers a novel and effective approach to tackling a fundamental challenge in neural decoding.
>
> ---
>
> > **Q6:Regarding the definition of private and shared codebooks, what is the total number of sEEG channels across multiple subjects? I am confused about the private codebook with K=32?**
>
> **R**: The sEEG channel counts for the participants in our study are A, B, C, and D, respectively. Each subject’s data is represented as $\mathcal{X}_{i} \in \mathbb{R}^{N_i \times T \times C_i}$, where $N_i$ is the number of data samples, $C_i$ is the number of channels (which varies across subjects), and $T$ is the segment length, which is consistent for all subjects.
>
> To address this heterogeneity in channel counts, neural recordings $\mathcal{X}$ are first mapped into a latent feature space, $z = E_{i}(x; \theta_i) \in \mathbb{R}^{L \times D}$, using a set of subject-specific encoders ${E_{i}(\cdot; \theta_i)}_{i=1}^m$. Here, $L$ is the sequence length of tokens in the latent space, and $D$ is the dimensionality of each token.
>
> For quantization, we employ a public codebook with $K=128$ and private codebooks with $K=32$ learnable key-value pairs, represented as $\mathcal{C} = {(k, e(k))}_{k \in [K]}$, where each code $k$ has an associated learnable embedding $e(k) \in \mathbb{R}^{D}$. Each token in $z$ is discretized into a code using the quantization function described in Equation 1 of our paper. This discretization process is crucial for capturing both subject-specific (private) and shared features across subjects, enabling the disentanglement of heterogeneity and homogeneity.
>
> The private codebook with $K=32$ is specifically designed to handle subject-specific variations in neural recordings, ensuring effective encoding of private information. By contrast, the shared codebook is responsible for modeling similarities across subjects. Together, they enable our homogeneity-heterogeneity disentanglement (H2D) approach to address the variability in channel counts and individual differences.
>
> ### References
>
> [1] Si, Xiaopeng, et al. "Cooperative cortical network for categorical processing of Chinese lexical tone." Proceedings of the National Academy of Sciences (2017): 12303-12308.
>
> [2] Oganian, Yulia, et al. "A speech envelope landmark for syllable encoding in human superior temporal gyrus." Science advances (2019): eaay6279.
>
> [3] Lu, Junfeng, et al. "Neural control of lexical tone production in human laryngeal motor cortex." Nature Communications (2023): 6917.
>
> ---
>
> Thank you for your detailed feedback. We believe our clarifications address your concerns and strengthen the technical rigor of our work.  We hope this work can reach a broader audience within the community, as we feel it tackles an important challenge.
>
> Your evaluation is highly valuable to us, and we respectfully invite you to consider increasing your rating. We welcome any further questions or points for clarification.
>
> We look forward to your feedback and deeply appreciate your consideration.
>
> Warm regards,
>
> Authors

---

> > ### Author Response · Authors · 2024-11-29
> > **Follow-Up on Addressed Concerns**
> >
> > Dear Reviewer mtfq,
> >
> > We are writing to kindly follow up on your feedback on our responses to your questions. As the deadline for the discussion process is fast approaching, we would greatly appreciate it if you could let us know whether our responses have addressed your concerns.
> >
> > If you find our responses satisfactory, we would greatly appreciate it if you could confirm whether all your concerns have been resolved. Should you feel our revisions merit reconsideration of your evaluation score, we kindly request that you update your score to reflect this.
> >
> > Thank you again for your time and effort in reviewing our work.
> >
> > Best regards,
> >
> >  Authors

---

> ### Author Response · Authors · 2024-12-02
> **Follow-Up on Feedback on Our Response**
>
> Dear Reviewer mtfq,
>
> As the review deadline of December 2 approaches, we wanted to kindly follow up regarding your feedback on our response to your questions.
>
> We would greatly appreciate it if you could confirm whether all your concerns have been resolved. Should you feel our response merits reconsideration of your evaluation score, we kindly request that you update your score to reflect this.
>
> Thank you again for your time and effort in reviewing our work.
>
> Best regards,
>
> Authors

---

> > ### Author Response · Authors · 2024-12-02
> >
> > Dear Reviewer mtfq,
> >
> > We are reaching out to follow up on our previous message regarding your valuable feedback and our responses in the rebuttal.
> >
> > We greatly value your opinion on whether our responses have adequately addressed your concerns. If there are any questions or further points you'd like us to address, we would be more than happy to provide additional details.
> >
> > Thank you once again for your time and effort in reviewing our work.
> >
> > Authors

---

> > > ### Author Response · Authors · 2024-12-03
> > > **Urgent Follow-Up: Discussion Deadline Approaching**
> > >
> > > Dear Reviewer mtfq,
> > >
> > > I am writing to kindly follow up as the discussion deadline is now less than 10 hours away. Your feedback and potential reassessment are crucial to the evaluation of our work.
> > >
> > > We deeply appreciate the time and effort you have already dedicated to reviewing our submission. If there are any additional concerns or points you would like us to address, we would be more than happy to provide further clarifications promptly.
> > >
> > > Thank you for your valuable contribution to this process.
> > >
> > > Best regards,
> > >
> > > On behalf of the Authors

---

> > > > ### Author Response · Authors · 2024-12-03
> > > >
> > > > Dear Reviewer mtfq,
> > > >
> > > > We once again thank you for your time and effort in reviewing our work and providing valuable feedback. We believe our responses have thoroughly addressed your concerns, and we kindly invite you to consider increasing your rating after reviewing our clarifications.
> > > >
> > > > Your feedback and evaluation are greatly appreciated, and we value the opportunity to improve our work based on your insights.
> > > >
> > > > Thank you again for your support in this process.
> > > >
> > > > Best regards,
> > > >
> > > > Authors

---

### Meta-Review · Area_Chair_oWjs · 2024-12-23

**Metareview:**

This paper introduces a new method for lexical tone decoding from intracranial EEG (iEEG) recordings. The main idea behind their approach is to create shared and private codebooks with a VQ-VAE based tokenization method.

The reviewers agreed on the significance of the problem and the value of new datasets in this area. However, multiple reviewers brought up concerns about needing further justification for the claim that their approach  disentangles homogeneous and heterogeneous features across different individuals. Additionally, one reviewer had major concerns on the fairness of comparisons across different ablations of their approach (hetero training vs combined hetero-homo training). If indeed the hetero-only model was not appropriately tuned to have a large enough codebook (as in other previous works), then the improvements in hetero+homo could potentially be overstated.

After the rebuttal, the reviewers were split, with two favoring acceptance and two voting against. While the AC does agree that the claims about disentanglement may be overstated (and should be revised), the contributions appear to be significant over previous methods. Furthermore, the additional experiments in the Appendix on seizure prediction provide another example of their approach beating other sota methods, which is helpful for building confidence that the approach can generalize to other tasks. The contribution of the dataset will also be useful for the field.

The authors should conduct the experiments on larger codebooks that are requested by Aheb, and include all of the changes suggested by the reviewers and discussed during the response period.

**Additional Comments On Reviewer Discussion:**

There was a lengthy discussion during the rebuttal period. After some back and forth, one reviewer decided to increase their score to accept. AheB raised valid concerns about the size of the codebook for the hetero variant of their method, which were unaddressed by the authors.

---

### Decision · Program_Chairs · 2025-01-22

Accept (Poster)